# APEBench: A Benchmark for Autoregressive Neural Emulators of PDEs

**Felix Koehler**
Technical University of Munich
Munich Center for Machine Learning
f.koehler@tum.de

**Simon Niedermayr**
Technical University of Munich
simon.niedermayr@tum.de

**Rüdiger Westermann**
Technical University of Munich
westermann@tum.de

**Nils Thuerey**
Technical University of Munich
nils.thuerey@tum.de

## Abstract

We introduce the **A**utoregressive **P**DE **E**mulator Benchmark (APEBench), a comprehensive benchmark suite to evaluate autoregressive neural emulators for solving partial differential equations. APEBench is based on JAX and provides a seamlessly integrated differentiable simulation framework employing efficient pseudo-spectral methods, enabling 46 distinct PDEs across 1D, 2D, and 3D. Facilitating systematic analysis and comparison of learned emulators, we propose a novel taxonomy for unrolled training and introduce a unique identifier for PDE dynamics that directly relates to the stability criteria of classical numerical methods. APEBench enables the evaluation of diverse neural architectures, and unlike existing benchmarks, its tight integration of the solver enables support for differentiable physics training and neural-hybrid emulators. Moreover, APEBench emphasizes rollout metrics to understand temporal generalization, providing insights into the long-term behavior of emulating PDE dynamics. In several experiments, we highlight the similarities between neural emulators and numerical simulators. The code is available at https://github.com/tum-pbs/apebench and APEBench can be installed via pip install apebench.

## 1 Introduction

The language of nature is written in partial differential equations (PDEs). From the behavior of subatomic particles to the earth's climate, PDEs are used to model phenomena across all scales. Typically, approximate PDE solutions are computed with numerical simulations. Almost all relevant simulation techniques stem from the *consistent* discretization involving symbolic manipulations of the differential equations into a discrete computer program. This laborious task yields algorithms that converge to the continuous dynamics for fine resolutions. For realistic models, established techniques require immense computational resources to attain high accuracy. Recent advances of machine learning-based *emulators* challenge this. Purely data-driven or with little additional constraints and symmetries, neural networks can surpass traditional methods in the accuracy-speed tradeoff (Kochkov et al., 2021; List et al., 2022; Lam et al., 2022).

The field of neural PDE solvers advanced rapidly over the past years, applying convolutional architectures (Tompson et al., 2017; Thuerey et al., 2020; Um et al., 2020), graph convolutions (Pfaff et al., 2021; Brandstetter et al., 2022), spectral convolutions (Li et al., 2021), or mesh-free approaches (Ummenhofer et al., 2020; Wessels et al., 2020) to replace or enhance classical numerical solvers. However, the relationship between classical solvers, which supply training data, and neural emulators,

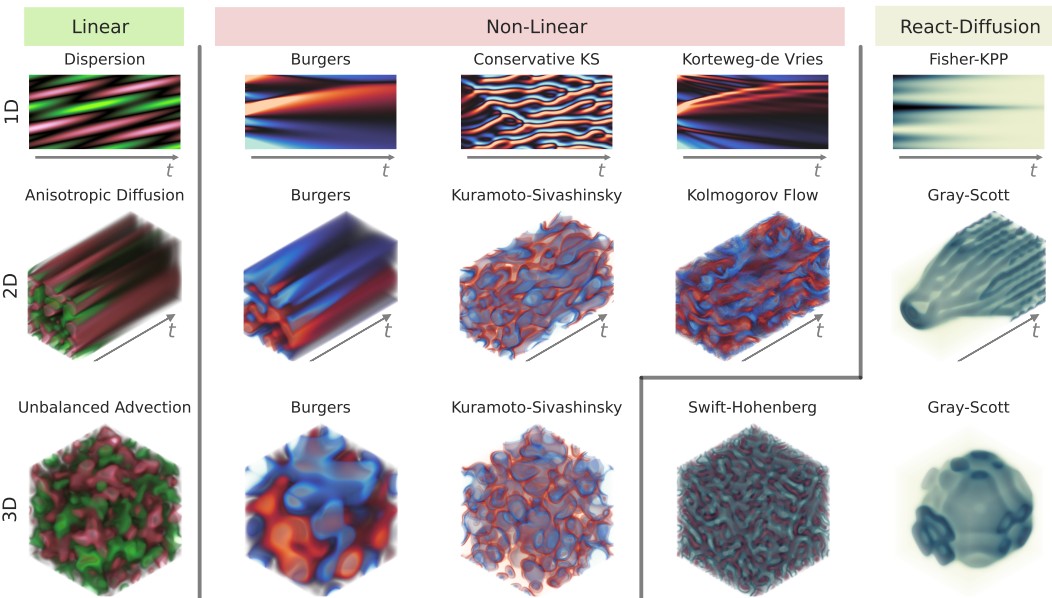

Figure 1: APEBench provides an efficient pseudo-spectral solver to simulate 46 PDE dynamics across one to three spatial dimensions. Shown are examples visualized with APEBench's custom volume renderer.

which attempt to emulate their behavior, is often underexplored. For example, convolutional networks bear a strong resemblance to finite difference methods, and spectral networks can be thought of as pseudo-spectral techniques (McCabe et al., 2023). These parallels suggest that a better understanding of this interplay could help inform how emulator architectures are designed and how effectively neural emulators can learn from classical solvers.

To help address these questions, we introduce APEBench, a new benchmark suite designed to complement existing efforts in evaluating autoregressive neural emulators for time-dependent PDEs. While previous benchmarks such as PDEBench (Takamoto et al., 2022) and PDEArena (Gupta and Brandstetter, 2023) have provided valuable insights into architectural comparisons based on fixed datasets, APEBench aims to extend these efforts by focusing on emulator-simulator interaction via supporting neural-hybrid approaches and emphasizing training using differentiable physics. Additionally, we place particular focus on unrolled training and rollout metrics, which have been less systematically explored in other benchmarks.

The key innovation of APEBench lies in its tight integration of a highly efficient pseudo-spectral solver. This method is used both for procedural data generation and as a differentiable solver the networks can dynamically interact with during training. APEBench offers four key contributions:

- **Large Selection of Dynamics**: The benchmark offers a wide array of 46 PDE dynamics that allow for drawing analogies between classical and learned approaches.
- **Unique Dynamics Identifier**: For each distinct type of dynamics, APEBench provides a unique set of identifiers that encodes its difficulty of emulation.
- **Differentiable Simulation Suite**: We provide a novel (differentiable) JAX-based simulation framework employing efficient pseudo-spectral methods, which seamlessly integrates into emulator training and serves as a fast data generator.
- **Taxonomy and Metrics for Unrolling Methodologies**: We propose a broad and systematic framework for analyzing the impact of different training paradigms on emulator performance.

Our benchmark includes recipes that rapidly adapt to new architectures and training methodologies. Datasets are re-generated procedurally (and deterministically) in seconds on modern hardware. This avoids the need to distribute fixed datasets, improving adoption, and allows for quick modification of the underlying phyiscs. For visual analysis of the emergent structures, the benchmark is accompanied by a fast volume visualization module. This module seamlessly interfaces with the PDE dynamics to provide immediate feedback for emulator development in 2D and 3D.

## 2 From Classical Numerics to Learned Emulation

We first discuss a motivational example that illustrates several key aspects of APEBench, namely **rollout metrics**, **training methodologies**, and **PDE identifiers**. We choose the simple case of one-dimensional advection with periodic boundary conditions and velocity $c$,

$$\partial_t u + c\partial_x u = 0 \qquad \text{with} \qquad u(t, x = 0) = u(t, x = L),$$

which admits an *analytical solution* where the initial condition moves with $c$ over the domain $\Omega = (0, L)$. Let $\mathcal{P}_h$ represent a discrete analytical time stepper that operates on a fixed resolution with $N$ equidistantly spaced degrees of freedom and time step size $\Delta t$. This *simulator* advances a discrete state $u_h \in \mathbb{R}^N$ to a future time, i.e., $u_h^{[t+1]} = \mathcal{P}_h(u_h^{[t]})$. Emulator learning is the task of approximating this time stepper using a neural network $f_\theta \approx \mathcal{P}_h$.

The simplest possible network is a linear convolution with a kernel size of two as

$$f_\theta(u_h) = w_\theta \star u_h,$$

where $\star$ denotes cross-correlation. We frame finding $w_\theta = [\theta_{\text{center}}, \theta_{\text{right}}]^T \in \mathbb{R}^2$ to approximate $\mathcal{P}_h$ with $f_\theta$ as a data-driven learning problem, using trajectories produced by the analytical time stepper. If the neural emulator predicts one step into the future, the learning problem over the two-dimensional parameter space $\theta \in \mathbb{R}^2$ becomes *convex*. Since even-sized convolutions are typically biased to the right, one could suspect that the learned minimum of such a problem is given by the first-order upwind (FOU) method. This numerical (non-analytical) time stepper is found via a *consistent* approach to discretizing the advection equation using a Taylor series. If we assume $c < 0$, it leads to $\theta_{\text{center}} = 1 + c\frac{\Delta t}{\Delta x}$ and $\theta_{\text{right}} = -c\frac{\Delta t}{\Delta x}$. However, despite convexity the learned solution is different. In Figure 2, we benchmark the long-term

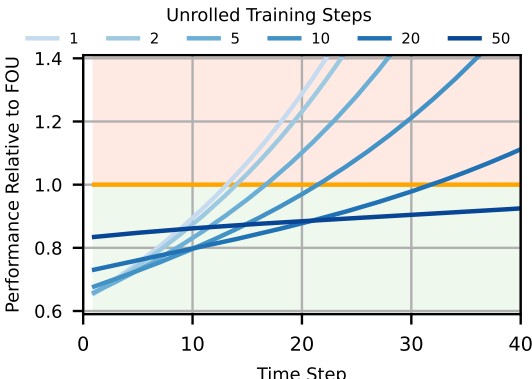

Figure 2: Test rollout performance of linear convolution emulators when learned with different training rollout lengths relative to the performance of a FOU method. All learned emulators surpass the numerical method in an initial operating regime. More unrolling improves long-term accuracy for a small sacrifice in short-term performance.

performance of the learned emulator relative to the FOU scheme. It is superior to the numerical method, with lower errors for the first 13 steps. Eventually, it diverges because it is *not consistent*. We can improve the long-term performance of the emulator by training it to predict multiple steps autoregressively. We call this approach *unrolled* training in the following; it more closely aligns with the inference task of long-term accuracy. Indeed, doing so enhances the performance for a small sacrifice in short-term accuracy. The learned emulator improves in *temporal generalization*, i.e., it runs stably and accurately for more time steps. For example, when unrolling for 20 steps during training, the learned solution still performs better after 30 steps while having an 11% increased error at the first step. In the two-dimensional parameter space, more unrolling moves the learned stencil closer to the FOU scheme. The distance reduces from 0.034 to 0.024 to 0.01 for 1-step, 10-step, and 50-step training, respectively. This indicates that unrolling successfully injects knowledge about becoming a good simulator.

The FOU stencil depends only on the Courant-Friedrichs-Lewy (CFL) number $c\frac{\Delta t}{\Delta x}$. It represents a way to assess the *difficulty* of the advection problem. With APEBench, we generalize it and compute similar stability numbers as intuitive identifiers for all available dynamics. By doing so, we obtain a minimal set of information to describe an experiment, which serves as an exchange protocol in our benchmark suite.

Our motivational example reveals that even combining simple (linear) PDEs and linear emulators leads to non-trivial learning tasks offering interesting insights. We also see that the emulators share similarities with classical numerical methods. In this case, there is a strong relation between convolutional networks and finite difference methods. However, since the emulator's free parameters result from a data-driven optimization problem, not a human-powered symbolic manipulation, they may deviate from the strict assumptions underlying traditional schemes. Learned emulators can use

this to their advantage, and outperform their numerical counterparts for a specific operating regime, i.e., a certain test rollout length. Since this superiority varies with unrolled training steps and is certainly dependent on the underlying dynamics (in terms of its *difficulty*), we emphasize that using rollout metrics is important to understand the temporal behavior of neural emulators. In summary, APEBench provides a suite that holistically assesses all ingredients of the emulator learning pipeline, including a highly accurate solver.

## 3 Related Work

**Neural PDE Solvers** Early efforts in neural PDE solvers focused on learning the continuous solution function for an (initial-)boundary value problem via a coordinate network (Dissanayake and Phan-Thien, 1994; Lagaris et al., 1998). With the rise of automatic differentiation capabilities in modern machine learning frameworks, this approach experienced a resurgence under the name of *Physics-Informed Neural Networks* (PINNs) (Raissi et al., 2019). However, PINNs do not use autoregressive inference. Early works on stationary neural emulators include solutions to the pressure Poisson equation (Tompson et al., 2017) and fluid simulations (Thuerey et al., 2020). Notable works employing the autoregressive paradigm are Brandstetter et al. (2022) using supervised data from numerical simulators. Successful unsupervised approaches for autoregressive emulators are Geneva and Zabaras (2020) and Wandel et al. (2021). Seminal works highlighting the supremacy of neural-hybrid emulators are Um et al. (2020) and Kochkov et al. (2021). Oftentimes, the architectures employed are inspired by image-to-image tasks in computer vision. A closely-related line of work utilizes *neural operators* (Kovachki et al., 2023) which impose stronger mathematical requirements on emulators and their architectures, such as representation-equivariance (Bartolucci et al., 2023; Raonic et al., 2023). The most popular operator architecture in autoregressive settings is the Fourier Neural Operator (FNO) (Li et al., 2021) with numerous modifications and improvements existing (Tran et al., 2023; McCabe et al., 2023). Like implicit methods for numerical simulators, autoregressive models can have internal iterations. For example, this includes autoregressive diffusion models (Kohl et al., 2023) or iterative refinements (Lippe et al., 2023).

**Neural Emulator Benchmarks and Datasets** Notable benchmark papers comparing emulator architectures are PDEBench (Takamoto et al., 2022) and PDEArena (Gupta and Brandstetter, 2023). Benchmarks based on more complicated fluid simulations include Luo et al. (2023), Bonnet et al. (2022), and Janny et al. (2023). BubbleML (Hassan et al., 2023) focuses on two-phase boiling problems. All the aforementioned benchmark papers release fixed and pre-computed datasets. APEBench is the first to tightly integrate an efficient reference solver, which procedurally generates all data. Moreover, we thereby uniquely enable benchmarking approaches involving differentiable physics, e.g., neural-hybrid emulators under unrolled training.

Another notable benchmark for ordinary differential equations and deterministic chaos is Gilpin (2021). This benchmark shares our goal of relating the (temporal) performance of emulators with characteristic properties of the underlying dynamics, focusing on ODEs instead of PDEs.

Beyond works dedicated to benchmarks, several datasets from seminal papers gained popularity in the community. This includes data on the Burgers equation, Kolmogorov flow and Darcy problem used in the FNO paper (Li et al., 2021). Also, the Kolmogorov trajectories of Kochkov et al. (2021) are widely used. Other examples include the datasets of the DeepONet paper (Lu et al., 2021a), also used as part of the DeepXDE library (Lu et al., 2021b).

**Data Generators and Differentiable Physics** Physics-based deep learning often utilizes simple simulation suites with high-level interfaces like JAX-CFD (Kochkov et al., 2021), PhiFlow (Holl et al., 2020), JAX-MD (Schoenholz and Cubuk, 2020) or Warp (Macklin, 2022). Our reference solver is based on Fourier pseudo-spectral ETDRK methods for which there is currently no equally comprehensive package available in the Python deep learning ecosystem. For the problems that fit into the method's constraints, it is one of the most efficient approaches (Montanelli and Bootland, 2020). Writing numerical solvers in deep learning frameworks provides discrete differentiability, beneficially used in recent research (Um et al., 2020; Kochkov et al., 2021). Existing non-differentiable simulation software typically serves as reference generators for purely data-driven approaches. Popular examples are the Dedalus library in Python (Burns et al., 2020) and the Chebfun package in MATLAB (Driscoll et al., 2014). Fluid related work often employs OpenFoam (Weller et al., 1998).

# 4 Components of the APEBench benchmark

APEBench provides a wide array of PDE dynamics that, among others, allow studying different architectures, training methodologies, dataset sizes, and evaluation metrics. Below, we describe its design and capabilities, particularly the choice of numerical reference solver.

**Differentiable ETDRK Solver Suite**    We focus on semi-linear PDEs

$$\partial_t u = \mathcal{L}u + \mathcal{N}(u),$$

where the linear differential operator $\mathcal{L}$ contains a higher order derivative than the non-linear operator $\mathcal{N}(\cdot)$. This includes linear dynamics like advection, diffusion, and dispersion, and it also covers popular nonlinear dynamics like the viscid Burgers equation, the Korteweg-de Vries equation, the Kuramoto-Sivashinsky equation, as well as the incompressible Navier-Stokes equations for low to medium Reynolds numbers. Additionally, we consider reaction-diffusion equations like the Fisher-KPP equation, the Gray-Scott model, and the Swift-Hohenberg equation, demonstrating the applicability beyond fluid-like problems. The continuous form of all supported dynamics is given in Table 1 and a visual overview can be found in Figure 1.

For semi-linear PDEs, the class of Exponential Time Differencing Runge-Kutta (ETDRK) methods, first formalized by Cox and Matthews (2002), are one of the most efficient solvers known today (Montanelli and Bootland, 2020). Under periodic boundary conditions, a Fourier pseudo-spectral approach allows integrating the linear part $\mathcal{L}$ exactly via a (diagonalized) matrix exponential. As such, any linear PDE with constant coefficients can be solved *analytically*, without any temporal or spatial discretization error. This makes it possible to reliably and accurately evaluate the corresponding learning tasks. The non-linear part $\mathcal{N}(\cdot)$ is approximated by a Runge-Kutta method. We elaborate on the methods' motivation, implementation, and limitations in appendix B. For semi-linear problems with stiffness arising from the linear part, ETDRK methods show excellent stability and accuracy properties. Ultimately, the cost of one time step in $D$ dimensions is bounded by the Fast Fourier Transform (FFT) with $\mathcal{O}(N^D D \log(N))$. Due to pure explicit tensor operations, the method is well suited for GPUs, and discrete differentiability via automatic differentiation is straightforward.

**PDE Identifiers**    The ETDRK solver suite operates with a physical parametrization by specifying the number of spatial dimensions $D$, the domain extent $L$, the number of grid points $N$, the step size $\Delta t$ and constitutive parameters, like the velocity $c$ in case of the advection equation. All these parameters affect the difficulty of emulation and must, hence, be communicated when evaluating the forecasting of a specific PDE. As an exchange protocol or identifier of an experiment, APEBench also comes with a reduced interface tailored to identifying the present dynamics, including its discretization, in a minimal set of variables. This allows assigning an ID to each scenario in APEBench, which uniquely expresses the *discrete* dynamics to be emulated. For the $s$-th order linear derivative with coefficient $a_s$, we define two coefficients $\alpha_s$ and $\gamma_s$ as

$$\alpha_s = \frac{a_s \Delta t}{L^s} \qquad \text{and} \qquad \gamma_s = \alpha_s N^s 2^{s-1} D. \tag{1}$$

For a specific scenario, the $\alpha_s$ represent normalized *dynamics coefficients*, while the $\gamma_s$ quantify the *difficulty*. Gamma values correspond to the stability criteria of the most compact explicit finite difference stencils of the respective linear derivative (for $s = 1$, this is the CFL condition). Together, these values make it possible to quickly gain intuition about the dynamics encoded by a chosen PDE system and provide a convenient way to work with different scenarios: A list of gamma (or alpha) values together with $N$ and $D$ is sufficient to uniquely identify any linear dynamics. We have diffusion if only $\gamma_2 \geq 0$. For $\gamma_1 \neq 0 \wedge \gamma_2 \geq 0$, we obtain advection-diffusion, while only $\gamma_3 \neq 0$ yields dispersion.

The linear derivatives can be combined with a selection of nonlinear components to vary the system's dynamics, on whose reduction we elaborate in appendix B.7 and B.8. Combining a convection nonlinearity with $\gamma_2 \geq 0$ results in the viscous Burgers equation. If further combined with $\gamma_3 \neq 0$, the Korteweg-de Vries equation is obtained. Thus, the non-zero coefficients define the type of dynamics. Their relative and absolute scales define how challenging the emulator learning problem is. With this approach, APEBench intuitively communicates the emulated dynamics.

**Neural Emulator Architectures**    Our benchmark encompasses established neural architectures adaptable across spatial dimensions and compatible with Dirichlet, Neumann, and periodic boundary

conditions. This includes local convolutional architectures like ConvNets (Conv) and ResNets (Res) (He et al., 2016) as well as long-range architectures like UNets (Ronneberger et al., 2015) and Dilated ResNets (Dil) (Stachenfeld et al., 2021). Orthogonal to the two former classes, we consider pseudo-spectral architectures in the form of the Fourier Neural Operator (FNO), which have a global receptive field, but their performance instead depends on the spectrum of the underlying dynamics.

**Training Methodologies** Emulator training is the task of approximating a discrete numerical simulator $\mathcal{P}_h$. This solver advances a space-discrete state $u_h$ from one time step to the next. The goal is to replicate its behavior with the neural emulator $f_\theta$, i.e., to find weights $\theta$ such that $f_\theta \approx \mathcal{P}_h$. Since the neural emulator $f_\theta$ is trained on data from the numerical simulator $\mathcal{P}_h$, their interplay during learning is crucial. Many options exist, like one-step training (Tran et al., 2023), supervised unrolling (Um et al., 2020), or residuum-based unrolling (Geneva and Zabaras, 2020). We introduce a novel taxonomy based on unrolling steps during training and the reference branch length, unifying most approaches in the literature. Given a dataset of states $u_h \propto \mathcal{D}_h$, the objective is

$$L(\theta) = \mathbb{E}_{u_h \propto \mathcal{D}_h} \left[ \sum_{t=0}^{(T-B)} \sum_{b=1}^{B} l\left( f_\theta^{t+b}(u_h),\ \mathcal{P}_h^b(f_\theta^t(u_h)) \right) \right], \tag{2}$$

where $T$ is the number of unrolled steps at training time, and the per time step loss $l(\cdot, \cdot)$ typically is a mean squared error (MSE). During training, the emulator produces a trajectory $\{f_\theta^{t+b}\}$, which we call the *main chain*. The variable $B$ denotes the length of the *branch chain* $\{\mathcal{P}_h^b(\cdot)\}$, which defines how long the reference simulator is rolled out next to the main chain.

The popular one-step supervised learning problem is recovered by setting $T = B = 1$. Purely supervised unrolling is achieved with $T = B$. In such a case, all data can be pre-computed, allowing the reference simulator $\mathcal{P}_h$ to be turned off during training. We also consider the case of *diverted chain* learning with $T$ freely chosen and $B = 1$ providing a one-step difference from a reference simulator while maintaining an autoregressive rollout. This configuration necessitates the reference simulator $\mathcal{P}_h$ to be differentiable which is readily available within APEBench. While our results below do not modify the gradient flow (Brandstetter et al., 2022; List et al., 2022), the benchmark framework supports alterations of the backpropagation pass, as outlined in Appendix D.1.

**Neural-Hybrid Emulation** Next to the task of fully replacing the numerical simulator $\mathcal{P}_h$ with the neural network $f_\theta$, which we call *prediction*, APEBench is also designed for *correction*. For this, we introduce a coarse solver $\tilde{\mathcal{P}}_h$ that acts as a predictor together with a corrector network $\tilde{f}_h$ (Um et al., 2020; Kochkov et al., 2021). Together, they form a *neural-hybrid emulator*. For example, we support a *sequential* layout in which the $f_\theta$ of equation 2 is $f_\theta = \tilde{f}_\theta(\tilde{\mathcal{P}}_h)$. The coarse solver component $\tilde{\mathcal{P}}_h$ is also provided by the ETDRK solver suite. Any unrolling with $T \geq 2$ introduces a backpropagation-through-time that requires this coarse solver to be differentiable, which is also readily available.

**Metrics** Since this benchmark suite is concerned with *autoregressive neural emulator*, we emphasize the importance of *rollout metrics*. To compare two states $u_h^{[t]}$ and $u_h^{r,[t]}$ at time level $[t]$, we provide a range of established metric functions, which we elaborate in appendix F. This includes aggregations in state and Fourier space using different reductions and normalization. Moreover, we support metric computation for certain frequency ranges and the use of derivatives (i.e., Sobolev-based metrics) to highlight errors in the smaller scales/higher frequencies for which networks often produce blurry predictions (Rahaman et al., 2019).

## 5 Experiments

We present experiments highlighting the types of studies enabled by APEBench, focusing on temporal stability and generalization of trained emulators. We measure performance in terms of the normalized RMSE (see equation (31)) to allow comparisons over time if magnitudes decay and across dynamics. Aggregation over time is done with a geometric mean (see equation (32)). Plots show the median performance across network initializations, with error bars for the 50% inter-quantile range (IQR). Further specifics are provided in appendix H.

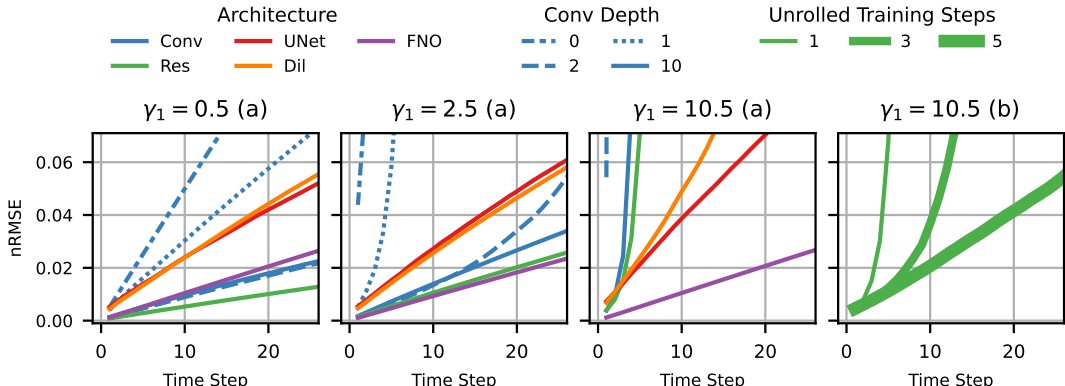

Figure 3: (a) Performance of various neural emulator architectures on a 1D advection problem with increasing difficulty ($\gamma_1$ = CFL). If the demands on the receptive field (given by $\gamma_1$) are not fulfilled, emulators diverge immediately. (b) Unrolling improves accuracy at the highest difficulty.

## 5.1 Bridging Numerical Simulators and Neural Emulators

The motivational example in section 2 demonstrated the emulation of 1D advection for CFL $\gamma_1 < 1$. We saw that purely linear convolutions with two learnable parameters are capable emulators. This section expands the experiment by training various nonlinear architectures on this linear PDE for varying difficulties. Since the CFL condition ($|\gamma_1| < 1$) restricts information flow across one cell for first-order methods, we hypothesize that emulators with larger receptive fields can handle difficulties beyond one. Indeed, when considering standard feedforward convolutional architectures of varying depths, there is a clear connection between the effective receptive field and the highest possible difficulty. With a kernel size of three, each additional stacked convolution adds a receptive field of one per direction. A depth of zero represents a linear convolution. As shown in Figure 3 (a), such an approach is only feasible for $\gamma_1 \leq 1$, aligning with the CFL stability criterion of the first-order upwind method and the results from section 2. Beyond this, the network fails to emulate the dynamics and diverges almost immediately. A similar behavior is observed for a convolution depth of one with an effective receptive field of two per direction. This is insufficient for advection dynamics of difficulty $\gamma_1 = 2.5$.

Long-range convolutional architectures, like UNets and Dilated ResNets, perform better across the difficulties, i.e., the error rollout does not steepen as strongly as with the local convolutional architectures. However, they never turn out to be the best architectures. Given their inductive biases for long-range dependencies, they spend parameters on interactions with degrees of freedom beyond the necessary receptive field. This likely explains their reduced ability to produce the best results in this relatively simple scenario of hyperbolic linear advection with an influence range known a priori.

The pseudo-spectral FNO has a performance which is agnostic to changes in $\gamma_1$. This behavior can be explained by its inherent capabilities to learn band-limited linear dynamics and its similarity with the data-generating solver. Despite these advantages, local convolutional architectures like a ResNet are on par with the FNO under low difficulties that do not demand a large receptive field.

Surprisingly, ResNet and the deepest ConvNet fail at the highest difficulty despite having sufficient receptive field. However, Figure 3 (b) reveals that under additional unrolling during training, the same ResNet greatly improves in temporal generalization. In line with the motivational example of section 2, this suggests that unrolling during training, rather than exposure to more physics, is key to a better learning signal and achieving desirable numerical properties.

## 5.2 Diverted Chain: A Learning Methodology with A Differentiable Fine Solver

APEBench's tight integration with its ETDRK solver suite enables the exploration of promising training setups beyond purely data-driven supervised unrolling. One such setup is the *diverted chain* approach as obtained with Eq. 2, combining autoregressive unrolling with the familiar one-step difference. Here, the reference solver branches off after each autoregressive network prediction. providing a continuous source of ground truth data during training. As it hinges on fully integrating the (differentiable) reference solver, this variant has not been studied in previous work.

Figure 4 compares the performance of a ResNet emulator trained using one-step supervised training, 5-step supervised unrolling, and 5-step diverted chain training on three nonlinear 1D dynamics: viscous Burgers, Kuramoto-Sivashinsky (KS), and hyper-viscous Korteweg-de Vries (KdV). The results demonstrate that training with unrolling, regardless of the specific approach, generally improves long-term accuracy, as indicated by the lower 100-step error. This improvement is particularly pronounced for the KdV equation, likely due to the increased effective receptive field seen during training, which is especially beneficial for strongly hyperbolic problems. Notably, the diverted-chain approach further enhances long-term accuracy for the KdV scenario. While it does not surpass supervised unrolling for Burgers and KS in terms of long-term accuracy, it excels in short-term performance, only slightly underperforming compared to the one-step trained emulator. These findings confirm that the diverted-chain approach effectively combines the benefits of training time rollout and one-step differences, demonstrating the flexibility of APEBench to explore diverse training strategies due to its differentiable solver.

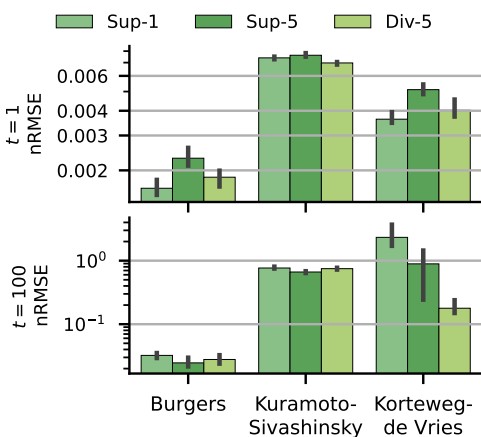

Figure 4: Comparison of training methodologies for a ResNet emulator on three nonlinear 1D dynamics. Emulators benefit from rollout training, the strongest visible for the KdV case. Diverted-Chain offers the advantage of long-term accuracy without sacrificing short-term performance.

## 5.3 Neural-Hybrid Emulators and Sequential Correction

Neural-hybrid emulators, which combine neural networks with traditional numerical solvers, are a promising area of research in physics-based deep learning (Kochkov et al., 2021; Um et al., 2020). APEBench's differentiable ETDRK solver framework facilitates the exploration of such hybrid approaches. In this section, we investigate the sequential correction of a defective solver $\tilde{\mathcal{P}}_h$ using both a ResNet and an FNO for a 2D advection problem with a difficulty of $\gamma_1 = 10.5$. Three variations of this task are explored: full prediction, and sequential correction with the coarse solver handling 10% ($\tilde{\gamma}_1 = 1.05$) or 50% ($\tilde{\gamma}_1 = 5.25$) of the difficulty.

Figure 5 displays the geometric mean of the test rollout error over 100 time steps. The results reveal that supervised unrolling consistently improves the performance of the ResNet and ResNet-hybrid models, outperforming the FNO in every case. Notably, the FNO's performance remains almost unaffected by unrolling and changes in difficulty, likely due to its global receptive field and ability to capture long-range dependencies. In contrast, the ResNet, with its limited receptive field, benefits significantly from unrolling. The ResNet performs best in the 50% correction task, highlighting the potential of neural-hybrid approaches that leverage the strengths of both convolutional and pseudo-spectral methods. These

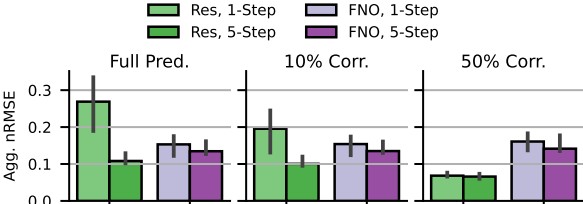

Figure 5: ResNet and FNO either as full prediction emulators or neural-hybrid emulators for 2D advection ($\gamma_1 = 10.5$) with a coarse solver doing 10% or 50% of the difficulty. The geometric mean of the rollout error over 100 time steps is shown. Training with unrolling benefits the ResNet yet only shows marginal improvement for the FNO. The ResNet can work in symbiosis with a coarse simulator.

findings underscore the importance of tailoring the training strategy and architecture to the specific task and difficulty level and emphasize the potential of hybrid approaches for superior performance in PDE emulation.

## 5.4 Relating Architectures and PDE Dynamics

APEBench's diverse collection of semi-linear PDEs provides a robust testing ground for emulator architectures across various dynamics. Below, we compare the five main neural architectures regarding

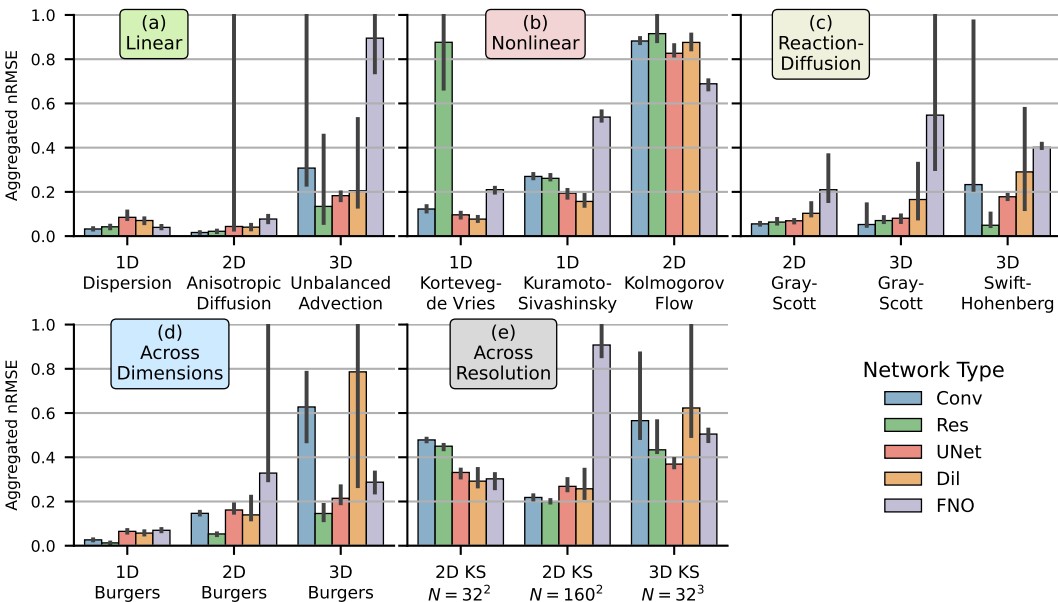

Figure 6: Comparison of emulator architectures across various PDE dynamics in 1D, 2D, and 3D. The ResNet consistently performs well across all dynamics and dimensions. Local architectures struggle with higher-order derivatives, while limited active modes hinder the FNO's performance in some cases. The Dilated ResNet is the better long-range architecture in 1D, whereas the UNet is better suited for higher dimensions.

the geometric mean of the error over 100 rollout steps, summarized in Figure 6. The parameter counts across architectures are almost identical within the same spatial dimensions. We provide ablations on the choices of optimization configuration (section I.2), training dataset size (section I.3), and network parameter size (section I.4) in the appendix.

**Linear PDEs**  For the 1D hyperbolic dispersion problem, local convolutional architectures (with sufficient receptive fields and low problem difficulty) again excel, closely followed by the pseudo-spectral FNO. The considered higher-dimensional linear PDEs introduce *spatial mixing*, making the learning task more challenging. For 2D anisotropic diffusion, both local and global convolutional architectures perform well, with the FNO lagging slightly and the UNet showing a significant spread in performance. In the 3D unbalanced advection case, the FNO's limited active modes hinder capture of the solution's full complexity. Convolutional architectures struggle to balance short- and long-range interactions across different dimensions, with UNets showing the most consistent performance.

**Nonlinear PDEs**  For the 1D Korteweg-de Vries and Kuramoto-Sivashinsky equations, local architectures likely struggle with the hyper-diffusion term due to insufficient receptive fields, giving long-range architectures an advantage. The FNO's performance is also suboptimal, potentially due to limited active modes. Notably, the FNO excels in the challenging Navier-Stokes Kolmogorov Flow case.

**Reaction-Diffusion**  Reaction-diffusion problems, characterized by polynomial nonlinearities with no spatial dependence that develop rich (high-frequency) patterns (see Figure 14), are best handled by local convolutional architectures, particularly ResNets. The FNO was the least suitable architecture for this class of problems. We attribute this to its low-frequency bias in that it learns predictions in the frequencies beyond its active modes only indirectly via the energy transfer of the nonlinear activation.

**Performance across Dimensions**  Emulating the Burgers equation across dimensions reveals an exponential decrease in performance as dimensionality increases (Figure 6 (d)). Across all dimensions, ResNets consistently emerge as the top-performing architecture, showcasing their adaptability to varying spatial complexities. Dilated ResNets, while effective in 1D and 2D, experience a significant performance drop in 3D. This likely stems from their dilation-based mechanism for long-range interactions resulting in less uniform coverage of the receptive field compared to UNets. The performance gap between standard ConvNets and ResNets widens in 3D, highlighting the increasing importance of skip connections.

**Performance across Resolution** In this example, we emulate a KS equation in 2D using $N = 32^2$ and $N = 160^2$ as well as in 3D with $N = 32^3$. Due to the difficulty mode, the dynamics are adapted based on resolution and dimensionality. Counterintuitively, emulation often improves with increasing resolution until the emulators' architectures are fully utilized. In contrast, the FNO struggles in this scenario because, due to the difficulty-based rescaling, the spectrum is fully populated in both resolutions for the 2D case (see the spectra in Figure 14). Across all architectures, the jump to 3D worsens their performance, which reinforces the observations shown in Figure 6 (d). Notably, the UNet emerges as the best architecture in 3D likely because it has a global receptive field at this resolution.

## 6 Limitations

We currently focus on periodic boundary conditions and uniform Cartesian grids. Broadening the scope of the benchmark w.r.t. other numerical solvers and discretizations, forced dynamics, and specialized network architectures constitute highly interesting future work.

## 7 Conclusions and Outlook

We presented APEBench, a benchmark suite for autoregressive neural emulators of time-dependent PDEs, focusing on training methodologies, temporal generalization, and differentiable physics. The benchmark's efficient JAX-based pseudo-spectral solver framework enables rapid experimentation across 1D, 2D, and 3D dynamics. We introduced the concept of *difficulties* to uniquely identify dynamics and scale experiments. The unified treatment of unrolling methodologies was demonstrated as a foundation to investigate learned emulators across a wide range of dynamics and training strategies. We revealed connections between the performance of an architecture, problem type, and difficulty that make it possible to understand their behavior with analogies to classical numerical simulators. Specifically, our benchmark experiments highlight the importance of:

- Matching the network architecture to the specific problem characteristics. Local problems benefit significantly from local convolutions, while global receptive fields are less impacted by unrolled training.
- Utilizing training with unrolling to significantly improve performance, particularly for challenging problems and under limited receptive fields.
- Exploring hybrid approaches that combine neural networks with coarse numerical solvers (correction) and differentiable reference solvers (diverted-chain training) to further enhance the capabilities of learned emulations.

In this context, many interesting areas for future work remain. Particularly notable are parameter-conditioned emulators and foundation models that can solve larger classes of PDE dynamics. Perhaps the most crucial avenue for future research with APEBench lies in conducting an even deeper investigation of unrolling and the intricate interplay between emulator and simulator. A deeper understanding here could significantly impact the field of neural PDE emulation.

## 8 Acknowledgements

Felix Koehler acknowledges funding from the Munich Center for Machine Learning (MCML). The authors are grateful for constructive discussions with Bjoern List, Patrick Schnell, Georg Kohl, Qiang Liu, and Dion Haeffner.

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

# Appendix

## A  How to use APEBench

The APEBench benchmark suite is hosted as open source under an MIT license at `https://github.com/tum-pbs/apebench` and can be installed as a Python package via `pip install apebench`. Publically hosted documentation is available at `https://tum-pbs.github.io/apebench/`. A project page is available under `https://github.com/tum-pbs/apebench-paper`.

Next to the benchmark suite, we also release parts of it as individual Python packages:

- **Exponax** is a standalone Fourier pseudo-spectral exponential time-differencing Runge-Kutta solver in JAX with a rich feature set, including the ability to be differentiated. It is hosted open source under an MIT license at `https://github.com/Ceyron/exponax` and can be installed as a Python package via `pip install exponax`. Publically hosted documentation is available at `https://fkoehler.site/exponax/`.

- **PDEquinox** is a collection of neural emulator architectures built on top of Equinox (Kidger and Garcia, 2021). It is hosted open source under an MIT license at `https://github.com/Ceyron/pdequinox` and can be installed as a Python package via `pip install pdequinox`. Publically hosted documentation is available at `https://fkoehler.site/pdequinox/`.

- **Trainax** is a collection of abstract implementations for various unrolled training strategies, including the presented diverted-chain methodology. It is hosted open source under an MIT license at `https://github.com/Ceyron/trainax` and can be installed as a Python package via `pip install trainax`. Publically hosted documentation is available at `https://fkoehler.site/trainax/`.

- **Vape4d** is a performant spatiotemporal volume render that can be used to quickly assess the results of neural emulation in higher dimensions. It is hosted open source under a BSD 2-Clause license at `https://github.com/KeKsBoTer/vape4d` and can be installed as a Python package via `pip install vape4d`. A web-based version running locally in the browser working with multi-axis numpy arrays is accessible at `https://vape.niedermayr.dev`.

Beyond the software included in this publication, we also release data and experimental results.

- A curated version of representative data trajectories, which was scraped from the APEBench defaults, is hosted here: `https://huggingface.co/datasets/thuerey-group/apebench-scraped`.

- All experimental scripts, raw data, processed data, and visualization scripts for the main part can be found here: `https://huggingface.co/thuerey-group/apebench-paper`.

- The same for the ablation studies is hosted here: `https://huggingface.co/thuerey-group/apebench-paper-ablations`.

## B  More Details on Fourier Pseudo-Spectral ETDRK Methods

### B.1  Motivation and Background

Exponential Time Differencing Runge Kutta (ETDRK) methods offer a powerful approach to solving time-dependent partial differential equations (PDEs) by leveraging the exact solution of linear ordinary differential equations (ODEs) through matrix exponentials. This approach is particularly advantageous when dealing with stiff systems, where traditional numerical methods may require impractically small time steps to maintain stability.

**Linear ODEs**   The core idea of exponential time differencing methods is that *linear* ordinary differential equations can be solved exactly by the matrix exponential. For this, first consider a scalar-valued linear ODE for the solution function $u(t) : \mathbb{R}_+ \to \mathbb{R}$ of the form

$$\frac{\mathrm{d}u}{\mathrm{d}t} = \lambda u(t),$$
$$u(0) = u_0.$$

The analytical solution can be found by separation of variables to be

$$u(t) = u_0 e^{\lambda t}.$$

We can frame it as a timestepper with a fixed time step size $\Delta t$ as

$$u^{[t+1]} = e^{\lambda \Delta t} u^{[t]}.$$

A timestepper of this form is unconditionally stable, given the underlying dynamics are stable (here requiring $\mathrm{Re}(\lambda) \leq 0$).

**System of Linear ODEs**   For a linear ODE system, we seek a solution function of the form $u(t) : \mathbb{R}_+ \to \mathbb{R}^C$ with $C$ components (= channels or degrees of freedom). The system is of the form

$$\frac{\mathrm{d}u}{\mathrm{d}t} = Au(t),$$
$$u(0) = u_0,$$

where $A \in \mathbb{R}^{C \times C}$ is a constant matrix. The analytical solution can be found by the matrix exponential to be

$$u(t) = e^{At} u_0.$$

Framing this as a timestepper with a fixed time step size $\Delta t$, we get

$$u^{[t+1]} = e^{A\Delta t} u^{[t]}.$$

The matrix exponential of $A\Delta t$ can be precomputed. Hence, advancing the state in time is a matrix-vector multiplication. As long as the underlying dynamics are stable (requiring the real part of all eigenvalues of $A$ to be non-positive), the timestepper is unconditionally stable. Such a strategy is viable for small systems with a few degrees of freedom $C$.

**Discretizing Linear PDEs by Method of Lines**   Now consider linear partial differential equations (PDEs) of the form

$$\frac{\partial u}{\partial t} = \mathcal{L}u(t, x)$$
$$u(0, x) = u_0,$$
$$\mathcal{B}u(t, x) = 0,$$

where $u(t, x) : \mathbb{R}_+ \times \Omega \to \mathbb{R}^C$ is the solution function, $\mathcal{L}$ is a linear differential operator, and $\mathcal{B}$ is a boundary operator. The domain is $\Omega \subseteq \mathbb{R}^D$. As a motivating example, consider the one-dimensional heat equation on periodic boundary conditions

$$\frac{\partial u}{\partial t} = \nu \frac{\partial^2 u}{\partial x^2},$$
$$u(0, x) = u_0,$$
$$u(t, 0) = u(t, L),$$

where $\nu$ is the diffusion coefficient. This equation only has a single channel ($C = 1$) that can be interpreted as a temperature. Following a method-of-lines approach, let us discretize the spatial domain $\Omega = (0, L)$ into $N$ intervals of size $\Delta x = L/N$. We will consider the left end of each interval as a degree of freedom. As such, the left boundary of the domain is part of the grid and the right boundary is excluded. This naturally encodes the periodic boundary conditions. We can approximate

the second spatial derivative by a centered finite difference scheme that can be represented by the matrix

$$
\mathcal{L}_h = \nu \frac{1}{\Delta x^2}
\begin{bmatrix}
-2 & 1 & 0 & \cdots & 0 & 1 \\
1 & -2 & 1 & \cdots & 0 & 0 \\
0 & 1 & -2 & \cdots & 0 & 0 \\
\vdots & \vdots & \vdots & \ddots & \vdots & \vdots \\
0 & 0 & 0 & \cdots & -2 & 1 \\
1 & 0 & 0 & \cdots & 1 & -2
\end{bmatrix}.
$$

Notice that this matrix is almost tridiagonal. It also has entries in the top right and bottom left to account for the periodic boundary conditions. It is sparse because the number of nonzero entries are $\mathcal{O}(N)$, not $\mathcal{O}(N^2)$. After spatial discretization, the PDE becomes a system of ODEs

$$
\frac{\mathrm{d}u_h}{\mathrm{d}t} = \mathcal{L}_h u_h(t),
$$

$$
u_h(0) = u_{h,0},
$$

where $u_h(t) \in \mathbb{R}^N$ is the state vector at time $t$ and $u_{h,0} \in \mathbb{R}^N$ is the initial condition. Similar to the ordinary differential equation, the solution can be found by the matrix exponential to be

$$
u_h(t) = e^{\mathcal{L}_h t} u_{h,0},
$$

or in a timestepper form

$$
u_h^{[t+1]} = e^{\mathcal{L}_h \Delta t} u_h^{[t]}.
$$

However, implementing a time-stepper by the matrix exponential is not feasible for large $N$ because the matrix exponential is $\mathcal{O}(N^3)$ in cost and the result of the matrix exponential on sparse matrices is dense.

**Spectral Derivatives**  If we have a function $u(x)$ on a periodic domain $\Omega = (0, L)$, we can compute its derivative using the Fourier transform $\mathcal{F}$ via

$$
\frac{\partial u}{\partial x} = \mathcal{F}^{-1}\left(ik\mathcal{F}(u)\right).
$$

Here, $i$ is the imaginary unit, and $k$ is the wavenumber. In a discrete setting, if we have $u(x)$ sampled at $N$ points (on a grid similarly as before) we can denote the state vector as $u_h \in \mathbb{R}^N$. We can use the discrete analogon of the Fourier transform, the discrete Fourier transform (DFT) $\mathcal{F}_h$, to compute the derivative giving

$$
\left(\frac{\partial u}{\partial x}\right)_h = \mathcal{F}_h^{-1}\left(ik_h \odot \mathcal{F}_h(u_h)\right).
$$

Such an approximation converges quickly (exponentially fast) for smooth functions (Trefethen, 2000; Boyd, 2001). However, more importantly, is that the **derivative operator diagonalizes in Fourier space**. In the heat equation example, the discrete second-order derivative $\mathcal{L}_h$ was a sparse but non-diagonal matrix. In Fourier space, any derivative operator is simply an element-wise multiplication with the general Fourier derivative operator (elementwise) raised to the order of the derivative, $(ik)^s$. Hence, the spectral version of a $s$-th order derivative is

$$
\left(\frac{\partial^s u}{\partial x^s}\right)_h = \mathcal{F}_h^{-1}\left((ik_h)^s \odot \mathcal{F}_h(u_h)\right).
$$

**Matrix Exponentials in Fourier Space**  If we transformed the space-discrete heat equation into Fourier space before solving, we get

$$
\frac{\partial \hat{u}_h}{\partial t} = \nu(ik_h)^2 \odot \hat{u}_h,
$$

where $\hat{u}_h = \mathcal{F}_h(u_h) \in \mathbb{C}^N$ is the Fourier transform of the state vector. An element-wise multiplication can be represented by a diagonal matrix

$$
\mathrm{diag}(\nu(ik_h)^2) =
\begin{bmatrix}
\nu(ik_1)^2 & 0 & \cdots & 0 \\
0 & \nu(ik_2)^2 & \cdots & 0 \\
\vdots & \vdots & \ddots & \vdots \\
0 & 0 & \cdots & \nu(ik_N)^2
\end{bmatrix}.
$$

Matrix exponentials of diagonal matrices are straightforward to compute by element-wise exponentiation of the diagonal elements. To distinguish the elementwise exponential and the matrix exponential, let us use $\exp(\cdot)$ and $e^{\cdot}$, respectively. Hence, the solution to the state vector in Fourier space is

$$\hat{u}_h(t) = \exp(\hat{\mathcal{L}}_h t) \odot \hat{u}_{h,0},$$

or in a timestepper form

$$\hat{u}_h^{[t+1]} = \exp(\hat{\mathcal{L}}_h \Delta t) \odot \hat{u}_h^{[t]}.$$

If we wanted to solve the heat equation in state space, we could use a forward and inverse Fourier transform

$$u_h^{[t+1]} = \mathcal{F}_h^{-1}\left(\exp(\hat{\mathcal{L}}_h \Delta t) \odot \mathcal{F}_h(u_h^{[t]})\right).$$

Since the spectral differentiation is exact if the function is bandlimited, this method of solving the heat equation is exact. More generally speaking, if we select a bandlimited initial condition $u_{h,0}$ (and have periodic boundary conditions), we can integrate trajectories of **any linear PDE** (with constant coefficients) without discretization errors in space and time, and with arbitrarily large time steps. This requires the underlying dynamics to be stable. In the case of the heat equation, we need $\nu \geq 0$.

**Pseudo-Spectral Methods for Nonlinear Terms** Now consider nonlinear PDEs of the form

$$\frac{\partial u}{\partial t} = \mathcal{L}u(t,x) + \mathcal{N}(u(t,x))$$
$$u(0,x) = u_0,$$
$$\mathcal{B}u(t,x) = 0,$$

where $\mathcal{N}$ is a nonlinear differential operator. An example could be the one-dimensional Burgers equation on periodic boundary conditions

$$\frac{\partial u}{\partial t} = \nu \frac{\partial^2 u}{\partial x^2} - \frac{1}{2}\frac{\partial u^2}{\partial x},$$
$$u(0,x) = u_0,$$
$$u(t,0) = u(t,L).$$

An exponential time differencing approach breaks down because the nonlinear differential operator does **not** diagonalize in Fourier space. If we transform the space-discrete equation into Fourier space, we get

$$\frac{\partial \hat{u}_h}{\partial t} = \hat{\mathcal{L}}_h \odot \hat{u}_h + \hat{\mathcal{N}}_h(\hat{u}_h). \tag{3}$$

We can design a pseudo-spectral evaluation strategy for the nonlinear term by evaluating the square in state space using an inverse and forward Fourier transform. For the Burgers equation, the nonlinear term is

$$\hat{\mathcal{N}}_h(\hat{u}_h) = -\frac{1}{2}(ik_h) \odot \mathcal{F}_h\left(\left(\mathcal{F}_h^{-1}(\hat{u}_h)\right)^2\right).$$

This nonlinear operator is not diagonal. Hence, we cannot easily compute the matrix exponential. Cox and Matthews (2002) multiplied equation (3) using the integrating factor $\exp(-\hat{\mathcal{L}}_h t)$ and integrated over a time step $\Delta t$ to get

$$u_h^{[t+1]} = \exp(\hat{\mathcal{L}}_h \Delta t) \odot u_h^{[t]} + \exp(\hat{\mathcal{L}}_h \Delta t) \odot \int_0^{\Delta t} \exp(\tau \hat{\mathcal{L}}_h) \odot \hat{\mathcal{N}}_h(u_h(\tau)) \, d\tau. \tag{4}$$

Note the usage of the elementwise exponentials on the linear operator that diagonalizes. From this format, they devised Runge-Kutta approximations to the integral. We will use a naming scheme that slightly deviates from theirs.

**ETDRK0** This method does not evaluate the integral at all. As such, only the linear part of the PDE is integrated.

$$\hat{u}_h^{[t+1]} = \exp(\hat{\mathcal{L}}_h \Delta t) \odot \hat{u}_h^{[t]}. \tag{5}$$

**ETDRK1**   This method is similar to an Euler step for the nonlinear part.

$$\hat{u}_h^{[t+1]} = \exp(\hat{\mathcal{L}}_h \Delta t) \odot \hat{u}_h^{[t]} + \frac{\exp(\hat{\mathcal{L}}_h \Delta t) - 1}{\hat{\mathcal{L}}_h} \odot \hat{\mathcal{N}}_h(\hat{u}_h^{[t]}). \tag{6}$$

All fractions hold elementwise.

**ETDRK2**   This method uses two stages to approximate the integral.

$$\hat{u}_h^* = \exp(\hat{\mathcal{L}}_h \Delta t) \odot \hat{u}_h^{[t]} + \frac{\exp(\hat{\mathcal{L}}_h \Delta t) - 1}{\hat{\mathcal{L}}_h} \odot \hat{\mathcal{N}}_h(\hat{u}_h^{[t]}). \tag{7}$$

$$\hat{u}_h^{[t+1]} = \hat{u}_h^* + \frac{\exp(\hat{\mathcal{L}}_h \Delta t) - 1 - \hat{\mathcal{L}}_h \Delta t}{\hat{\mathcal{L}}_h^2 \Delta t} \left( \hat{\mathcal{N}}_h(\hat{u}_h^*) - \hat{\mathcal{N}}_h(\hat{u}_h^{[t]}) \right). \tag{8}$$

**ETDRK3**   This method uses three stages to approximate the integral.

$$\hat{u}_h^* = \exp(\hat{\mathcal{L}}_h \Delta t/2) \odot \hat{u}_h^{[t]} + \frac{\exp(\hat{\mathcal{L}}_h \Delta t/2) - 1}{\hat{\mathcal{L}}_h} \odot \hat{\mathcal{N}}_h(\hat{u}_h^{[t]}). \tag{9}$$

$$\hat{u}_h^{**} = \exp(\hat{\mathcal{L}}_h \Delta t/2) \odot \hat{u}_h^{[t]} + \frac{\exp(\hat{\mathcal{L}}_h \Delta t) - 1}{\hat{\mathcal{L}}_h} \odot \left( 2\hat{\mathcal{N}}_h(\hat{u}_h^*) - \hat{\mathcal{N}}_h(\hat{u}_h^{[t]}) \right). \tag{10}$$

$$\hat{u}_h^{[t+1]} = \exp(\hat{\mathcal{L}}_h \Delta t) \odot \hat{u}_h^{[t]} \tag{11}$$

$$+ \frac{-4 - \exp(\hat{\mathcal{L}}_h \Delta t) + \exp(\hat{\mathcal{L}}_h \Delta) \left( 4 - 3\hat{\mathcal{L}}_h \Delta t + \left( \hat{\mathcal{L}}_h \Delta t \right)^2 \right)}{\hat{\mathcal{L}}_h^3 (\Delta t)^2} \odot \hat{\mathcal{N}}_h(\hat{u}_h^{[t]}). \tag{12}$$

$$+ 4 \frac{2 + \hat{\mathcal{L}}_h \Delta t + \exp(\hat{\mathcal{L}}_h \Delta t) \left( -2 + \hat{\mathcal{L}}_h \Delta t \right)}{\hat{\mathcal{L}}_h^3 (\Delta t)^2} \odot \hat{\mathcal{N}}_h(\hat{u}_h^*) \tag{13}$$

$$+ \frac{-4 - 3\hat{\mathcal{L}}_h \Delta t - \left( \hat{\mathcal{L}}_h \Delta t \right)^2 + \exp(\hat{\mathcal{L}}_h \Delta t) \left( 4 - \hat{\mathcal{L}}_h \Delta t \right)}{\hat{\mathcal{L}}_h^3 (\Delta t)^2} \odot \hat{\mathcal{N}}_h(\hat{u}_h^{**}). \tag{14}$$

**ETDRK4**   This method uses four stages to approximate the integral.

$$\hat{u}_h^* = \exp(\hat{\mathcal{L}}_h \Delta t/2) \odot \hat{u}_h^{[t]} + \frac{\exp(\hat{\mathcal{L}}_h \Delta t/2) - 1}{\hat{\mathcal{L}}_h} \odot \hat{\mathcal{N}}_h(\hat{u}_h^{[t]}). \tag{15}$$

$$\hat{u}_h^{**} = \exp(\hat{\mathcal{L}}_h \Delta t/2) \odot \hat{u}_h^{[t]} + \frac{\exp(\hat{\mathcal{L}}_h \Delta t/2) - 1}{\hat{\mathcal{L}}_h} \odot \hat{\mathcal{N}}_h(\hat{u}_h^*). \tag{16}$$

$$\hat{u}_h^{***} = \exp(\hat{\mathcal{L}}_h \Delta t) \odot \hat{u}_h^* + \frac{\exp(\hat{\mathcal{L}}_h \Delta t/2) - 1}{\hat{\mathcal{L}}_h} \odot \left( 2\hat{\mathcal{N}}_h(\hat{u}_h^{**}) - \hat{\mathcal{N}}_h(\hat{u}_h^{[t]}) \right). \tag{17}$$

$$\hat{u}_h^{[t+1]} = \exp(\hat{\mathcal{L}}_h \Delta t) \odot \hat{u}_h^{[t]} \tag{18}$$

$$+ \frac{-4 - \hat{\mathcal{L}}_h \Delta t + \exp(\hat{\mathcal{L}}_h \Delta t) \left( 4 - 3\hat{\mathcal{L}}_h \Delta t + \left( \hat{\mathcal{L}}_h \Delta t \right)^2 \right)}{\hat{\mathcal{L}}_h^3 (\Delta t)^2} \odot \hat{\mathcal{N}}_h(\hat{u}_h^{[t]}) \tag{19}$$

$$+ 2 \frac{2 + \hat{\mathcal{L}}_h \Delta t + \exp(\hat{\mathcal{L}}_h \Delta t) \left( -2 + \hat{\mathcal{L}}_h \Delta t \right)}{\hat{\mathcal{L}}_h^3 (\Delta t)^2} \odot \left( \hat{\mathcal{N}}_h(\hat{u}_h^*) + \hat{\mathcal{N}}_h(\hat{u}_h^{**}) \right) \tag{20}$$

$$+ \frac{-4 - 3\hat{\mathcal{L}}_h \Delta t - \left( \hat{\mathcal{L}}_h \Delta t \right)^2 + \exp(\hat{\mathcal{L}}_h \Delta t) \left( 4 - \hat{\mathcal{L}}_h \Delta t \right)}{\hat{\mathcal{L}}_h^3 (\Delta t)^2} \odot \hat{\mathcal{N}}_h(\hat{u}_h^{***}). \tag{21}$$

**Numerically Stable Coefficient Computation**    Setting up a timestepper using the EDTRK methods requires the precomputation of coefficients based on the discrete linear operator in Fourier space $\hat{\mathcal{L}}_h$ and the time step size $\Delta t$. These coefficients are of the form

$$g(z) = \frac{\exp(z) - 1}{z},$$

which encounter numerical instabilities for small $z$. To avoid these problems, Kassam and Trefethen (2005) designed a contour integral method in the complex plain to compute the coefficients more stably and accurately, which we chose to implement.

**Dealiasing for the Nonlinear Part**    Like any pseudo-spectral method, evaluating the nonlinear term moves energy between the modes. This can move energy into wavenumbers that cannot be represented by the grid. Hence, they appear as aliases, potentially corrupting the solution and leading to instabilities. A common strategy is to set all Fourier coefficients above a certain threshold to zero *before* evaluating the nonlinearity. Orszag (1971) proposed that for quadratic nonlinearities (like in the Burgers equation), keeping the first $2/3$ modes and setting the rest to zero is sufficient to avoid issues caused by aliasing. This does not fully eliminate aliasing (which would require keeping only half of the modes) but only produces aliases for the modes, which will be zeroed in the next step. Let $\mathcal{M}$ be a zero mask at the wavenumber position we want to remove and one otherwise. The correct evaluation of the Burgers nonlinearity then becomes

$$\hat{\mathcal{N}}_h(\hat{u}_h) = \frac{1}{2}(ik_h) \odot \mathcal{F}_h \left( \left( \mathcal{F}_h^{-1}(\mathcal{M} \odot \hat{u}_h) \right)^2 \right).$$

**Higher Dimensions**    Fourier pseudo-spectral Exponential Time Differencing methods work in arbitrary spatial dimensions as long each dimension uses periodic boundaries, and employs a uniform Cartesian grid to be compatible with the Fast Fourier Transform. For instance, consider the two-dimensional heat equation

$$\frac{\partial u}{\partial t} = \nu \left( \frac{\partial^2 u}{\partial x^2} + \frac{\partial^2 u}{\partial y^2} \right),$$
$$u(0, x, y) = u_0,$$
$$u(t, 0, y) = u(t, L, y),$$
$$u(t, x, 0) = u(t, x, L).$$

We assume that the domain is the scaled unit-cube $\Omega = (0, L)^2$, which is discretized with the same number of points in each dimension. Again, the left boundary is part of the discretization, and the right boundary is excluded. Let $k_{h,0}$ denote the discrete wavenumber grid in $x$-direction and $k_{h,1}$ in $y$-direction. Hence, the linear operator in Fourier space can be written as

$$\hat{\mathcal{L}}_h = \nu \left( (ik_{h,0})^2 + (ik_{h,1})^2 \right) \in \mathbb{C}^{N \times N}.$$

This operator is of shape $N \times N$ where $N$ is the number of points in each dimension. It is still diagonal because the state in Fourier space has the same shape. Hence, the timestepper in Fourier space is again

$$\hat{u}_h^{[t+1]} = \exp(\hat{\mathcal{L}}_h \Delta t) \odot \hat{u}_h^{[t]}.$$

## B.2   ETDRK Methods in other Software Libraries

The popular ChebFun package in *MATLAB* (Driscoll et al., 2014) implements pseudo-spectral ETDRK methods with a range of spectral bases under their `spinX.m` module. It served as a reliable data generator for early works in physics-based deep learning. For instance, it was used to produce the training and test data for Raissi and Karniadakis (2018) and parts of the experiments of Li et al. (2021). Due to the two-language nature, with most deep learning research happening in Python, dynamically calling MATLAB solvers is hard to impossible. Naturally, this also excludes the option to differentiate over them to allow differentiable physics, for instance, to enable diverted-chain learning as discussed in section 5.2 or correction setups as discussed in section 5.3. We view our ETDRK solver framework as a spiritual successor of this `spinX.m` module. JAX, as the computational backend, elevates the power of this solver type with automatic vectorization, backend-agnostic execution, and tight integration for deep learning via the versatile automatic differentiation engine.

Beyond ChebFun, popular implementations of pseudo-spectral implementations can be found in the Dedalus package (Burns et al., 2020) in the Python world and the FourierFlows.jl (Constantinou et al., 2023) package in the Julia ecosystem.

### B.3 Limitations of Fourier Pseudo-Spectral ETDRK Methods

Fourier pseudo-spectral ETDRK methods are a powerful class of numerical techniques for solving semi-linear partial differential equations (PDEs), where the highest-order derivative is linear. These methods excel in scenarios where the stiffness of the linear part poses the primary challenge in integration. By analytically treating the linear component, they effectively eliminate linear stiffness, enabling efficient and accurate solutions.

However, like any numerical method, Fourier pseudo-spectral ETDRK solvers have inherent limitations:

1. **Periodic Domain and Uniform Cartesian Grid:** The method relies on the Fast Fourier Transform (FFT), which necessitates a periodic domain and a uniform Cartesian grid. This requirement stems from the diagonalization of the linear derivative operator in Fourier space, a crucial step for the method's effectiveness. While the general ETDRK framework can be adapted to other spectral methods like Chebyshev, the efficiency might be reduced.

2. **No Channel Mixing in Linear Part:** The method assumes that each equation in a system of PDEs depends solely on its own variables in the linear part. If there's "channel mixing," where the linear terms of one equation depend on variables from other equations, the linear operator becomes non-diagonal in Fourier space, leading to the method's breakdown.

3. **First-Order in Time:** ETDRK methods are specifically designed for first-order PDEs in time. Higher-order time derivatives do not conform to the method's structure. Attempts to reformulate higher-order PDEs into first-order systems often introduce channel mixing, rendering the method inapplicable.

4. **Smooth and Bandlimited Solutions:** The method assumes smooth and bandlimited solutions, meaning that the solution's Fourier spectrum decays rapidly at high frequencies. This limitation precludes the simulation of strongly hyperbolic PDEs with discontinuities, such as the inviscid Burgers, Euler, or shallow water equations. The method can only handle their viscous counterparts, where viscosity dampens high-frequency modes.

5. **Difficulty from Nonlinear Part:** When the primary challenge in integration arises from the nonlinear part, the advantage of analytically treating the linear part diminishes. The Navier-Stokes equations at high Reynolds numbers exemplify this scenario, where small time steps are necessary due to the dominant nonlinear effects.

6. **Cartesian Domains:** The method assumes a Cartesian domain. On a sphere, the linear part no longer diagonalizes, and the method breaks down.

Despite these limitations, if a problem aligns with the constraints, the Fourier pseudo-spectral ETDRK approach is one of the most efficient methods available for semi-linear PDEs on periodic boundaries (Montanelli and Bootland, 2020). Its tensor-based operations are well-suited for modern GPUs, and its straightforward integration with automatic differentiation frameworks like JAX simplifies the computation of derivatives.

### B.4 Limitations of the Implementation

Beyond the fundamental limitations of Fourier pseudo-spectral ETDRK methods, our specific implementation introduces additional constraints:

1. **Same Extent in Each Dimension:** We currently support only problems on scaled unit cubes, where each dimension has the same extent, i.e., $\Omega = (0, L)^D$.

2. **Equal Discretization Points in Each Dimension:** The number of discretization points $N$ is uniform across all dimensions. While this simplifies the interface, it limits the range of problems that can be addressed. However, we believe the remaining problem space remains substantial, especially for studying the learning dynamics of autoregressive neural emulators.

3. **Real-Valued PDEs Only:** Our implementation focuses on real-valued PDEs. Although the Fourier pseudo-spectral ETDRK method is also suitable for complex-valued PDEs, like the Schrödinger or complex Ginzburg-Landau equations, restricting to real-valued problems simplifies the interface and allows the exclusive use of the real-valued FFT, enhancing computational efficiency.

4. **Constant Time Step Size:** We currently require a constant time step size $\Delta t$, although ETDRK methods theoretically support adaptive time stepping. This decision was made to align with the specific requirements of training autoregressive neural emulators, which often have a fixed time step embedded in their architecture.

5. **Limited Set of ETDRK Methods:** Our implementation only includes the original ETDRK0, ETDRK1, ETDRK2, ETDRK3, and ETDRK4 methods. A recent study has shown that these methods remain competitive among solvers for stiff semi-linear PDEs (Montanelli and Bootland, 2020).

## B.5 Linear and Nonlinear Differential Operators

Our JAX-based package simplifies the implementation of ETDRK methods by requiring only the discrete linear differential operator in Fourier space ($\hat{\mathcal{L}}_h$), the discrete nonlinear differential operator ($\hat{\mathcal{N}}_h$), and the time step size ($\Delta t$). This modularity allows for easy customization of the solver for various dynamics.

Any linear operator can be represented by manipulating the scaled differential operator $\frac{2\pi}{L}ik_h$. This allows for arbitrary orders of derivatives, scaling by arbitrary coefficients, and spatial mixing in higher dimensions. However, as mentioned previously, channel mixing in the linear operator is not supported.

Custom nonlinear differential operators can also be defined, with the necessary dealiasing readily implemented. Additionally, a collection of common nonlinear operators is available:

1. **Convection Nonlinearity:**
   - 1D: $\mathcal{N}(u) = -\frac{1}{2}\frac{\partial u^2}{\partial x}$
   - Multi-D: $\mathcal{N}(u) = -\frac{1}{2}\nabla \cdot (u \otimes u)$[1] (number of channels scales with spatial dimensions)

2. **Gradient Norm Nonlinearity:**
   - 1D: $\mathcal{N}(u) = -\frac{1}{2}\left(\frac{\partial u}{\partial x}\right)^2$
   - Multi-D: $\mathcal{N}(u) = -\frac{1}{2}\|\nabla u\|^2$ (single channel)

3. **Polynomial Nonlinearity:**
   - $\mathcal{N}(u) = \sum_{j=0}^{D} c_j u^j$ (element-wise, arbitrary channels)

4. **Vorticity Convection Nonlinearity (2D Navier-Stokes in stream-function vorticity):**
   - $\mathcal{N}(u) = -\left(\begin{bmatrix} 1 \\ -1 \end{bmatrix} \odot \nabla(\Delta^{-1}u)\right) \cdot \nabla u$ (requires Poisson solve)

5. **Single Channel Convection:**
   - $\mathcal{N}(u) = -\frac{1}{2}(\nabla \cdot \vec{1})u^2$ (multi-D, single channel)

6. **General Nonlinearity:**
   - $\mathcal{N}(u) = b_0 u^2 + b_1 \frac{1}{2}(\nabla \cdot \vec{1})u^2 + b_2 \frac{1}{2}\|\nabla u\|^2$ (combines quadratic, single-channel convection, and gradient norm)

---

[1]We choose this particular "conservative" formulation of the convection nonlinearity for our main experiments. It differs from the convection nonlinearity that arises when deriving from first principles ($\mathcal{N}(u) = -(u \cdot \nabla)u$). Both versions are available in our implementation.

## B.6 Generic Time Steppers for General Dynamics

Our framework, based on defining linear and nonlinear differential operators, enables us to cover a wide range of PDEs. Often, PDEs exhibit structural similarities. For instance, transitioning from the Burgers equation to the Korteweg-de Vries (KdV) equation involves simply changing the order of the linear term from second to third. Moreover, the viscous KdV equation encompasses the Burgers equation as a special case when dispersivity is absent.

This observation motivates the development of generic time steppers that accommodate arbitrary combinations of linear and nonlinear differential operators. To achieve this, we focus on isotropic linear operators, which lack spatial mixing (e.g., cross-derivatives). This allows us to represent the linear operator uniquely by a list of coefficients $\{a_j\}_{j=0}^{S}$, where S is the highest derivative order considered. Each coefficient corresponds to the scaling of a particular derivative order, with zeros indicating the absence of specific terms.

By combining this linear operator representation with various nonlinearities, we create a collection of versatile time steppers:

1. **General Linear Stepper:** This stepper handles purely linear dynamics, with the 1D and higher-dimensional forms given by:

$$\frac{\partial u}{\partial t} = \sum_j a_j \frac{\partial^j u}{\partial x^j} \quad \text{and} \quad \frac{\partial u}{\partial t} = \sum_j a_j (\nabla^j \cdot \vec{1})u,$$

   respectively.

2. **General Convection Stepper:** This stepper incorporates the convection nonlinearity found in the Burgers equation:

$$\frac{\partial u}{\partial t} = \sum_j a_j \frac{\partial^j u}{\partial x^j} + b\frac{1}{2}\nabla \cdot (u \otimes u).$$

3. **General Gradient Norm Stepper:** This stepper includes the gradient norm nonlinearity, present in the Kuramoto-Sivashinsky equation:

$$\frac{\partial u}{\partial t} = \sum_j a_j \frac{\partial^j u}{\partial x^j} + b\frac{1}{2}\|\nabla u\|^2.$$

4. **General Polynomial Stepper:** This stepper allows for arbitrary polynomial nonlinearities:

$$\frac{\partial u}{\partial t} = \sum_j a_j \frac{\partial^j u}{\partial x^j} + \sum_j c_j u^j.$$

5. **General Vorticity Convection Stepper:** This stepper caters to the two-dimensional Navier-Stokes equations in the streamfunction-vorticity formulation:

$$\frac{\partial u}{\partial t} = \sum_j a_j \frac{\partial^j u}{\partial x^j} + b\left(\begin{bmatrix} 1 \\ -1 \end{bmatrix} \odot \nabla(\Delta^{-1}u)\right) \cdot u.$$

6. **General Nonlinear Stepper:** This is the most comprehensive stepper, combining quadratic polynomial, single-channel convection, and gradient norm nonlinearities:

$$\frac{\partial u}{\partial t} = \sum_j a_j \frac{\partial^j u}{\partial x^j} + b_0 u^2 + b_1 \frac{1}{2}(\nabla \cdot \vec{1})u^2 + b_2 \frac{1}{2}\|\nabla u\|^2.$$

To illustrate, the Burgers equation can be expressed as a special case of the general convection stepper with coefficients $a_0 =, a_1 = 0, a_2 = \nu$, and $b_1 = -1$. This flexibility showcases the power of our framework in accommodating a wide range of PDEs with diverse linear and nonlinear components.

### B.7 Normalized Dynamics: A Unifying Framework

When simulating the dynamics of partial differential equations (PDEs), it's crucial to identify the parameters that uniquely determine their behavior. For instance, the one-dimensional advection equation is governed by the domain extent $L$, advection speed $c$, and time step size $\Delta t$. However, the dynamics remain unchanged as long as the ratio $c\Delta t/L$ stays constant. This observation leads to the concept of *normalized dynamics*, a framework that unifies the characterization of diverse PDEs.

**Generalizing to Linear Operators**  This concept extends beyond advection. In the diffusion equation, the dynamics are uniquely determined by the ratio $\nu\Delta t/L^2$, where $\nu$ represents the diffusion coefficient. Similarly, for dispersion and hyper-diffusion equations, the governing ratios are $\xi\Delta t/L^3$ and $\zeta\Delta t/L^4$, respectively. We can generalize this observation to any linear operator involving the $j$-th derivative with coefficient $a_j$. The normalized dynamics, denoted by $\alpha_j$, are given by

$$\alpha_j = \frac{a_j\Delta t}{L^j}. \tag{22}$$

**Nonlinear Operators and Composite Dynamics**  For nonlinear operators, the situation becomes more nuanced. We must consider the interplay between the order of derivatives, the nonlinearity itself, and any subsequent derivatives. Taking a coefficient $b_{l_{\text{pre}},p,l_{\text{post}}}$, where $l_{\text{pre}}$ and $l_{\text{post}}$ represent the orders of derivatives before and after the nonlinearity, and $p$ denotes the order of the polynomial, the normalized coefficient is expressed as

$$\beta_{l_{\text{pre}},p,l_{\text{post}}} = \frac{b_{l_{\text{pre}},p,l_{\text{post}}}\Delta t}{L^{l_{\text{pre}}\cdot p+l_{\text{post}}}}. \tag{23}$$

This expression accounts for the "amplification" of derivatives before the nonlinearity due to the polynomial order. Notably, any polynomial nonlinearity without additional derivatives is normalized by $L^0 = 1$.

**Practical Implications for Neural Emulators**  Neural emulators are designed to learn and emulate the dynamics of complex systems, and their performance is inherently linked to the speed and nature of those dynamics. By characterizing dynamics with a reduced set of normalized coefficients, we simplify the assessment of neural emulator performance. Instead of manipulating multiple parameters, we can focus on varying the relevant normalized coefficients, enabling a more targeted and efficient evaluation.

### B.8 Difficulty of Emulating Dynamics: Bridging the Continuous and Discrete

While normalized dynamics effectively characterize the time-discrete form of a PDE, they do not fully capture the challenges associated with emulating those dynamics in a space-discrete setting. Discretizing a PDE introduces additional complexities due to finite spatial resolution, numerical approximations, and the potential for instabilities. Thus, understanding the difficulty of emulation requires considering both the continuous nature of the dynamics, as captured by the normalized coefficients, and the discrete aspects of the numerical implementation.

The spatial resolution, represented by the number of grid points $N$ in each dimension $D$, plays a critical role in emulation. As $N$ increases, the receptive field of convolutional architectures, which is given in terms of cells per directions, spans a much smaller physical area. Stability is a key factor in the emulation process. Explicit numerical methods, which compute future states directly from current values, often have stability limitations. For instance, the Courant-Friedrichs-Lewy (CFL) condition dictates the maximum allowable time step for a first-order upwind scheme applied to the advection equation by

$$\text{CFL} = \frac{a_1\Delta t}{\Delta x} \leq 1.$$

This condition ensures that information does not propagate faster than one grid cell per time step, preventing numerical instabilities. Interestingly, the CFL condition can be expressed in terms of the

normalized dynamics and spatial resolution as

$$\text{CFL} = \alpha_1 N.$$

Similarly, for diffusion and higher-order equations, as well as in higher dimensions, we can define analogous difficulty factors $\gamma_j$ by

$$\gamma_j = \alpha_j N^j 2^{j-1} D. \tag{24}$$

These factors provide a quantitative measure of emulation difficulty. When $\gamma_j$ approaches 1, we are nearing the stability limit of (the most compact) explicit finite difference method. Similarly, we compute the difficulty of supported nonlinear components as

$$\delta_j = \beta_j N^j D m \tag{25}$$

with $m$ denoting the (expected) maximum absolute of the state throughout the trajectory.

Ultimately, in order to identify a dynamic (and its difficulty of emulation), it is sufficient to know the respective nonzero $\gamma_j$ and $\delta_j$ values (the defaults used for the dynamics in APEBench are listed in table 2) and the resolution $N$ (and the dimension $D$). In this benchmark suite, difficulties serve as an exchange protocol or an identifier for an experiment where possible.

## C   Emulator Architectures: Leveraging Equinox for Flexible and Powerful Neural PDE Emulators

Our suite of neural PDE emulator architectures is built upon the *Equinox* library (Kidger and Garcia, 2021) for JAX (Bradbury et al., 2018). We provide seamless support for various boundary conditions (Dirichlet, Neumann, and periodic) and enable architectures that are agnostic to spatial dimensions (1D, 2D, and 3D). Our implementations are inspired by PDEArena (Gupta and Brandstetter, 2023) to a large extent.

**Wrapped Convolutions and Building Blocks**   At the core, we provide higher-level abstractions for convolutional layers that allow defining the boundary condition instead of the padding kind and padding amount. Internally, we set up the corresponding padding to ensure a "SAME" convolution. We also implemented a spectral convolution which is agnostic to the spatial dimensions to build FNOs. This routine is inspired by the generalized spectral convolution of the *Serket* library [2]. We combine the fundamental convolutional modules into blocks, such as residual or downsampling blocks.

**Architectural Constructors and Curated Networks**   Based on the blocks, we have architectural constructors for sequential and hierarchical networks. With those, we provide a range of curated architectures, including the ones used in main text: feedforward convolutional networks (Conv), convolutional ResNets (Res) (He et al., 2016), UNets (Ronneberger et al., 2015), Dilated ResNets (Dil) (Stachenfeld et al., 2021), and Fourier Neural Operators (FNO) (Li et al., 2021).

**Diagnostic Tools for Deeper Insights**   We equip our framework with diagnostic tools that provide valuable insights into the behavior and performance of our neural PDE emulators:

- **Parameter Counting:** Accurately tracking the number of trainable parameters in a neural network is crucial for understanding its complexity.
- **Effective Receptive Field:** For convolutional architectures, determining the effective receptive field reveals the spatial extent over which the network can gather information, a key factor influencing its ability to capture long-range dependencies relevant for fast moving dynamics.

**Notable Advantages of using Equinox & JAX**

- **Single-Batch by Design**: All emulators have a call signature that does not require arrays to have a leading batch axis. Vectorized operation is achieved with the `jax.vmap` transformation. We believe that this design more closely resembles how classical simulators are usually set up.

---

[2]https://github.com/ASEM000/Serket

- **Seed-Parallel Training**: With only a little additional modification, the automatic vectorization of `jax.vmap` can also be used to run multiple initialization seeds in parallel. This approach is especially helpful when a training run of one network does not fully utilize an entire GPU, like in all 1D scenarios. Essentially, this allows for free seed statistics.
- **Similar Python Structure for Neural Network and ETDRK Solver**: Both simulator and emulator are implemented as `equinox.Module` allowing them to operate seamlessly with one another.

## D  Training methodologies

The general objective for unrolled training is

$$L(\theta) = \mathbb{E}_{u_h \propto \mathcal{D}_h}\left[\sum_{t=0}^{(T-B)}\sum_{b=1}^{B} w_t w_b\, l\left(f_\theta^{t+b}(u_h),\ \mathcal{P}_h^b(f_\theta^t(u_h))\right)\right], \tag{26}$$

with $l(\cdot, \cdot)$ being a time-level loss, which typically is the mean squared error (MSE). Optional time step weights $w_t$ and $w_b$ can be supplied to differently weigh contributions (e.g., to exponentially discount the error over unrolled steps). We used the notation that a function raised to an exponent denotes an autoregressive/recursive application. If it is raised to zero, this should be interpreted as the function not being applied (i.e., resorting to the identity). In this case, supervised unrolling ($T = B$) would only leave the application of the fine solver $\mathcal{P}_h$ in the second entry and the second sum over the branch chain (together with $t = 0$), which then reads

$$L_{\text{supervised unrolled}}(\theta) = \mathbb{E}_{u_h \propto \mathcal{D}_h}\left[\sum_{b=1}^{B=T} l(f_\theta^b(u_h), \mathcal{P}_h^b(u_h))\right]. \tag{27}$$

Clearly, we recover the popular one-step supervised training with $T = B = 1$, which leaves only one summand. Beyond that, the main text investigates *diverted chain* unrolled training with a freely chosen rollout length $T$ and $B = 1$ for a one-step difference. It requires the numerical simulator $\mathcal{P}_h$ on the fly and in a differentiable way. It reads

$$L_{\text{diverted chain (branch length 1)}}(\theta) = \mathbb{E}_{u_h \propto \mathcal{D}_h}\left[\sum_{t=0}^{T-1} l(f_\theta^{t+1}(u_h), \mathcal{P}_h(f_\theta^t(u_h)))\right]. \tag{28}$$

Again, there are as many loss contributions as time steps in the main chain. APEBench also supports the most general case with $T$ freely chosen and $B \geq 2$. In this case, one would get cross terms, and both sums remain. For brevity, we did not present any results under such a configuration.

Figure 7 displays a schematic for a three-step supervised unrolled objective. The curvy lines indicate the gradient flow. A purple gradient flow represents input-output differentiation over the neural emulator, which causes the backpropagation through time. Similarly, we present the three-step diverted chain unrolled training schematic in Figure 8. The yellow box around the references denotes that only the target for the first one-step difference can be precomputed. All targets beyond that require the reference solver to be called dynamically. Consequently, we must also differentiate over it, indicated by the yellow reversely pointing arrow. In Figure 9, we present the more general scenario of a diverted-chain setup with $T = 3$ and $B = 2$.

### D.1  Continued Taxonomy

Gradient cuts interrupt the reverse flow in automatic differentiation and decouple the loss landscape from the gradient landscape. Oftentimes, they are used strategically to avoid the problem of vanishing or exploding gradients. In the case of training rollout, this can become relevant for long main chains (List et al., 2022). Alternatively, they can be used with different motivations, for example, to better compensate for distribution shift (Brandstetter et al., 2022). It can also be used to avoid undesirable minima and saddle points (Schnell and Thuerey, 2024). One can also imagine a strategic application to avoid having the simulator be differentiable (List et al., 2024).

Generally speaking, APEBench allows for all the aforementioned approaches by either (sparsely) cutting the backpropagation through time and/or the differentiable physics. For example, the pushforward

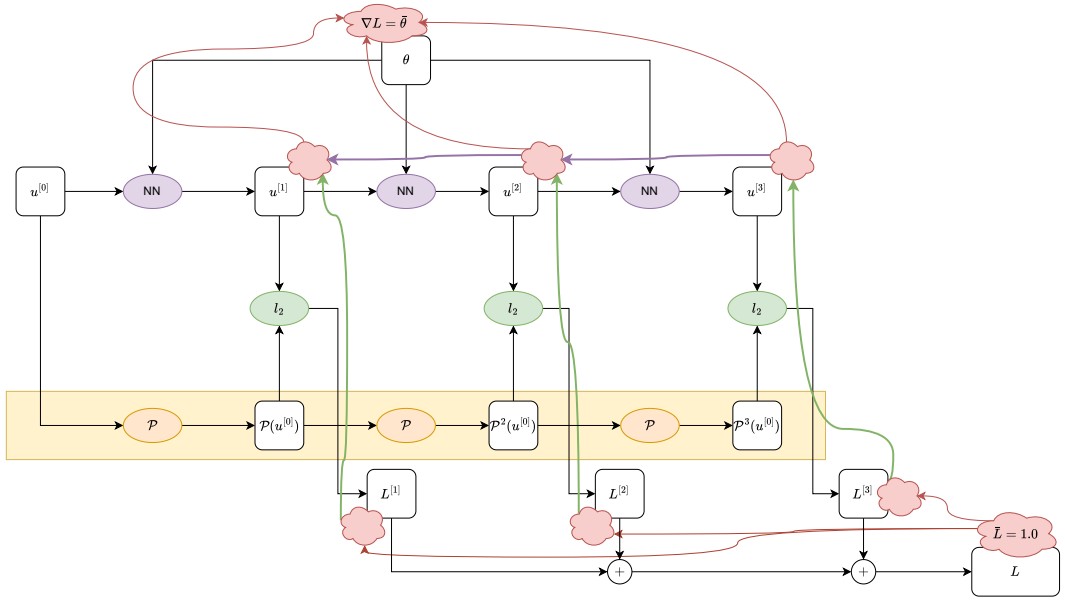

Figure 7: Three step supervised learning. The reference simulator $\mathcal{P}$ produces a reference trajectory, which is as long as the main chain of autoregressive network calls. Since this reference is computed starting from a known initial condition, it can be fully precomputed, allowing the reference simulator to be turned off during training. Backpropagation-through-time is indicated by the purple arrow.

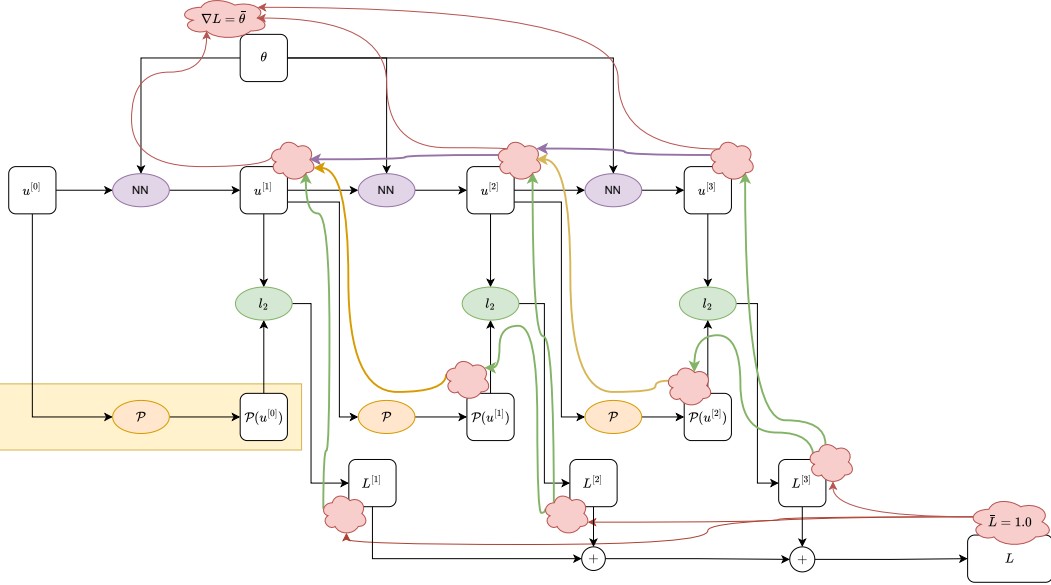

Figure 8: Three step diverted-chain learning. The autoregressive main chain is the same as in a similar-length supervised setting (see Figure 7). However, the reference simulator $\mathcal{P}$ is called dynamically on the outputs produced by the network to generate a one-step error. Only the reference for the first one-step difference can be precomputed. For the latter two, the reference solver has to be called during training. As such, we also require it to be differentiable as seen by the gradient flow indicated by the yellow arrow.

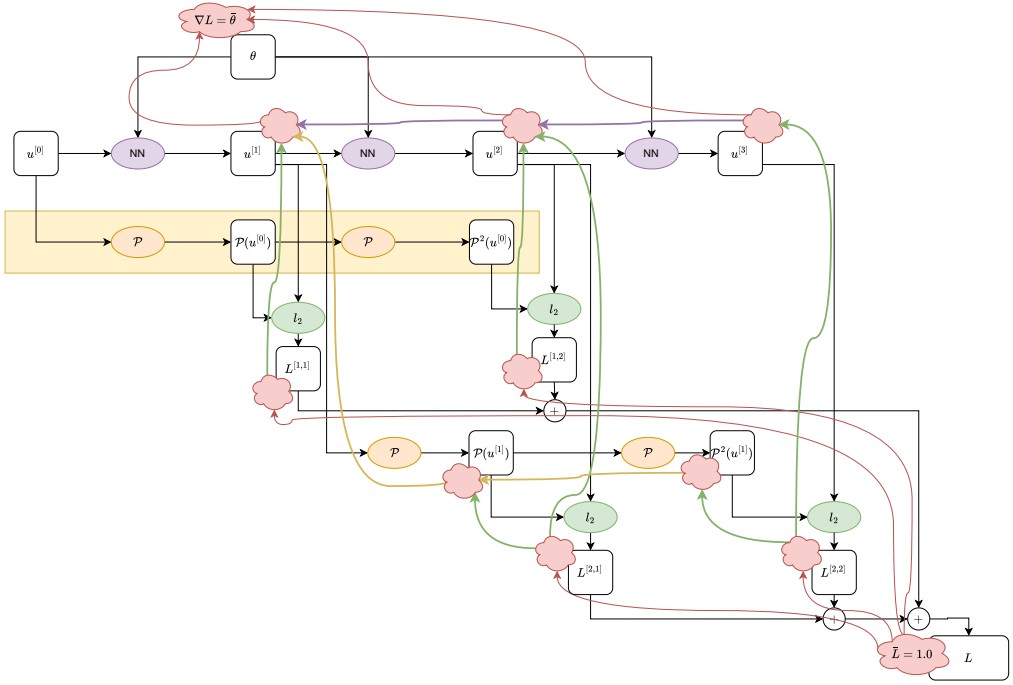

Figure 9: Three step diverted chain learning but branches of length two.

trick of Brandstetter et al. (2022) can be described with equation 26, by setting $T = B = 2$ using the weights $w_1 = 0$ and $w_2 = 0$ (weighing only the second step) and the backpropagation-through-time being disabled.

### D.2 Differentiable Physics in Learning Setups

Below, we provide a decision guide as to which setup requires the autodifferentiable solver. For each version, a green color indicates whether we need an autodifferentiable solver. A setup with purple color does require a solver to be called on-the-fly, but does not require differentiability. To the best of our knowledge, configurations with either of the two colors are unique to APEBench and have not yet been part of other benchmark publications. Pure prediction means that the neural emulator $f_\theta$ is exclusively made up of the neural network. On the other hand, a correction task is defined in that the neural emulator contains both a neural network and a coarse solver component $\tilde{\mathcal{P}}_h$ as discussed in section E.3.

1. One-Step Supervised Training ($T = B = 1$):
    (a) Pure Prediction: No autodifferentiable solver required.
    (b) Correction: Dynamically called coarse solver required, but not autodifferentiable.
2. Supervised Unrolling ($T = B$):
    (a) Pure Prediction: No autodifferentiable solver required.
    (b) Correction:
        i. If using a coarse solver without modifications: Autodifferentiable coarse solver required.
        ii. If backpropagation-through-time (BPTT) is disabled: Dynamically called coarse solver required, does not need to be differentiable (List et al., 2024).
3. Diverted Chain Unrolling ($T > B$):
    (a) Pure Prediction:
        i. If using the fine solver without modifications: Autodifferentiable fine solver required.

    ii. If the diverted chain is cut: Dynamically called fine solver required, does not need to be differentiable.

  (b) Correction:

    i. If using both fine and coarse solvers without modifications: Both autodifferentiable fine and coarse solvers required.

    ii. If the gradients through the diverted chain are cut: Dynamically called but not autodifferentiable fine solver required, along with an autodifferentiable coarse solver.

    iii. If BPTT is cut: Dynamically called but not autodifferentiable coarse solver required, along with an autodifferentiable fine solver.

# E   APEBench Details

In APEBench, we call a *scenario* the fully encapsulated pipeline to train and evaluate an autoregressive neural emulator. Aside from smaller statistical variations, it is a fully reproducible setup since it procedurally generates its training data. Each scenario embeds a *dynamic* defined by the continuous PDE to be emulated, its constitutive parameters, and numerical discretization choices. In the simplest case, one uses the reduced interface via *difficulties*.

## E.1   Available Dynamics

In Table 1, we give an overview of the dynamics available within APEBench. Note that some equations only have versions in 2D/3D (most interesting reaction-diffusion problems), while some only have 2D versions (Navier-Stokes in stream function vorticity).

Additionally, we classify equations/dynamics as follows:

- (L)inear
- (N)onlinear
- (D)ecaying
- (I)infinitely running (essentially the opposite of decaying)
- Reaching a (S)teady state (reaching a state that is not a domain-wide constant)
- (M)ulti-Channel (if the number of channels is greater than one)
- (C)haotic (if the system is sensitive to the initial condition)

All (L)inear problems stay bandlimited if they start from a bandlimited initial condition. This might be different for (N)nonlinear problems, which can produce higher modes and can become unresolved. This can lead to instability in the simulation, which might require a stronger diffusion.

Additionally, we count how many individual dynamics each equation contributes depending on how many spatial dimensions it is available in.

Table 1: Overview of the PDE dynamics available in APEBench through its efficient pseudo-spectral solver. Class indicates the underlying properties of the dynamics. # highlights the number of scenarios it contributes to APEBench based on the number of spatial dimensions it is available for.

| Name | Dynamic | Class | # | 1D Equation | 2D/3D Equation |
|------|---------|-------|---|-------------|----------------|
| adv | Advection | L-I | 3 | $\frac{\partial u}{\partial t} = -c\frac{\partial u}{\partial x}$ | $\frac{\partial u}{\partial t} = -c\vec{1} \cdot \nabla u$ |
| diff | Diffusion | L-D | 3 | $\frac{\partial u}{\partial t} = \nu\frac{\partial^2 u}{\partial x^2}$ | $\frac{\partial u}{\partial t} = \nu\nabla \cdot \nabla u$ |
| adv_diff | Advection-Diffusion | L-D | 3 | $\frac{\partial u}{\partial t} = -c\frac{\partial u}{\partial x} + \nu\frac{\partial^2 u}{\partial x^2}$ | $\frac{\partial u}{\partial t} = -c\vec{1} \cdot \nabla u + \nu\nabla \cdot \nabla u$ |
| disp | Dispersion | L-I | 3 | $\frac{\partial u}{\partial t} = \xi\frac{\partial^3 u}{\partial x^3}$ | $\frac{\partial u}{\partial t} = \xi\vec{1} \cdot \nabla^3 u$ |
| hyp | Hyper-Diffusion | L-D | 3 | $\frac{\partial u}{\partial t} = -\zeta\frac{\partial^4 u}{\partial x^4}$ | $\frac{\partial u}{\partial t} = -\zeta\vec{1} \cdot \nabla^4 u$ |
| unbal_adv | Unbalanced Advection | L-I | 2 | | $\frac{\partial u}{\partial t} = -\vec{c} \cdot \nabla u$ |

Table 1: Overview of the PDE dynamics available in APEBench through its efficient pseudo-spectral solver. Class indicates the underlying properties of the dynamics. # highlights the number of scenarios it contributes to APEBench based on the number of spatial dimensions it is available for.

| Name | Dynamic | Class | # | 1D Equation | 2D/3D Equation |
|---|---|---|---|---|---|
| diag_diff | Diagonal Diffusion | L-D | 2 | | $\frac{\partial u}{\partial t} = \nabla \cdot (\vec{\nu} \odot \nabla u)$ |
| aniso_diff | Anisotropic Diffusion | L-D | 2 | | $\frac{\partial u}{\partial t} = \nabla \cdot A \nabla u$ |
| mix_disp | Spatially-Mixed Dispersion | L-I | 2 | | $\frac{\partial u}{\partial t} = \xi \vec{1} \cdot \nabla(\nabla \cdot \nabla u)$ |
| mix_hyp | Spatially-Mixed Hyper-Diffusion | L-I | 2 | | $\frac{\partial u}{\partial t} = -\zeta(\nabla \cdot \nabla)(\nabla \cdot \nabla u)$ |
| burgers | Burgers [3] | N-D-M | 3 | $\frac{\partial u}{\partial t} = -b\frac{1}{2}\frac{\partial u^2}{\partial x} + \nu\frac{\partial^2 u}{\partial x^2}$ | $\frac{\partial u}{\partial t} = -b\frac{1}{2}\nabla \cdot (u \otimes u) + \nu\nabla \cdot \nabla u$ |

---

[3] See the footnote under the definition of the convection nonlinearity why we chose this particular form in higher dimensions. APEBench also supports a non-conservative via $(u \cdot \nabla)u$.

Table 1: Overview of the PDE dynamics available in APEBench through its efficient pseudo-spectral solver. Class indicates the underlying properties of the dynamics. # highlights the number of scenarios it contributes to APEBench based on the number of spatial dimensions it is available for.

| Name | Dynamic | Class | # | 1D Equation | 2D/3D Equation |
|------|---------|-------|---|-------------|----------------|
| `burgers_sc` | Burgers (single-channel) | N-D | 2 | | $\frac{\partial u}{\partial t} = -b\frac{1}{2}(\vec{1}\cdot\nabla)u^2 + \nu\nabla\cdot\nabla u$ |
| `kdv` | Korteweg-de-Vries (single-channel) | N-D | 3 | $\frac{\partial u}{\partial t} = -b\frac{1}{2}\frac{\partial u^2}{\partial x} + \xi\frac{\partial^3 u}{\partial x^3} - \zeta\frac{\partial^4 u}{\partial x^4}$ | $\frac{\partial u}{\partial t} = -b\frac{1}{2}(\vec{1}\cdot\nabla)u + \xi\vec{1}\cdot\nabla^3 u - \zeta\vec{1}\cdot\nabla^4 u$ |
| `ks_cons` | Kuramoto-Sivashinsky (conservative) | N-I-C | 1 | $\frac{\partial u}{\partial t} = -b\frac{1}{2}\frac{\partial u^2}{\partial x} - \nu\frac{\partial^2 u}{\partial x^2} - \zeta\frac{\partial^4 u}{\partial x^4}$ | |
| `ks` | Kuramoto-Sivashinsky (combustion) | N-I-C | 3 | $\frac{\partial u}{\partial t} = -b\frac{1}{2}\left(\frac{\partial u}{\partial x}\right)^2 - \nu\frac{\partial^2 u}{\partial x^2} - \zeta\frac{\partial^4 u}{\partial x^4}$ | $\frac{\partial u}{\partial t} = -b\frac{1}{2}\|\nabla u\|^2 - \nu\nabla\cdot\nabla u - \zeta\vec{1}\cdot\nabla^4 u = 0$ |
| `fisher` | Fisher-KPP | N-S | 3 | $\frac{\partial u}{\partial t} = \nu\frac{\partial^2 u}{\partial x^2} + ru(1-u)$ | $\frac{\partial u}{\partial t} = \nu\nabla\cdot\nabla u + ru(1-u)$ |

Table 1: Overview of the PDE dynamics available in APEBench through its efficient pseudo-spectral solver. Class indicates the underlying properties of the dynamics. # highlights the number of scenarios it contributes to APEBench based on the number of spatial dimensions it is available for.

| Name | Dynamic | Class | # | 1D Equation | 2D/3D Equation |
|---|---|---|---|---|---|
| gs | Gray-Scott | N-S/C/I-M | 2 | | $\dfrac{\partial u_0}{\partial t} = \nu_0 \nabla \cdot \nabla u_0 - u_0 u_1^2 + f(1 - u_0)$ 
 $\dfrac{\partial u_1}{\partial t} = \nu_1 \nabla \cdot \nabla u_1 + u_0 u_1^2 - (f + k)u_1$ |
| sh | Swift-Hohenberg | N-S | 2 | | $\dfrac{\partial u}{\partial t} = ru - (k + \nabla \cdot \nabla)^2 u + u^2 - u^3$ |
| decay_turb | Navier-Stokes (streamfunction vorticity) | N-D | 1 | | $\dfrac{\partial u}{\partial t} = -b\left( \begin{bmatrix} 1 \\ -1 \end{bmatrix} \odot \nabla(\Delta^{-1}u) \right) \cdot \nabla u + \nu \nabla \cdot \nabla u$ |
| kolm_flow | Navier-Stokes (Kolmogorov forcing) | N-I-C | 1 | | $\dfrac{\partial u}{\partial t} = -b\left( \begin{bmatrix} 1 \\ -1 \end{bmatrix} \odot \nabla(\Delta^{-1}u) \right) \cdot \nabla u + \nu \nabla \cdot \nabla u + \lambda u - k \cos(k \frac{2\pi}{L} y)$ |
| | Sum | | 46 | | |

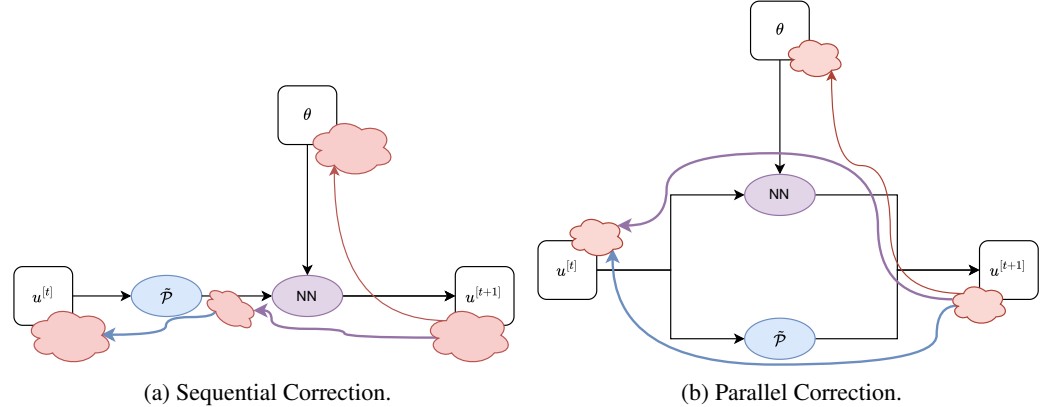

| (a) Sequential Correction. | (b) Parallel Correction. |

Figure 10: Correction setups for a coarse solver $\tilde{\mathcal{P}}$ using a neural network. Curvy arrows depict gradient flow in backpropagation, purple signifies input-output differentiation over the network, and blue is associated with *differentiable* physics over the coarse solver.

## E.2 Preferred Interface Mode and Typical Values

Most dynamics listed in Table 1 are best addressed using the difficulties mode (with gammas and deltas). Alternatively, the normalized interface via alphas and betas can also be chosen. Some equations are only available in the physical parameterization. This includes the special linear scenarios of `unbal_adv`, `diag_diff`, `aniso_diff`, `mix_disp`, and `mix_hyp` since their spatial mixing does not align with the requirements of isotropic linear derivatives. The reaction-diffusion scenarios of Gray-Scott and Swift-Hohenberg are also only available in physical parameterization to more closely follow prior research (Pearson, 1993). For the Navier-Stokes dynamics, we decided to also only have them in physical parameterization since these dynamics are typically adjusted via the Reynolds number. The Reynolds number, the resolution $N$, and the time step $\Delta t$ determine the emulation's difficulty. Table 2 lists the default values for each dynamics in APEBench under their preferred mode. For nonlinear difficulties, we use a maximum absolute $m = 1$ for simplicity (see Eq. 25), which aligns in that most initial conditions are ensured to have an absolute magnitude of one.

The default discretization is $N = 160$ in 1D, $N = 160^2$ in 2D, and $N = 32^3$ in 3D. Note that some dynamics require specific initial conditions.

**Accessing Specific Modes** To utilize a specific scenario in a particular interface mode (DIFF, NORM, or PHY), simply prepend the desired mode to the scenario name. After the scenario has been executed, APEBench will automatically prepend the spatial dimension to the scenario name for clarity. For instance, to execute the Burgers scenario using the difficulty interface in three dimensions, the user would input `diff_burgers` with `num_spatial_dims=3`. The name assigned will be `3d_diff_burgers`.

**Gray-Scott Dynamics** For the Gray-Scott scenarios, we use an interface to directly define the feed rate $f$ and kill rate $k$. Alternatively, there is also the `gs_type` interface, which requires defining the dynamics type. We chose the values according to Pearson (1993), and the defaults are listed in Table 3.

Particularly interesting are the alpha, epsilon and theta pattern. The former is periodic pattern movement, the second is spot multiplication and the latter is pattern formation.

## E.3 Correction Tasks and Neural-Hybrid Emulators

While the default task for neural emulators in APEBench is *prediction*, in which the neural network completely replaces the numerical simulator, an alternative *correction* mode is also available. In this mode, a coarse solver ($\tilde{\mathcal{P}}_h$) and the neural network work together to form a *neural-hybrid emulator*. The default configuration corrects a "defective" solver, which only performs a portion of the integration time step compared to the reference simulator.

Table 2: Overview of (**preferred**) interface mode (DIFFiculty, NORMalized, and PHYsical) for all scenarios listed in Table 1. $q$ is the number of substeps. $w$ is the number of warmup steps (relevant only for chaotic problems). $\delta_{XXX}$ refers to the difficulty of the following nonlinear differential operators: CONVection, CONVection_SingleChannel, GradientNorm, and QUADratic polynomial.

| Name | Mode | Default Values under preferred mode |
|------|------|-------------------------------------|
| adv | **diff**,norm,phy | $\gamma = \{0, -4, 0, 0, 0\}$ |
| diff | **diff**,norm,phy | $\gamma = \{0, 0, 4, 0, 0\}$ |
| adv_diff | **diff**,norm,phy | $\gamma = \{0, -4, 4, 0, 0\}$ |
| disp | **diff**,norm,phy | $\gamma = \{0, 0, 0, 4, 0\}$ |
| hyp | **diff**,norm,phy | $\gamma = \{0, 0, 0, 0, -4\}$ |
| unbal_adv | **phy** | $L = 1, \Delta t = 0.1, \vec{c} = [0.01, -0.04, (0.005)]^T$ |
| diag_diff | **phy** | $L = 1, \Delta t = 0.1, \nu = [0.001, 0.002, (0.0004)]^T$ |
| aniso_diff | **phy** | $L = 1, \Delta t = 0.1, A = [0.001, 0.0005; 0.0005, 0.002]$ |
| mix_disp | **phy** | $L = 1, \Delta t = 0.001, \xi = 0.00025$ |
| mix_hyp | **phy** | $L = 1, \Delta t = 0.00001, \zeta = -0.000075$ |
| burgers | **diff**,norm,phy | $\gamma = \{0, 0, 1.5, 0, 0\}, \delta_{\text{conv}} = -1.5$ |
| burgers_sc | **diff**,norm,phy | $\gamma = \{0, 0, 1.5, 0, 0\}, \delta_{\text{conv\_sc}} = -1.5$ |
| kdv | **diff**,norm,phy | $\gamma = \{0, 0, 0, -14, -9\}, \delta_{\text{conv\_sc}} = -2$ |
| ks_cons | **diff**,norm,phy | $\gamma = \{0, 0, -2, 0, -18\}, \delta_{\text{conv}} = -1, w = 500$ |
| ks | **diff**,norm,phy | $\gamma = \{0, 0, -1.2, 0, -15\}, \delta_{\text{gn}} = -6, w = 500$ |
| fisher | **diff**,norm,phy | $\gamma = \{0.02, 0, 0.2, 0, 0\}, \delta_{\text{quad}} = -0.02$ |
| gs | **phy** | $L = 1.0, \Delta t = 10, q = 10, \nu_0 = 2 \cdot 10^{-5}, \nu_1 = 10^{-5},$ 
 $f = 0.04, k = 0.06$ |
| gs_type | **phy** | $L = 2.5, \Delta t = 20, q = 20, \nu_0 = 2 \cdot 10^{-5}, \nu_1 = 10^{-5}$ 
 $t = \text{theta}$ |
| sh | **phy** | $L = 10\pi, \Delta t = 0.1, q = 5, r = 0.7, k = 1.0$ |
| decay_turb | **phy** | $L = 1, \Delta t = 0.1, \nu = 0.0001$ |
| kolm_flow | **phy** | $L = 2\pi, \Delta t = 0.1, \lambda = -0.1, k = 4, \nu = 1/Re,$ 
 $Re = 100, q = 20, w = 500$ |

Table 3: Different types of dynamics for the Gray-Scott equation depending on the feed and kill rate values. The values are extracted from Figure 1 of Pearson (1993).

| Type $t$ | Feed Rate $f$ | Kill Rate $k$ |
|----------|---------------|---------------|
| alpha | 0.008 | 0.046 |
| beta | 0.020 | 0.046 |
| gamma | 0.024 | 0.056 |
| delta | 0.028 | 0.056 |
| epsilon | 0.02 | 0.056 |
| theta | 0.04 | 0.06 |
| iota | 0.05 | 0.0605 |
| kappa | 0.052 | 0.063 |

The two main corrective layouts are **sequential** (Figure 10a) and **parallel** (Figure 10b). APEBench allows the selection of either layout, including modified versions that incorporate gradient cuts (List et al., 2024). For the experiments presented in this paper, sequential correction was chosen, as it facilitates the sharing of the necessary receptive field between the coarse solver and the neural network.

## F  Metrics

APEBench supports a range of metric functions to compare two discrete simulation states $u_h$ and $u_h^r$ or reduce a single simulation state to a scalar value. In neural emulator learning, next to the discrete states containing a value to each spatial degree of freedom, they can also hold multiple channels (=species/fields) and more than one sample.

**Normalization & Aggregation Process**  Normalization is applied independently per channel. Subsequently, each channel's contribution is summed, and the average across all samples is calculated. This process ensures consistent metric computation as follows:

- For two states with both batch, channel and spatial axes
- Per sample in the batch axis:
  - Per field in the channel axis
    * Compute the difference between prediction and target
    * Reduce the difference over all spatial degrees of freedom
    * Either:
      1. Do not divide by anything to obtain an *absolute* metric
      2. Divide by the spatially reduced target to obtain a *normalized* metric
      3. Divide by the average of spatially reduced target and prediction to obtain a *symmetric* metric
  - Sum over all channels
- Take the average across all samples

Since some operations (e.g., spatial aggregation and division) are nonlinear, they are applied in this precise order to ensure accuracy.

**Metrics Categorization in APEBench**  APEBench's metrics can be broadly categorized based on several criteria:

1. How the spatial axes are aggregated
   (a) In the state space
   (b) In Fourier space
2. To what exponent the state or difference in states are raised before and after aggregation in space yielding
   (a) MSE (inner exponent 2, outer exponent 1)
   (b) MAE (inner exponent 1, outer exponent 1)
   (c) RMSE (inner exponent 2, outer exponent $1/2$)
   (d) more combinations are possible
3. How two states are compared
   (a) Via their difference
   (b) Via their inner product (i.e., a correlation)
4. Whether the difference in states is further normalized
   (a) Is it not normalized
   (b) Via the reduction of the target state $u_h^r$ (normalized/relative metrics)
   (c) Via the average of the reduction of both simulation states (symmetric metrics)
5. If modifications in the spectrum are made

Table 4: Overview of the metric functions supported by default in APEBench and how they can be classified.

| Name | 1. | 2. | 3. | 4. | 5. |
|------|----|----|----|----|----|
| mean_MAE | (a) | (b) | (a) | (a) | No |
| mean_nMAE | (a) | (b) | (a) | (b) | No |
| mean_sMAE | (a) | (b) | (a) | (c) | No |
| mean_MSE | (a) | (a) | (a) | (a) | No |
| mean_nMSE | (a) | (a) | (a) | (b) | No |
| mean_sMSE | (a) | (a) | (a) | (c) | No |
| mean_RMSE | (a) | (c) | (a) | (a) | No |
| mean_nRMSE | (a) | (c) | (a) | (b) | No |
| mean_sRMSE | (a) | (c) | (a) | (c) | No |
| mean_fourier_MAE | (b) | (b) | (a) | (a) | optional: (a) & (b) |
| mean_fourier_nMAE | (b) | (b) | (a) | (b) | optional: (a) & (b) |
| mean_fourier_MSE | (b) | (a) | (a) | (a) | optional: (a) & (b) |
| mean_fourier_nMSE | (b) | (a) | (a) | (b) | optional: (a) & (b) |
| mean_fourier_RMSE | (b) | (c) | (a) | (a) | optional: (a) & (b) |
| mean_fourier_nRMSE | (b) | (c) | (a) | (b) | optional: (a) & (b) |
| mean_H1_MAE | (b) | (b) | (a) | (a) | (a), optional: (b) |
| mean_H1_nMAE | (b) | (b) | (a) | (b) | (a), optional: (b) |
| mean_H1_MSE | (b) | (a) | (a) | (a) | (a), optional: (b) |
| mean_H1_nMSE | (b) | (a) | (a) | (b) | (a), optional: (b) |
| mean_H1_RMSE | (b) | (c) | (a) | (a) | (a), optional: (b) |
| mean_H1_nRMSE | (b) | (c) | (a) | (b) | (a), optional: (b) |
| mean_correlation | (a) | N/A | (b) | (a) | No |

   (a) Taking Derivatives

   (b) Extracting certain frequency ranges

An overview of metric functions provided by APEBench and their classifications are shown in Table 4.

**Consistency with Parseval's Theorem and Function Norms**  According to Parseval's theorem, the mean_<X>MSE metric in APEBench is guaranteed to be equivalent to mean_fourier_<X>MSE, provided that the Fourier-based variant (mean_fourier_<X>MSE) does not introduce any modifications to the spectrum, such as selecting specific frequency ranges or applying derivative operations. This equivalence similarly applies to the RMSE metrics, meaning mean_<X>RMSE and mean_fourier_<X>RMSE will yield identical values under these conditions.

However, Parseval's identity does *not* hold for metrics such as mean_<X>MAE and mean_fourier_<X>MAE. This discrepancy arises because Parseval's theorem is only applicable to metrics that consistently relate to the $L^2(\Omega)$ function norm. Consequently, absolute-error-based metrics like MAE, which do not involve squaring, deviate from Parseval's conditions.

This limitation extends to Sobolev-inspired metrics, specifically the mean_H1_<X>MAE metric. Although it is derived from a Sobolev space perspective, it relies on Fourier aggregation, meaning it is not directly consistent with the $H^1(\Omega)$ function norm. The Fourier aggregation used internally in mean_H1_<X>MAE prevents full alignment with the $H^1$ norm, as Parseval's identity does not apply outside of $L^2$-aligned norms.

In cases where derivatives are applied within the Sobolev-based losses, all "gradient" directions are summed. Sobolev-based losses such as those labeled with H1 contain contributions from both the function values and their first derivatives, which highlights errors in higher frequencies/smaller scales.

**About metrics used in this paper**  We focused on the nRMSE metric in this paper, as it provides intuitive and comparable evaluations. Additionally, for metric calculations, we applied a channel

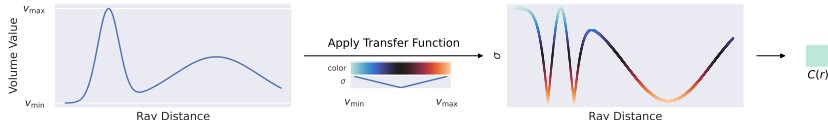

Figure 11: Volume rendering with a transfer function. For every pixel, a ray is marched through the volume, and the pixel color $C(r)$ is determined according to Equation 29.

averaging approach to ensure metrics are always in the same range, as opposed to a channel summing approach, which is the default in the release version of APEBench.

## G An Interactive Transient 3D Volume Renderer for Simulation Trajectories in Python

The seamless exploration and visualization of 3D time-varying data has shown to be difficult with existing tools. As part of the benchmark, we publish a real-time interactive 4D volume rendering tool for seamless visualization of volumes within a Python environment.

The tool uses a user-defined transfer function and volume rendering for visualization. For each pixel, a ray $r$ is marched through the volume. $N$ samples with distance $\delta$ are taken along the ray, and the transfer function is used to map each sample to a density $\sigma$ and color $c$. The final pixel color is computed using the volume rendering equation:

$$C(r) = \sum_{i=1}^{N} T_i(1 - \exp(-\sigma_i\delta_i))c_i \tag{29}$$

$$T_i = \exp(-\sum_{j=1}^{i-1} \sigma_j\delta_j) \tag{30}$$

Figure 11 shows the rendering process for a single pixel.

The visualizer is written in Rust and uses the WebGPU Graphics API, allowing it to run in a modern browser. This enables embedding it in interactive Python environments such as Jupyter Notebooks or Visual Studio Code. Figure 12 shows a screenshot of the tool.

## H Experimental Details

### H.1 General Details

**Initial conditions** Most experiments use an initial condition according to a *truncated Fourier series* (Bar-Sinai et al., 2019). In one dimension, it can be described by

$$u(t = 0, x) = o + \sum_{k=1}^{K} a_k \sin(k\frac{2\pi}{L}x) + b_k \cos(\frac{2\pi}{L}x),$$

with the offset $o$ and coefficients for sine modes $a_k$ and cosine modes $b_k$ drawn according to

$$o \propto \mathcal{U}(-0.5, 0.5),$$
$$a_k \propto \mathcal{U}(-1, 1),$$
$$b_k \propto \mathcal{U}(-1, 1).$$

Though most experiments have $o = 0$. Additionally, we limit the absolute magnitude of the discrete initial state to 1. In higher dimensions, the initial condition is found similarly but with the combination of all possible modes up to cutoff $K$.

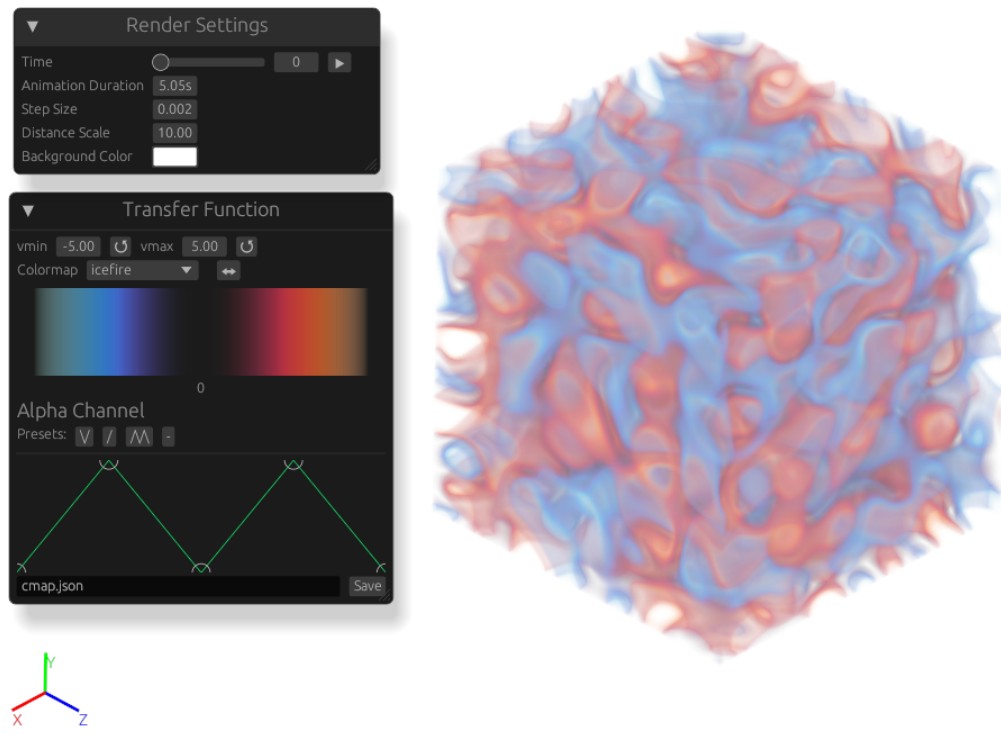

Figure 12: Screenshot of our volume visualization tool. The tool allows for real-time rendering of time-varying data with interactive transfer function editing.

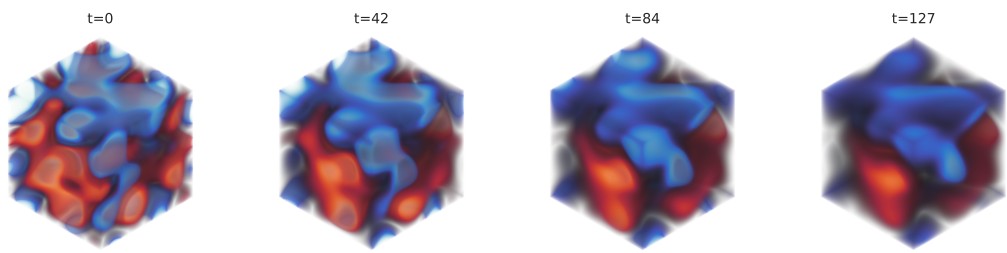

Figure 13: Visualization of the time evolution of the first channel of a 3D Burgers dynamic.

**Physics and Numerics Setup** Unless otherwise stated, we employ $N = 160$ degrees of freedom per spatial dimension. The dynamics are characterized by their combination of linear difficulties $\gamma_s$ and nonlinear difficulties $\delta_s$, when applicable. For certain cases that deviate from this framework, such as Navier-Stokes examples (adjusted via Reynolds number) and reaction-diffusion scenarios (often also with non-standard initial conditions), the domain extent $L$, time step size $\Delta t$, and relevant constitutive parameters are directly specified. Moreover, linear scenarios in higher dimensions that involve spatial mixing are also handled in this manner.

Across all experiments, we utilize the second-order ETDRK method (ETDRK2, section B.1), as we observed a favorable cost-accuracy trade-off in single-precision floating-point calculations compared to higher-order methods. For computing the complex-valued coefficients, we set the circle radius to 1.0 and choose 16 points on the complex unit circle (Kassam and Trefethen, 2005). The reference stepper typically does not employ substepping unless explicitly mentioned.

**Train Data Trajectories** If not specified otherwise, we draw 50 initial conditions for training and discretize them on the given resolution. The ETDRK solver is then used to autoregressively roll them out for 50 time steps. Effectively, this results in an array of shape

(`SAMPLES=50,TIME_STEPS=51,CHANNELS,...`) with the ellipsis denoting an arbitrary number of spatial axes. Within these 50 trajectories, we randomly sample windows of size `ROLLOUT+1` for stochastic minibatching.

**Test Data Trajectories**   We draw 30 initial conditions from the same distribution as for the training dataset but with a different random seed. The solver suite produces trajectories of 200 time steps.

**Neural Architectures**   We chose our architectures to have $\approx$ 30k parameters in 1D, $\approx$ 60k in 2D, and $\approx$ 200k in 3D. In section I.4, we ablate parameter sizes of the architectures. All convolutions use `"SAME"` periodic/circular padding and a kernel size of three. We do not consider learning the boundary condition. All architectures are implemented agnostic to the spatial dimension.

We employ the following architectural constructors:

- `Conv;WIDTH;DEPTH;ACTIVATION`: A feedforward convolutional network with `DEPTH` hidden layers of `WIDTH` size. Each layer transition except for the last uses `ACTIVATION`. The effective receptive field is `DEPTH` $+ 1$.

- `Res;WIDTH;BLOCKS;ACTIVATION`: A classical/legacy ResNet with post-activation and no normalization scheme. Each residual block has two convolutions and operates at `WIDTH` channel size. The `ACTIVATION` follows each of the convolutions in the residual block. There are `BLOCKS` number of residual blocks. Lifting and projection are point-wise linear convolutions (=1x1 convs).

- `UNet;WIDTH;LEVELS;ACTIVATION`: A classical UNet using double convolution blocks with group activation in-between (number of groups is set to one). `WIDTH` describes the hidden layer's size on the highest resolution level. `LEVELS` indicates the number of times the spatial resolution is halved by a factor of two while the channel count doubles. Skip connections exist between the encoder and decoder part of the network.

- `Dil;DIL-FACTOR;WIDTH;BLOCKS;ACTIVATION`: Similar to the classical post-activation ResNet but uses a series of stacked convolutions of different dilation rates. Each convolution is followed by a group normalization (number of groups is set to one) and the `ACTIVATION`. `DIL-FACTOR` of 1 refers to one convolution of dilation rate 1. If it is set to 2, this refers to three convolutions of rates [1, 2, 1]. If it is 3, then this is [1, 2, 4, 2, 1], etc.

- `FNO;MODES;WIDTH;BLOCKS;ACTIVATION`: A vanilla FNO using spectral convolutions with `MODES` equally across all spatial dimensions. Each block operates at `WIDTH` channel size and has one spectral convolution with a point-wise linear bypass. The activation is applied to the sum of spectral convolution and bypass result. There are `BLOCKS` total blocks. Lifting and projection are point-wise linear (=1x1) convolutions.

The concrete architectures, their parameter counts, and effective receptive fields are listed in Table 5.

**Seed Statistics**   We report statistics over various random seeds. We fix the random seeds for train and test data generation for each experiment. Then, we use this one set of data to train an ensemble of networks. Each network uses the same initialization routines (following the defaults in Equinox and a reasonable default for the spectral convolution in FNOs) but a different random key. This random key also modifies the stochastic mini-batching to which the networks are subject during training. For one-dimensional (at most realistic resolutions) and two-dimensional problems (with low resolution $N \leq 50^2$), seed statistics can be obtained virtually for free since one network training does not fully utilize an entire GPU. We run the seed statistics sequentially for three-dimensional experiments and the two-dimensional problems at $N = 160^2$. We use 50 seeds for experiments in 1D, and 20 seeds for experiments in 2D and 3D. Statistics are aggregated using the *median* and display the corresponding *50 % inter-quantile range (IQR)* (from the 25 percentile to the 75 percentile). We chose the median aggregator to reduce the influence of seed outliers. Similarly, the 50% IQR is less susceptible to outliers.

**Training and Optimization**   If not specified otherwise, we use the Adam optimizer (Kingma and Ba, 2015) with a warmup cosine decay learning rate scheduling (Loshchilov and Hutter, 2017). This scheduler was also found beneficial in recent physics-based deep learning publications (Tran et al., 2023; Lam et al., 2022). The default optimization duration is $10'000$ update steps. The number of

Table 5: Concrete architectures used across the dimensions. Receptive Field must be understood per direction.

| Dimension | Descriptor | # Params | Rec. Field |
|:---:|:---:|:---:|:---:|
| 1 | `Conv;34;10;relu` | 31'757 | 11 |
| | `Res;26;8;relu` | 32'943 | 16 |
| | `UNet;12;2;relu` | 27'193 | 29 |
| | `Dil;2;32;2;relu` | 31'777 | 20 |
| | `FNO;12;18;4;gelu` | 32'527 | inf |
| 2 | `Conv;26;11;relu` | 61'595 | 12 |
| | `Res;26;5;relu` | 61'179 | 10 |
| | `UNet;10;2;relu` | 55'661 | 29 |
| | `Dil;2;26;2;relu` | 61'699 | 20 |
| | `FNO;10;6;4;gelu` | 57'787 | inf |
| 3 | `Conv;26;12;relu` | 202'489 | 13 |
| | `Res;25;6;relu` | 202'876 | 12 |
| | `UNet;11;2;relu` | 200'322 | 29 |
| | `Dil;2;27;2;relu` | 192'722 | 20 |
| | `FNO;5;7;4;gelu` | 196'246 | inf |

effective epochs depends on the number of initial conditions, the train temporal horizon, and the unrolled training length. If not specified otherwise, the batch size is set to 20. This usually reduces to $50 \sim 100$ epochs of $100 \sim 200$ minibatches. The first $2'000$ update steps are a linear warmup of the learning rate from $0.0$ to $10^{-3}$. Afterward, it decays according to a cosine schedule, reaching $0.0$ at the last, the $10'000$-th iteration. We found a decay all the way to zero helpful in forcing the networks into convergence and reducing the impact of the last minibatch variation on the deduced metric.

**Evaluation Metrics**  We evaluate the performance of the emulators using the *mean normalized root mean squared error* (mean-nRMSE, in the main text just called `nRMSE`)

$$L_{\text{nRMSE}} = \frac{1}{M} \sum_{j=1}^{M} \sqrt{\frac{\sum_{i=1}^{N} (\hat{u}_{j,i} - u_{j,i})^2}{\sum_{i=1}^{N} (u_{j,i})^2}} \tag{31}$$

over $M$ samples and $N$ degrees of freedom. This metric adjusts for differences in scale with regard to the reference state $u_h$. A value of 1 indicates that the magnitude of the error is the same as the magnitude of the reference, i.e., the predicted state $\hat{u}_h$ is completely different from the reference. This normalization is helpful for error rollouts over time for dynamics that have states changing in magnitude, e.g., decaying phenomena like the Burgers equation. Additionally, this allows for a clear comparison across dynamics, which can also have states that differ in magnitude.

We use an upper index $[t]$ to denote the loss after $t$ time steps $L_{\text{nRMSE}}^{[t]}$. The loss at $[t] = [0]$ is zero since autoregressive prediction trajectory and reference start at the same initial conditions. We are interested in how the error develops over the time steps since this reveals insights into the emulators' temporal generalization capabilities. For aggregating the metric, we choose an upper index $T$ (by default 100) and use the geometric mean (gmean-mean-nRMSE, in the main text just called `Aggregated nRMSE` or `Agg. nRMSE`)

$$L_{\text{agg-nRMSE}} = \exp\left( \frac{1}{T} \sum_{t=1}^{T} \log(L_{\text{nRMSE}}^{[t]}) \right). \tag{32}$$

Employing the geometric mean reduces the need to handcraft upper limits for temporal aggregation in case error metrics go beyond the value 1.

**Hardware & Runtime**  We conducted our experiments on a cluster of eight Nvidia RTX 2080 Ti with 12GB of video memory each. We used the collection of GPUs to distribute runs with different initialization seeds but not to distribute a single network over multiple GPUs. Table 6 displays the cost of all individual experiments. The total runtime is $\approx 900$ GPU-hours. Under an ideal load distribution, this equals $\approx 5$ days of full runtime on the cluster we used.

Table 6: Computational cost in terms of runtime on an Nvidia RTX 2080 Ti for all experiments presented in this work.

| Experiment | Seeds per Run | Different Runs | Total Seeds | Time per Run | Total Time |
|---|---|---|---|---|---|
| Motivational, section 2 | N/A | 1 | N/A | <1min | <1min |
| Bridging, vary $\gamma_1$, section 2 | 10 | 5 | 50 | 1h | 5h |
| Bridging, vary rollout, section 2 | 10 | 5 | 50 | 1h | 5h |
| Diverted Chain, section 5.2 | 10 | 5 | 50 | 1h | 5h |
| Advection Correction, section 5.3 | 1 | 20 | 20 | 2h | 40h |
| Broad Comparison 1D, section 5.4 | 10 | 5 | 50 | 2h | 10h |
| Broad Comparison 2D, section 5.4 | 1 | 20 | 20 | 4h | 80h |
| 2D KS reduced res, section 5.4 | 10 | 2 | 20 | 20min | 40min |
| Broad Comparison 3D, section 5.4 | 1 | 20 | 20 | 15h | 300h |
| Sum, main experiments | - | 83 | - | - | $\approx 450$h |
| Ablation, Unrolled Training, section I.1 | 10 | 5 | 50 | 3h | 15h |
| Ablation, Opt Config, section I.2 | 10 | 5 | 50 | 20h | 100h |
| Ablation, Train Size, section I.3 | 10 | 5 | 50 | 45h | 225h |
| Ablation, Parameter Scaling, section I.4 | 10 | 5 | 50 | 20h | 100h |
| Sum, ablation experiments | - | 20 | - | - | $\approx 450$h |
| Sum, all experiments | - | 103 | - | - | $\approx 900$h |

## H.2 Motivational Experiment

This subsection describes the details of the experiment in section 2.

The difficulty of the problem is set to $\gamma_1 = 0.75$. This is in combination with $N = 30$ degrees of freedom and $D = 1$ spatial dimensions. The initial condition distribution follows a truncated Fourier series with cutoff $K = 5$, zero mean, and max one absolute. For training, we draw five initial conditions and integrate them for 200 time steps with the analytical solver. The EDTRK solver suite gives this analytical stepper since it can integrate any linear PDE with a bandlimited initial condition exactly.

The optimization is performed over the full batch of all samples (across trajectories and all possible windows within each trajectory) with a Newton optimizer. We initialize the optimization with the FOU stencil. For long unrolled training (beyond what we display in this work), we observed convergence problems, which needed us to initialize the optimization for $T + 1$ unrolled steps with the minimizer of $T$ unrolled steps. This also indicates that training with unrolled steps is a tougher optimization problem, which might need such curriculum strategies. The Newton method is run until convergence to double floating machine precision ($\approx 10^{-16}$); typically achieved within ten iterations.

The found stencils are measured against the analytical solution according to the mean-nRMSE error (31) for 50 new initial conditions drawn from the same distribution as for the training dataset but compared over 200 time steps. The FOU stencil is measured similarly. In Table 7, we display the numeric error values at relevant time steps $[t]$.

Table 8 displays the optimizers in parameter space. We also present the found stencils for three variations to highlight that the data-driven optimization problem is non-trivial and sensitive to the concrete setup:

- **Div**: Uses unrolling with diverted chain employing the analytical solver as a differentiable fine stepper.

- **More Points**: Uses a resolution of $N = 60$ instead of $N = 30$

- **More Modes**: Uses a cutoff of $K = 10$ instead of $K = 5$

Table 7: Numeric Values for the linear convolution learning experiment of the 1D advection equation. "-" indicates that the value is beyond 1.0. This table also displays the error up to time step 200 which was not shown in the main part of the paper. We see that beyond a certain point, the FOU stencil again becomes superior because it is consistent with the advection equation. Note that after step 30, each row makes a bigger step.

| | | **Mean nRMSE** ↓ | | | | | |
| | Unrolled Steps | 1 | 2 | 5 | 10 | 20 | 50 |
| Time Step | FOU | | | | | | |
|---|---|---|---|---|---|---|---|
| 1 | 0.055 | **0.036** | **0.036** | **0.036** | 0.037 | 0.040 | 0.046 |
| 2 | 0.105 | **0.071** | **0.071** | **0.071** | 0.072 | 0.078 | 0.088 |
| 3 | 0.151 | 0.106 | 0.106 | **0.105** | 0.106 | 0.113 | 0.127 |
| 4 | 0.194 | 0.141 | 0.140 | **0.138** | 0.139 | 0.146 | 0.164 |
| 5 | 0.233 | 0.175 | 0.174 | **0.170** | **0.170** | 0.177 | 0.198 |
| 6 | 0.270 | 0.210 | 0.207 | 0.202 | **0.200** | 0.207 | 0.229 |
| 7 | 0.303 | 0.244 | 0.240 | 0.233 | **0.229** | 0.235 | 0.259 |
| 8 | 0.334 | 0.279 | 0.274 | 0.263 | **0.257** | 0.261 | 0.286 |
| 9 | 0.363 | 0.313 | 0.307 | 0.294 | **0.284** | 0.287 | 0.312 |
| 10 | 0.389 | 0.348 | 0.340 | 0.324 | **0.310** | **0.310** | 0.336 |
| 11 | 0.414 | 0.384 | 0.374 | 0.353 | 0.336 | **0.333** | 0.358 |
| 12 | 0.437 | 0.420 | 0.408 | 0.383 | 0.362 | **0.355** | 0.379 |
| 13 | 0.458 | 0.456 | 0.443 | 0.413 | 0.386 | **0.376** | 0.398 |
| 14 | 0.478 | 0.493 | 0.478 | 0.443 | 0.411 | **0.396** | 0.417 |
| 15 | 0.497 | 0.532 | 0.514 | 0.474 | 0.435 | **0.415** | 0.434 |
| 16 | 0.514 | 0.571 | 0.550 | 0.504 | 0.460 | **0.434** | 0.450 |
| 17 | 0.530 | 0.611 | 0.588 | 0.535 | 0.484 | **0.452** | 0.466 |
| 18 | 0.545 | 0.652 | 0.626 | 0.566 | 0.508 | **0.469** | 0.480 |
| 19 | 0.560 | 0.694 | 0.665 | 0.598 | 0.532 | **0.486** | 0.494 |
| 20 | 0.573 | 0.738 | 0.706 | 0.631 | 0.556 | **0.503** | 0.507 |
| 21 | 0.586 | 0.783 | 0.747 | 0.664 | 0.580 | **0.519** | 0.519 |
| 22 | 0.597 | 0.830 | 0.790 | 0.698 | 0.604 | 0.535 | **0.531** |
| 23 | 0.609 | 0.878 | 0.834 | 0.733 | 0.629 | 0.551 | **0.542** |
| 24 | 0.619 | 0.928 | 0.880 | 0.769 | 0.654 | 0.566 | **0.552** |
| 25 | 0.629 | 0.980 | 0.927 | 0.805 | 0.679 | 0.582 | **0.562** |
| 26 | 0.638 | - | 0.976 | 0.843 | 0.705 | 0.597 | **0.572** |
| 27 | 0.647 | - | - | 0.882 | 0.731 | 0.612 | **0.581** |
| 28 | 0.656 | - | - | 0.922 | 0.758 | 0.627 | **0.590** |
| 29 | 0.664 | - | - | 0.963 | 0.785 | 0.642 | **0.599** |
| 30 | 0.671 | - | - | - | 0.813 | 0.657 | **0.607** |
| 40 | 0.730 | - | - | - | - | 0.812 | **0.676** |
| 50 | 0.770 | - | - | - | - | 0.992 | **0.731** |
| 60 | 0.798 | - | - | - | - | - | **0.782** |
| 70 | **0.820** | - | - | - | - | - | 0.834 |
| 100 | **0.862** | - | - | - | - | - | - |
| 200 | **0.922** | - | - | - | - | - | - |

Table 8: Optimizers of the stencil learning problem (up to double floating precision accuracy with a Newton optimizer). The FOU stencil is $[0.25, 0.75]$.

| Location | Unrolled Steps | Default | Div | >Points | >Modes |
|----------|----------------|---------|--------|---------|--------|
| center | 1 | 0.2668 | 0.2668 | 0.2540 | 0.2856 |
| | 2 | 0.2662 | 0.2661 | 0.2539 | 0.2900 |
| | 5 | 0.2648 | 0.2643 | 0.2538 | 0.2902 |
| | 10 | 0.2629 | 0.2624 | 0.2537 | 0.2787 |
| | 20 | 0.2605 | 0.2601 | 0.2535 | 0.2682 |
| | 50 | 0.2568 | 0.2571 | 0.2529 | 0.2594 |
| right | 1 | 0.7797 | 0.7797 | 0.7570 | 0.8872 |
| | 2 | 0.7785 | 0.7782 | 0.7570 | 0.8489 |
| | 5 | 0.7752 | 0.7741 | 0.7568 | 0.7952 |
| | 10 | 0.7706 | 0.7686 | 0.7565 | 0.7745 |
| | 20 | 0.7641 | 0.7624 | 0.7559 | 0.7639 |
| | 50 | 0.7568 | 0.7565 | 0.7545 | 0.7581 |

## H.3 Bridging Experiment

This subsection details the settings of the experiments in section 5.1.

We use the default configuration of the `diff_adv` scenario in 1D but adapted the difficulty such that $\gamma_1 \in \{0.5, 2.5, 10.5\}$. The networks are the default configuration for one dimension. Only the feedforward convolutional network was modified to a different `DEPTH`. With the depth set to 10, it corresponds again to the default configuration.

In Table 9, we display the error metrics at specific time steps across architectures and difficulty. Table 10 displays the time step errors for the ResNet when being trained with more unrolled steps.

## H.4 Diverted Chain Experiment

This subsection details the settings of the experiments in section 5.2.

Each of the three nonlinear scenarios uses the default setup listed in Table 2. We use the default configuration for the ResNet in 1D as denoted in Table 5. In Table 11, we list errors at time steps 1 and 100 for all three training configurations in the median over 50 seeds and the limits of the 50% IQR.

## H.5 Task and Rollout Training Experiment

This subsection details the settings of the experiments in section 5.3.

This experiment uses the `diff_adv` scenario with `num_spatial_dims=2`. To create the three variations, we fixed the scenario's `advection_gamma=10.5` and varied the `coarse_proportion` in $\{0.0, 0.1, 0.5\}$. The ResNet and the FNO are in their default configuration for 2D as displayed in Table 5. In Table 12, we display the geometric mean (see Eq. 32) of the test error rollout over the first 100 time steps.

## H.6 Broad Comparison Experiment

This subsection details the settings of the experiments in section 5.4.

All emulators are trained for a pure prediction task using a one-step supervised configuration. We measure performance in terms of the geometric mean of the test rollout over 100 time steps. The resolution for the 3D problems is reduced to $N = 32^3$, and we reduce the number of trajectories in the test dataset from 30 to 10 to ensure that the experiments worked on 12GB GPU memory. For the one-dimensional problem, we produced 50 seeds. For 2D and 3D, we used 20 seeds.

Scenarios are under their default setting (see table 2), except for the below:

Table 9: Numeric Values of the bridging experiment. For brevity, only the median out of 50 seeds is presented. A "-" indicates that the value is above 1.0.

| | | | | | Mean nRMSE ↓ | | | | |
|---|---|---|---|---|---|---|---|---|---|
| $\gamma_1$ | Arch / Time Step | Conv 0 | Conv 1 | Conv 2 | Conv 10 | Res | UNet | Dil | FNO |
| | 1 | 0.005 | 0.004 | **0.001** | **0.001** | **0.001** | 0.005 | 0.004 | **0.001** |
| | 2 | 0.010 | 0.007 | 0.002 | 0.002 | **0.001** | 0.007 | 0.007 | 0.002 |
| | 3 | 0.015 | 0.010 | 0.003 | 0.003 | **0.002** | 0.010 | 0.009 | 0.003 |
| | 4 | 0.020 | 0.013 | 0.004 | 0.004 | **0.002** | 0.012 | 0.011 | 0.004 |
| 0.5 | 5 | 0.025 | 0.016 | 0.005 | 0.005 | **0.003** | 0.014 | 0.013 | 0.005 |
| | 10 | 0.050 | 0.030 | 0.009 | 0.010 | **0.005** | 0.024 | 0.024 | 0.010 |
| | 20 | 0.099 | 0.058 | 0.017 | 0.018 | **0.010** | 0.042 | 0.044 | 0.021 |
| | 30 | 0.146 | 0.084 | 0.025 | 0.026 | **0.015** | 0.058 | 0.063 | 0.030 |
| | 40 | 0.193 | 0.108 | 0.032 | 0.033 | **0.019** | 0.075 | 0.079 | 0.040 |
| | 50 | 0.240 | 0.129 | 0.040 | 0.039 | **0.023** | 0.090 | 0.095 | 0.049 |
| | 1 | 0.044 | 0.005 | 0.002 | 0.002 | **0.001** | 0.005 | 0.005 | **0.001** |
| | 2 | 0.088 | 0.011 | 0.003 | 0.003 | **0.002** | 0.008 | 0.007 | **0.002** |
| | 3 | 0.133 | 0.019 | 0.004 | 0.005 | **0.003** | 0.010 | 0.009 | **0.003** |
| | 4 | 0.178 | 0.033 | 0.005 | 0.006 | **0.004** | 0.013 | 0.012 | **0.004** |
| 2.5 | 5 | 0.223 | 0.058 | 0.007 | 0.007 | 0.006 | 0.015 | 0.014 | **0.005** |
| | 10 | 0.463 | - | 0.013 | 0.014 | 0.011 | 0.027 | 0.026 | **0.009** |
| | 20 | - | - | 0.034 | 0.027 | 0.020 | 0.049 | 0.046 | **0.018** |
| | 30 | - | - | 0.075 | 0.039 | 0.030 | 0.068 | 0.066 | **0.027** |
| | 40 | - | - | 0.152 | 0.051 | 0.039 | 0.086 | 0.084 | **0.036** |
| | 50 | - | - | 0.323 | 0.062 | 0.048 | 0.103 | 0.100 | **0.044** |
| | 1 | 0.571 | 0.221 | 0.054 | 0.004 | 0.004 | 0.007 | 0.007 | **0.001** |
| | 2 | 0.936 | 0.524 | 0.833 | 0.009 | 0.008 | 0.011 | 0.011 | **0.002** |
| | 3 | - | - | - | 0.025 | 0.015 | 0.014 | 0.015 | **0.003** |
| | 4 | - | - | - | 0.081 | 0.032 | 0.018 | 0.019 | **0.004** |
| 10.5 | 5 | - | - | - | 0.270 | 0.071 | 0.021 | 0.024 | **0.005** |
| | 10 | - | - | - | - | - | 0.039 | 0.049 | **0.010** |
| | 20 | - | - | - | - | - | 0.070 | 0.113 | **0.021** |
| | 30 | - | - | - | - | - | 0.101 | 0.247 | **0.031** |
| | 40 | - | - | - | - | - | 0.130 | 0.514 | **0.041** |
| | 50 | - | - | - | - | - | 0.157 | 0.901 | **0.051** |

- Both 2D and 3D Gray Scott do not use the random truncated Fourier series as initial condition distribution. Instead, they employ one Gaussian blob per channel (regardless of spatial dimension, Gray Scott always has two channels for the two species), with the second channel using a one-complement.

- In 2D, we use the `phy_gs_type` scenario with all default configurations; in 3D, we just use `phy_gs`, which has effectively stronger diffusion (due to the smaller domain), better suited for the reduced resolution. The pattern type in both is effectively `theta`.

- The KS scenario is run using both the default resolution of $N = 160^2$ and a reduced resolution of $N = 32^2$. Note that the difficulty interface adapts the complexity of the dynamics based on resolution and dimensionality.

We display the spectra for some of the dynamics in figure 14.

# I   Ablation Studies

In this section, we ablate choices made for the main experiments in this publication. We found them to be fair settings that also allow for compute-efficient broad comparison across the axes supported by the benchmark. We stress that APEBench is flexible and allows fine-grain control over these variables but also comes with reasonable defaults.

Table 10: Numeric results of advection learning at the highest difficulty when increasing the unrolled steps during training for the ResNet architecture. A "-" indicates that the value went beyond 1.0. We also showcase temporal results up to 200 (the plot in the main text was limited to a horizon of 25) and training with up to 15 steps of unrolling.

| | **Mean nRMSE** $\downarrow$ $(\gamma_1 = 10.5)$ | | | | | |
| Unrolled Steps
Time Step | 1 | 2 | 3 | 5 | 10 | 15 |
| --- | --- | --- | --- | --- | --- | --- |
| 1 | **0.004** | **0.004** | **0.004** | 0.005 | 0.006 | 0.015 |
| 2 | 0.008 | 0.006 | **0.005** | 0.006 | 0.008 | 0.021 |
| 3 | 0.015 | 0.009 | **0.007** | 0.008 | 0.009 | 0.026 |
| 4 | 0.030 | 0.013 | 0.010 | **0.009** | 0.011 | 0.029 |
| 5 | 0.068 | 0.021 | 0.012 | **0.011** | 0.013 | 0.034 |
| 10 | - | 0.242 | 0.037 | **0.021** | 0.022 | 0.050 |
| 15 | - | - | 0.121 | **0.031** | **0.031** | 0.067 |
| 20 | - | - | 0.387 | 0.042 | **0.040** | 0.082 |
| 25 | - | - | 0.760 | 0.052 | **0.049** | 0.099 |
| 30 | - | - | - | 0.066 | **0.058** | 0.114 |
| 40 | - | - | - | 0.097 | **0.076** | 0.143 |
| 50 | - | - | - | 0.144 | **0.093** | 0.169 |
| 75 | - | - | - | 0.463 | **0.140** | 0.234 |
| 100 | - | - | - | - | **0.198** | 0.297 |
| 150 | - | - | - | - | **0.389** | 0.418 |
| 200 | - | - | - | - | 0.732 | **0.559** |

Table 11: Numeric values for experiment of section 5.2, out of 50 seeds.

| | | | **Mean nRMSE** $\downarrow$ | | |
| Dynamics | Time Step | Method | Median | 25% Quantile | 75% Quantile |
| --- | --- | --- | --- | --- | --- |
| Burgers | 1 | one | **0.0016** | 0.0015 | 0.0018 |
| | | sup;05 | 0.0023 | 0.0021 | 0.0026 |
| | | div;05 | 0.0018 | 0.0017 | 0.0020 |
| | 100 | one | 0.0323 | 0.0293 | 0.0351 |
| | | sup;05 | **0.0246** | 0.0216 | 0.0297 |
| | | div;05 | 0.0278 | 0.0240 | 0.0325 |
| KS | 1 | one | 0.0074 | 0.0073 | 0.0075 |
| | | sup;05 | 0.0076 | 0.0075 | 0.0078 |
| | | div;05 | **0.0070** | 0.0068 | 0.0070 |
| | 100 | one | 0.7664 | 0.7507 | 0.8044 |
| | | sup;05 | **0.6631** | 0.6390 | 0.6823 |
| | | div;05 | 0.7485 | 0.7239 | 0.7725 |
| KdV | 1 | one | **0.0036** | 0.0035 | 0.0039 |
| | | sup;05 | 0.0051 | 0.0049 | 0.0054 |
| | | div;05 | 0.0040 | 0.0037 | 0.0046 |
| | 100 | one | 2.3276 | 1.7124 | 3.6859 |
| | | sup;05 | 0.8906 | 0.2435 | 1.4430 |
| | | div;05 | **0.1784** | 0.1500 | 0.2385 |

Table 12: Numerical results for experiment of section 5.3.

| | | | GMean of Mean nRMSE $\downarrow$ | | |
| | | | Median | 25% Quantile | 75% Quantile |
| Task | Network | Unrolled Steps | | | |
|---|---|---|---|---|---|
| $\gamma_1$=10.5, $\tilde{\gamma}_1$=0.0 | `Res;26;5;relu` | 1 | 0.269 | 0.190 | 0.335 |
| | | 5 | 0.108 | 0.102 | 0.129 |
| | `FNO;10;6;4;gelu` | 1 | 0.153 | 0.122 | 0.176 |
| | | 5 | 0.135 | 0.127 | 0.162 |
| $\gamma_1$=10.5, $\tilde{\gamma}_1$=1.05 | `Res;26;5;relu` | 1 | 0.196 | 0.131 | 0.245 |
| | | 5 | 0.100 | 0.095 | 0.120 |
| | `FNO;10;6;4;gelu` | 1 | 0.154 | 0.124 | 0.174 |
| | | 5 | 0.135 | 0.130 | 0.161 |
| $\gamma_1$=10.5, $\tilde{\gamma}_1$=5.25 | `Res;26;5;relu` | 1 | 0.068 | 0.066 | 0.077 |
| | | 5 | 0.066 | 0.060 | 0.073 |
| | `FNO;10;6;4;gelu` | 1 | 0.161 | 0.137 | 0.183 |
| | | 5 | 0.142 | 0.135 | 0.178 |

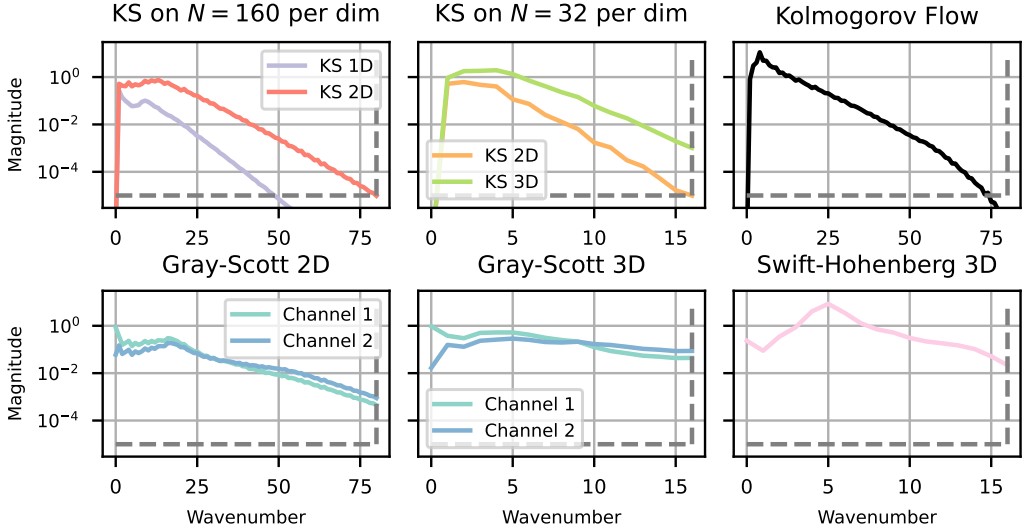

Figure 14: Spectra of the magnitude of Fourier coefficients. For the chaotic problems (Kuramoto-Sivashinsky and Kolmogorov Flow), the spectrum is averaged over all samples and time steps in the test trajectories. In the case of the reaction-diffusion problems (Gray-Scott and Swift-Hohenberg), the spectrum is averaged, excluding the initial states. For problems in higher dimensions ($D \geq 2$) we use a binning approach to compute the spectrum. A magnitude contribution is associated with wavenumber $k$ if its wavenumber norm is within a ring, i.e., $\|\mathbf{k}\|_2 \in [k - \frac{1}{2}, k + \frac{1}{2})$. Based on the resolution per dimension, the Nyquist mode is either at 80 or 16. We consider a problem under-resolved if the Nyquist mode (and the hypothetical modes beyond it) are *not* below the threshold of $10^{-5}$. Under this definition, the KS 3D and all reaction-diffusion problems are under-resolved.

Table 13: Numeric values for the comparison across architectures, dynamics and spatial dimensions. The median over initialization seeds is displayed for the geometric mean of 100 time steps of test rollout. Note that the concrete configuration of the architecture type varies across spatial dimensions; see Table 5 for details.

| | | GMean of Mean nRMSE↓ | | | | |
|---|---|---|---|---|---|---|
| Category | Architecture Scenario | Conv | Res | UNet | Dil | FNO |
| Linear | 1D Dispersion | **0.032** | 0.042 | 0.085 | 0.071 | 0.039 |
| | 2D Anisotropic Diffusion | **0.016** | 0.022 | 0.044 | 0.041 | 0.077 |
| | 3D Unbalanced Advection | 0.308 | **0.134** | 0.183 | 0.205 | 0.895 |
| Nonlinear | 1D Korteweg-de Vries | 0.123 | 0.876 | 0.096 | **0.077** | 0.210 |
| | 1D Kuramoto-Sivashinsky | 0.270 | 0.261 | 0.194 | **0.157** | 0.538 |
| | 2D Kolmogorov Flow | 0.882 | 0.916 | 0.827 | 0.876 | **0.689** |
| Reaction-Diffusion | 2D Gray-Scott | **0.055** | 0.064 | 0.069 | 0.103 | 0.210 |
| | 3D Gray-Scott | **0.052** | 0.070 | 0.081 | 0.165 | 0.668 |
| | 3D Swift-Hohenberg | 0.233 | **0.049** | 0.178 | 0.290 | 0.401 |
| Across Dimensions | 1D Burgers | 0.026 | **0.013** | 0.065 | 0.057 | 0.070 |
| | 2D Burgers | 0.146 | **0.053** | 0.162 | 0.139 | 0.328 |
| | 3D Burgers | 0.627 | **0.146** | 0.215 | 0.786 | 0.287 |
| Across Resolution | 2D KS $N = 160^2$ | 0.218 | **0.200** | 0.268 | 0.257 | 0.908 |
| | 2D KS $N = 32^2$ | 0.474 | 0.450 | 0.331 | 0.299 | **0.272** |
| | 3D KS $N = 32^3$ | 0.566 | 0.436 | **0.369** | 0.623 | 0.505 |

Similarly to the main experiments in section 5, all presented plots and values are aggregated over multiple runs. We used 50 initialization seeds and computed the median. All presented error bars/shaded areas are 50% inter-quantile ranges (IQR).

Table 14: Rollout error values for ResNet emulator at 1D advection with $\gamma_1 = 10.5$ when lower unrolled training steps are compensated with more training time.

| | **Mean nRMSE** $\downarrow$ | | | | | |
|---|---|---|---|---|---|---|
| Unrolled Steps
Time Step | 1 | 2 | 3 | 5 | 10 | 15 |
| 1 | **0.001** | 0.002 | 0.002 | 0.003 | 0.005 | 0.015 |
| 2 | **0.002** | **0.002** | 0.003 | 0.004 | 0.007 | 0.021 |
| 3 | **0.003** | **0.003** | 0.004 | 0.005 | 0.008 | 0.026 |
| 4 | **0.004** | **0.004** | 0.005 | 0.006 | 0.009 | 0.029 |
| 5 | **0.005** | **0.005** | 0.006 | 0.007 | 0.011 | 0.034 |
| 10 | **0.011** | 0.012 | 0.013 | 0.014 | 0.019 | 0.050 |
| 15 | **0.022** | 0.024 | 0.024 | 0.024 | 0.027 | 0.067 |
| 20 | 0.049 | 0.051 | 0.049 | 0.039 | **0.035** | 0.082 |
| 25 | 0.115 | 0.118 | 0.104 | 0.064 | **0.043** | 0.099 |
| 30 | 0.238 | 0.246 | 0.223 | 0.112 | **0.052** | 0.114 |
| 40 | 0.624 | 0.632 | 0.655 | 0.333 | **0.070** | 0.143 |
| 50 | 0.951 | 0.913 | - | 0.685 | **0.088** | 0.169 |
| 75 | - | - | - | - | **0.155** | 0.234 |
| 100 | - | - | - | - | **0.285** | 0.297 |
| 150 | - | - | - | - | 0.799 | **0.418** |
| 200 | - | - | - | - | - | **0.559** |

## I.1 Unrolled Training Ablation

In this section, we revisit the ResNet emulator for 1D advection at difficulty level $\gamma_1 = 10.5$, as previously discussed in Section 5.1. However, our focus now shifts to compensating for the shorter unrolling during training by employing a greater number of update steps of the network (i.e., a larger number of training iterations). This adjustment is motivated by the fact that computational cost, along with memory consumption due to reverse-mode automatic differentiation, scales linearly with the number of unrolled steps in the training phase. By conducting this ablation study, we establish a scenario where each emulator receives roughly equivalent wall clock time on the GPU (approximately 45 minutes for 10 seeds in parallel on an Nvidia RTX 2080 Ti).

Figure 3 presents the results of these experiments. Numeric values are listed in Table 14. Notably, we observe a substantial improvement in the test rollout capabilities of emulators with shorter unrolling. The one-step supervised trained emulator, for instance, demonstrates accuracy over a significantly extended duration. However, the conclusion drawn in the main paper does not change: configurations characterized by more unrolled steps (but fewer total update steps) continue to excel in terms of long-term accuracy.

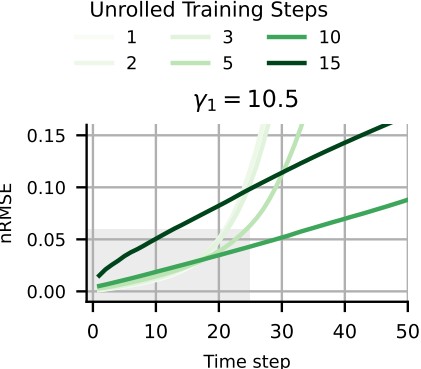

Figure 15: Test rollout performance of ResNet emulators for 1D advection at $\gamma_1 = 10.5$ when each unrolled training uses equal compute time. The gray area indicates the axis limits in the plots of Figure 3. Emulators with shorter unrolling improve, but long unrolling remains superior.

## I.2 Optimization Configuration Ablation

In this section, we investigate the impact of modifying the training duration and peak learning rate while retaining the cosine learning rate decay scheduler with linear warmup ending at one-fifth of the total training duration. Figure 16 displays the results, focusing on the geometric mean over 100 time steps of test rollout. All architectures utilize the default configuration, as do the five scenarios examined: two linear scenarios (advection and diffusion) and three nonlinear scenarios (Burgers, Korteweg-de Vries, and Kuramoto-Sivashinsky) in 1D. The gray line at $10^4$ represents the default

choice employed in all other experiments presented in this work, which is also the default setting across APEBench. The second column maintains the default peak learning rate of 0.001.

Our findings reveal that the highest investigated peak learning rate of 0.01 is excessive for ConvNet, ResNet, and Dilated ResNet architectures, as they fail to demonstrate clear convergence with additional update steps. In contrast, UNet and FNO architectures generally exhibit stable improvement, with the exception of the advection scenario. Across all cases, the default choice of 0.001 appears reasonable, although 0.003 could also be viable. A clear improvement in the geometric mean error (serving as the test error, as it measures temporal generalization on a new set of initial conditions) is observed when networks undergo extended training, as expected. Importantly, the relative ordering among the architectures remains largely consistent. Only the ResNet-type architectures (including Dilated ResNet) appear to benefit more strongly from extended training. Notably, we do not observe signs of overfitting (except at the highest peak learning rate); instead, model performance tends to plateau.

### I.3 Ablation for Size of the Training Dataset

Given that we train on reference trajectories, we ablate the number of initial conditions and the length of the training temporal horizon. It is crucial to note that an excessively short horizon might exclude certain regimes of the physics. For instance, a horizon that is too short for the Burgers equation could omit the shock propagation phase. This is particularly noteworthy because emulators must learn to handle such situations without explicitly encountering them in the training data, as evidenced by the geometric mean aggregation over 100 time steps in our test rollout. Figure 17 illustrates our results, with the gray dashed line indicating the configuration used in all other experiments within this work.

Across all architectures, there is a consistent performance improvement up to a certain number of training samples, beyond which gains become minor but still noticeable. Interestingly, the three classes of neural architectures exhibit distinct behaviors. Local convolutional architectures (ConvNet and ResNet) demonstrate remarkable parameter efficiency, converging with as few as five training samples. These are followed by global convolutional architectures and, finally, the FNO (representing pseudo-spectral architectures).

In the Korteweg-de Vries scenario, the ResNet underperforms notably. Consistent with the findings of the optimization configuration ablation in Section I.2, we consider this an intriguing failure mode for the ResNet, though its root cause remains unclear. However, we observe that unrolled training substantially enhances its performance, as detailed in Section 5.2.

The impact of extended temporal horizons varies depending on the dynamics under investigation. As previously mentioned, dynamics with multiple stages, such as Burgers and Korteweg-de Vries (which must first develop their characteristic spectra), exhibit the most significant improvement in emulator performance with longer horizons. Conversely, the Kuramoto-Sivashinsky equation begins within the chaotic attractor, rendering an extended temporal horizon effectively equivalent to additional samples, as evidenced by the faster convergence of the curves. Notably, all curves converge to approximately the same level. For the advection problem, different temporal horizons yield almost no difference, whereas diffusion displays a clear trend, though less pronounced than for Burgers and KdV. This is attributed to emulators encountering later stages of the dynamics with longer horizons, where low-magnitude states are mapped to even lower magnitudes.

In conclusion, we find that the default choice of 50 initial condition samples with a training temporal horizon of 50 strikes a reasonable balance. Crucially, it does not alter the relative performance ranking of different architectures, enabling fair comparisons.

### I.4 Parameter Scaling Ablation

In this section, we expand the parameter space of the neural emulator by increasing the number of hidden channels. Importantly, we refrain from altering settings that could influence the receptive field, ensuring that each ablated network configuration retains the default receptive field as outlined in Table 5.

Figure 18 presents the test error, quantified as the geometric mean over 100 rollout steps. As anticipated and consistent with observations by List et al. (2024), architectures demonstrate improved performance with increased parameter counts. However, the specific convergence rate appears to be

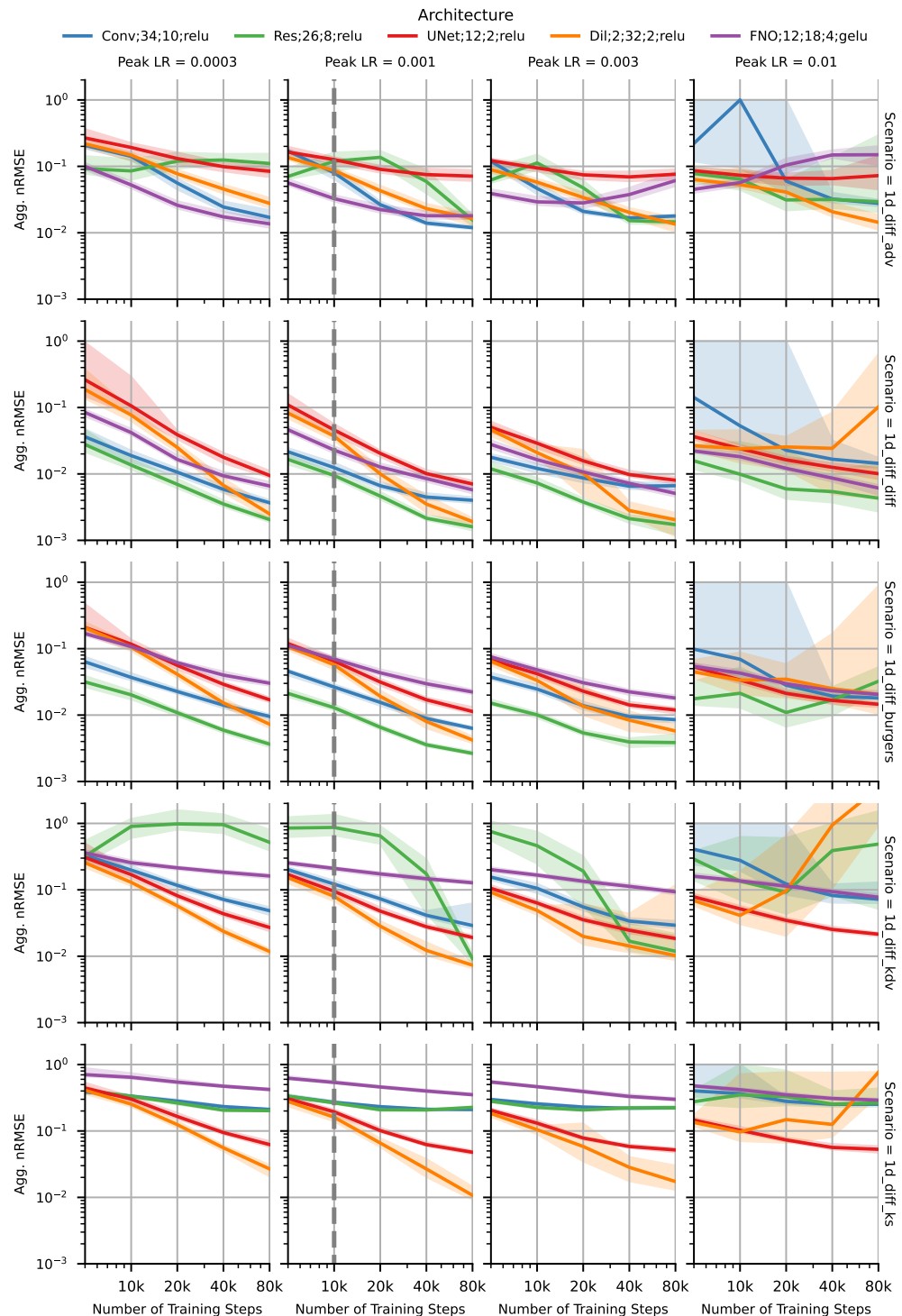

Figure 16: Ablation study of optimization configuration, examining the peak learning rate for the cosine decay scheduler and the total number of update steps. This study encompasses all major architecture classes and five 1D scenarios, displaying the geometric mean over 100 time steps of test rollout. Each row represents a different scenario, each column a different peak learning rate, with the x-axis indicating the total number of update steps. Notably, networks consistently improve in test accuracy with extended training time, sometimes exhibiting stronger convergence for ResNet-like architectures (ResNet and Dilated ResNet). The highest investigated peak learning rate of 0.01 appears to induce divergence in some architectures.

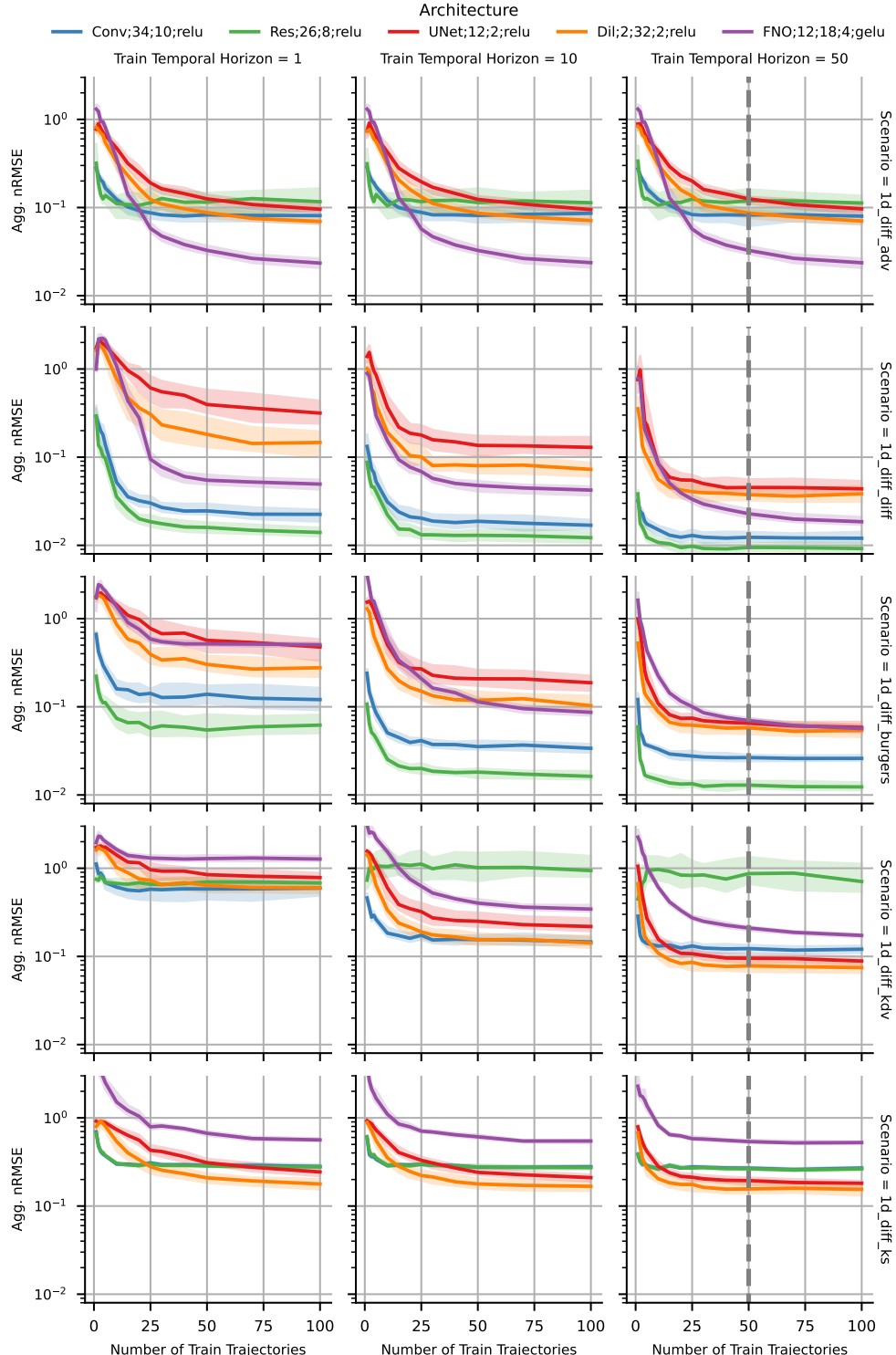

Figure 17: Ablation study of dataset size, examining both the temporal horizon of physics captured in the dataset and the number of samples (trajectories). Each row corresponds to one of the five 1D scenarios. For dynamics with multiple stages (e.g., Burgers and KdV with evolving spectra, decaying diffusion), a longer horizon proves highly beneficial. This effect is less pronounced for the conservative advection problem and the Kuramoto-Sivashinsky equation, which begins within the chaotic attractor. Notably, the convergence behavior over the number of samples is qualitatively similar across architectures, with local convolutional architectures demonstrating the highest sample efficiency.

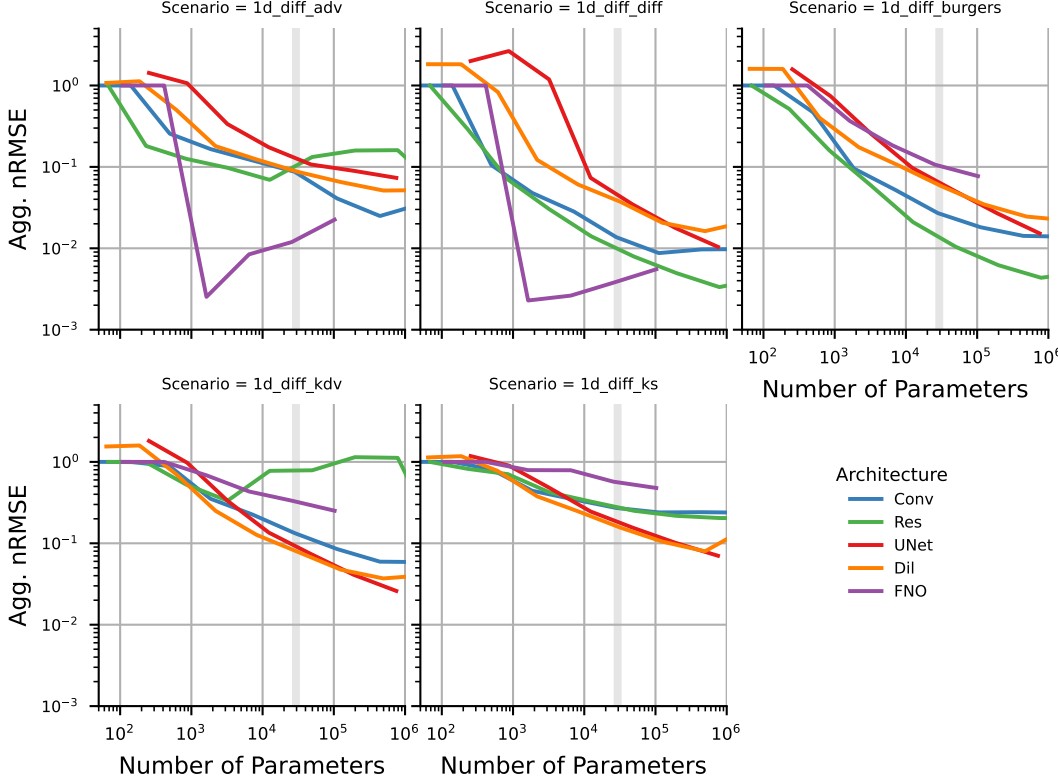

Figure 18: Parameter size ablation, depicted as the geometric mean over 100 time steps of test rollout. The gray area indicates the parameter range of the networks employed in the main experiments. Overall, all architectures benefit from larger parameter spaces, with convergence rates depending on the specific dynamics under investigation.

problem-dependent. Some architectures either reach a plateau or experience a decline in performance beyond a certain parameter threshold. We attribute this behavior to improper parameter scaling, which fails to increase the receptive field or adapt the optimization configuration effectively.

The gray shaded area in Figure 18 highlights the parameter region encompassing the default architectures for 1D scenarios, which we consider a setting that enables fair comparison.

# J Datasheet

We have included the datasheet below for completeness but emphasize that APEBench is designed as a benchmark suite with access to a powerful data generator framework in the form of the ETDRK solver framework. We leverage this by tightly integrating the numerical solver and procedurally re-creating all training and test trajectories with each new experiment execution. As such, there is no need to distribute separate datasets since the entire emulator learning specification is uniquely described in an APEBench scenario.

However, we acknowledge that this approach creates a strong dependency on the JAX and Equinox ecosystem. To mitigate this, we will release a subset of representative trajectories for the default dynamics supported by the benchmark. These trajectories can be utilized for purely data-driven emulator learning tasks and within other ecosystems like PyTorch or Julia. However, it's important to note that most functionalities, such as easy scenario modification, diverted-chain training, and correction learning, will remain exclusive to the full benchmark suite.

---

### Motivation

**For what purpose was the dataset created?** Was there a specific task in mind? Was there a specific gap that needed to be filled? Please provide a description.

The discrete emulation of a PDE effectively amounts to approximating a numerical simulator while interacting with it during training. Such interaction can be purely data-driven, and the simulator is turned off during training. However, more intricate combinations are possible. APEBench is the first benchmark that recognizes this situation and tightly integrates a differentiable JAX-based solver suite.

**Who created this dataset (e.g., which team, research group) and on behalf of which entity (e.g., company, institution, organization)?**

The dataset (or precisely the benchmark suite) was created by Felix Koehler (PhD student at the Technical University of Munich) and Nils Thuerey (Professor at the Technical University of Munich).

**Who funded the creation of the dataset?** If there is an associated grant, please provide the name of the grantor and the grant name and number.

The research of Felix Koehler is funded by the Munich Center for Machine Learning.

**Any other comments?**

No.

---

### Composition

**What do the instances that comprise the dataset represent (e.g., documents, photos, people, countries)?** Are there multiple types of instances (e.g., movies, users, and ratings; people and interactions between them; nodes and edges)? Please provide a description.

There are (more than) 46 distinct PDE scenarios (across three spatial dimensions). Each scenario has a (reproducible) procedural generation of a train and test dataset. These come in the form of arrays with defaults of 50 train trajectories of 51 time steps and 30 test trajectories of 201 time steps. The data is in a structured Cartesian format. The subsequent axes depend on channels, the number of spatial dimensions, and resolution. Each scenario also contains metadata that explicitly describes how the training is run.

**How many instances are there in total (of each type, if appropriate)?**

Answered above.

**Does the dataset contain all possible instances or is it a sample (not necessarily random) of instances from a larger set?** If the dataset is a sample, then what is the larger set? Is the sample representative of the larger set (e.g., geographic coverage)? If so, please describe how this representativeness was validated/verified. If it is not representative of the larger set, please describe why not (e.g., to cover a more diverse range of instances, because instances were withheld or unavailable).

N/A. There is the option to create new scenarios easily.

**What data does each instance consist of? "Raw" data (e.g., unprocessed text or im-**

**ages) or features?** In either case, please provide a description.

See above.

**Is there a label or target associated with each instance?** If so, please provide a description.

N/A. Each scenario is self-contained.

**Is any information missing from individual instances?** If so, please provide a description, explaining why this information is missing (e.g., because it was unavailable). This does not include intentionally removed information, but might include, e.g., redacted text.

N/A.

**Are relationships between individual instances made explicit (e.g., users' movie ratings, social network links)?** If so, please describe how these relationships are made explicit.

N/A.

**Are there recommended data splits (e.g., training, development/validation, testing)?** If so, please provide a description of these splits, explaining the rationale behind them.

Training and test data are created procedurally, and their sizes are baked into a scenario.

**Are there any errors, sources of noise, or redundancies in the dataset?** If so, please provide a description.

The reference data for any scenario using a linear PDE dynamic is fully exact, given the initial condition is bandlimited. For nonlinear PDE dynamics, the accuracy depends on the configuration but generally is decent. The user can increase the simulation method's order of consistency or add temporal substeps.

No noise exists since all simulated equations are deterministic, and the data is created synthetically. We also expect no redundancies because each initial condition is drawn separately.

**Is the dataset self-contained, or does it link to or otherwise rely on external resources (e.g., websites, tweets, other datasets)?** If it links to or relies on external resources, a) are there guarantees that they will exist, and remain constant, over time; b) are there official archival versions of the complete dataset (i.e., including the external resources as they existed at the time the dataset was created); c) are there any restrictions (e.g., licenses, fees) associated with any of the external resources that might apply to a future user? Please provide descriptions of all external resources and any restrictions associated with them, as well as links or other access points, as appropriate.

The dataset/benchmark is based on a Python library that is hosted on GitHub (`https://github.com/Ceyron/apebench`). It depends on three other Python libraries that are part of this publication, which are also hosted on GitHub (`https://github.com/Ceyron/exponax`, `https://github.com/Ceyron/pdequinox`, `https://github.com/Ceyron/trainax`). The APEBench package and the additional three packages can all be installed via pip. All libraries depend on JAX and Equinox (as well as a few other Python libraries), which are all open-source projects.

**Does the dataset contain data that might be considered confidential (e.g., data that is protected by legal privilege or by doctor-patient confidentiality, data that includes the content of individuals non-public communications)?** If so, please provide a description.

No.

**Does the dataset contain data that, if viewed directly, might be offensive, insulting, threatening, or might otherwise cause anxiety?** If so, please describe why.

No.

**Does the dataset relate to people?** If not, you may skip the remaining questions in this section.

No.

**Does the dataset identify any subpopulations (e.g., by age, gender)?** If so, please describe how these subpopulations are identified and provide a description of their respective distributions within the dataset.

Skipped.

**Is it possible to identify individuals (i.e., one or more natural persons), either directly or indirectly (i.e., in combination with other data) from the dataset?** If so, please describe how.

Skipped.

**Does the dataset contain data that might be considered sensitive in any way (e.g., data that reveals racial or ethnic origins, sexual orientations, religious beliefs, political opinions or union memberships, or locations; financial or health data; biometric or genetic data; forms of government identification, such as social security numbers; criminal history)?** If so, please provide a description.

Skipped.

**Any other comments?**

No.



**Collection Process**



**How was the data associated with each instance acquired?** Was the data directly observable (e.g., raw text, movie ratings), reported by subjects (e.g., survey responses), or indirectly inferred/derived from other data (e.g., part-of-speech tags, model-based guesses for age or language)? If data was reported by subjects or indirectly inferred/derived from other data, was the data validated/verified? If so, please describe how.

The concrete data will be procedurally generated each time a scenario is executed. Our process included setting reasonable defaults that allow for interesting yet challenging setups.

**What mechanisms or procedures were used to collect the data (e.g., hardware apparatus or sensor, manual human curation, software program, software API)?** How were these mechanisms or procedures validated?

The data is generated procedurally using JAX software components and can be produced on all computational backends JAX is available for. No custom additions to JAX's primitives were required.

**If the dataset is a sample from a larger set, what was the sampling strategy (e.g., deterministic, probabilistic with specific sampling probabilities)?**

The underlying ETDRK solver suite allows for more than the 46 PDE dynamics we presented in this work. We chose these particular equations because of their popularity.

**Who was involved in the data collection process (e.g., students, crowdworkers,** contractors) and how were they compensated (e.g., how much were crowdworkers paid)?

Only the authors were involved; no additional individuals.

**Over what timeframe was the data collected? Does this timeframe match the creation timeframe of the data associated with the instances (e.g., recent crawl of old news articles)?** If not, please describe the timeframe in which the data associated with the instances was created.

The ETDRK solver framework was developed from Sep 2023 to May 2024. APEBench was designed and implemented from Jan 2024 to May 2024.

**Were any ethical review processes conducted (e.g., by an institutional review board)?** If so, please provide a description of these review processes, including the outcomes, as well as a link or other access point to any supporting documentation.

N/A.

**Does the dataset relate to people?** If not, you may skip the remaining questions in this section.

No.

**Did you collect the data from the individuals in question directly, or obtain it via third parties or other sources (e.g., websites)?**

Skipped.

**Were the individuals in question notified about the data collection?** If so, please describe (or show with screenshots or other information) how notice was provided, and provide a link or other access point to, or otherwise reproduce, the exact language of the notification itself.

Skipped.

**Did the individuals in question consent to the collection and use of their data?** If so, please describe (or show with screenshots or other information) how consent was requested and provided, and provide a link or other access point to, or otherwise reproduce, the exact language to which the individuals consented.

Skipped.

**If consent was obtained, were the consenting individuals provided with a mechanism to revoke their consent in the future or for certain uses?** If so, please provide a description, as well as a link or other access point to the mechanism (if appropriate).

Skipped.

**Has an analysis of the potential impact of the dataset and its use on data subjects (e.g., a data protection impact analysis) been conducted?** If so, please provide a description of this analysis, including the outcomes, as well as a link or other access point to any supporting documentation.

Skipped.

**Any other comments?**

No.

---

**Preprocessing/cleaning/labeling**

**Was any preprocessing/cleaning/labeling of the data done (e.g., discretization or bucketing, tokenization, part-of-speech tagging, SIFT feature extraction, removal of instances, processing of missing values)?** If so, please provide a description. If not, you may skip the remainder of the questions in this section.

No. Each scenario's procedural generation was checked to only produce stable (non-NaN) trajectories.

**Was the "raw" data saved in addition to the preprocessed/cleaned/labeled data (e.g., to support unanticipated future uses)?** If so, please provide a link or other access point to the "raw" data.

Skipped.

**Is the software used to preprocess/clean/label the instances available?** If so, please provide a link or other access point.

Skipped.

**Any other comments?**

No.

---

**Uses**

**Has the dataset been used for any tasks already?** If so, please provide a description.

The dataset/benchmark was used for the experiments part of this publication and other internal projects of the authors.

**Is there a repository that links to any or all papers or systems that use the dataset?** If so, please provide a link or other access point.

We will collect use cases under the GitHub page of the benchmark suite: `https://github.com/Ceyron/apebench`.

**What (other) tasks could the dataset be used for?**

Options could be control and reinforcement learning. We also think that trying different numerical simulators using techniques other than pseudo-spectral discretization can be interesting.

**Is there anything about the composition of the dataset or the way it was collected and preprocessed/cleaned/labeled that might impact future uses?** For example, is there anything that a future user might need to know to avoid uses that could result in unfair treatment of individuals or groups (e.g., stereotyping, quality of service issues) or other undesirable harms (e.g., financial harms, legal risks) If so, please provide a description. Is there anything a future user could do to mitigate these undesirable harms?

No. The benchmark suite is self-contained.

**Are there tasks for which the dataset should not be used?** If so, please provide a description.

N/A

**Any other comments?**

No.

---

**Distribution**

**Will the dataset be distributed to third parties outside of the entity (e.g., company, institution, organization) on behalf of which the dataset was created?** If so, please provide a description.

All components of the benchmark are released as open-source on GitHub.

**How will the dataset will be distributed (e.g., tarball on website, API, GitHub)** Does the dataset have a digital object identifier (DOI)?

GitHub and as a registered PyPI package.

**When will the dataset be distributed?**

It is already available and can be installed by following the instructions on the GitHub page: `https://github.com/Ceyron/apebench`.

**Will the dataset be distributed under a copyright or other intellectual property (IP) license, and/or under applicable terms of use (ToU)?** If so, please describe this license and/or ToU, and provide a link or other access point to, or otherwise reproduce, any relevant licensing terms or ToU, as well as any fees associated with these restrictions.

The software components are released under a permissive license which is detailed in section A.

**Have any third parties imposed IP-based or other restrictions on the data associated with the instances?** If so, please describe these restrictions, and provide a link or other access point to, or otherwise reproduce, any relevant licensing terms, as well as any fees associated with these restrictions.

No.

**Do any export controls or other regulatory restrictions apply to the dataset or to individual instances?** If so, please describe these restrictions, and provide a link or other access point to, or otherwise reproduce, any supporting documentation.

No.

**Any other comments?**

No.

---
**Maintenance**
---

**Who will be supporting/hosting/maintaining the dataset?**

The authors of this paper.

**How can the owner/curator/manager of the dataset be contacted (e.g., email address)?**

It is best to open an issue on GitHub.

**Is there an erratum?** If so, please provide a link or other access point.

Since the benchmark suite is based on open-source software, the release notes (`https://github.com/Ceyron/apebench/releases`) will contain potential errata.

**Will the dataset be updated (e.g., to correct labeling errors, add new instances, delete instances)?** If so, please describe how often, by whom, and how updates will be communicated to users (e.g., mailing list, GitHub)?

Yes, we intent to update it in case of problems with the default setups of the scenarios if needed.

**If the dataset relates to people, are there applicable limits on the retention of the data associated with the instances (e.g., were individuals in question told that their data would be retained for a fixed period of time and then deleted)?** If so, please describe these limits and explain how they will be enforced.

N/A.

**Will older versions of the dataset continue to be supported/hosted/maintained?** If so, please describe how. If not, please describe how its obsolescence will be communicated to users.

Older versions will be tagged on GitHub and can be installed as specific version numbers from PyPI.

**If others want to extend/augment/build on/contribute to the dataset, is there a mechanism for them to do so?** If so, please provide a description. Will these contributions be validated/verified? If so, please describe how. If not, why not? Is there a process for communicating/distributing these contributions to other users? If so, please provide a description.

Pull Requests on GitHub.

**Any other comments?**

No.

