# OpenReview forum: "APEBench: A Benchmark for Autoregressive Neural Emulators of PDEs"
_NeurIPS.cc/2024/Datasets_and_Benchmarks_Track — NeurIPS 2024 Track Datasets and Benchmarks Poster_

### Official Review · Reviewer_rrDt · 2024-07-16
**Good material, but clarity could be greatly improved.**

**Rating:** 6
**Confidence:** 2

**Review:**

This paper contributes a novel data generator for evaluating multi-step training of neural PDE emulators. This a very useful contribution and makes me lean towards accepting this paper. However, it has been very difficult for me to understand key points of this paper, including the research gap. Also, the definition of multi-step training (eq 5) and the gamma difficulty were not entirely clear to me. Thus, much of the results section was hard to understand, as well and I was not able to verify the validity of all results. I am proposing some modifications below to make the paper more clear.

**Strengths:**

- My take-away from section 2 was that unrolled training can increase performance over long time horizons. That was clearly described and visualization in Figure 2.
- The ETDRK solver is very nicely described in the appendix section B.1. Honestly, this was my favorite section of this paper and I could follow it very well.
- The authors have clearly put a lot of time into the development of this dataset.
- It's very nicely chosen to have each model architecture approximately have the same number of weights. This makes it much easier to compare the architecture rather than the confounding factor of model size. It would have helped to point this design choice out in the main text rather than mentioning it in the appendix.

**Additional Feedback:**

-

**Clarity:**

- Unfortunately, it has taken me a long time to understand this paper. I believe the below suggestions could make it more clear.
- Following my confusion in eq 5, it was not clear to me which of the mentioned unrolling techniques require an autodifferentiable solver and which do not. Section 5.2 was also hard to understand, because I did not understand what diverted chain training means.
- How were the two theta parameters fitted in Section 2? Were they found with Taylor series expansion or gradient descent -- that was a bit unclear to me.
- I am not entirely sure what the purpose of Section 2 is. I like the idea of clarifying the main points of a paper with a motivating example. But, for me section 2 only made a clear example that multi-step training can improve performance over long horizons. The example did not help me better understand the PDE identifiers or rollout evaluation metrics. I would propose to remove line 107-118, such that section 2 focuses on explaining on Figure 2 only.
- The explanation of PDE identifiers (line 183) was somewhat unclear to me. I would use the space from removing line 107-118 to explain the interpretation of the PDE identifiers in more detail. For example, there's a really good explanation in the appendix line 901-908 and I would move that into the main text.
- The section 5.1 was not clear to me. I think it could be much improved if the authors introduce a table or overview of model architectures that illustrates the perspective field of each method and an illustration on how changing gamma leads to more globally distributed dynamics.
- Most sections of the appendix are not referenced in the main text. I would recommend to add references.
- Line 203-209 was not very clear to me. What is the difference between ConvNet and ResNet? Did you choose to implement the exact same architecture except for ResNet having skip connections? More generally, which actions were taken during implementation to make these 5 model architectures comparable with each other?

**Correctness:**

- Could the authors please double-check that eq 5 is correct? The diagram in Appendix Fig. 7 suggests that the formula reads l(..., P(u)) instead of l(...,P(f(u))). In my eyes, writing P_h^b(u_h) seems to be correct. This mismatch between eq 5 and Fig. 7 has lead to a lot of confusion in my evaluation. For example, there is a sentence that says "The popular one-step supervised learning problem is recovered by setting T = B = 1." But, in my mind a one-step supervised learning problem would evaluate the predicted next state, $f_\theta^{1}(u_h)$ against the simulated next state P_h^1(u_h). The one-step supervised learning problem as I understand it is not recovered by setting T=B=1. Possibly, with T=0 and B=1 and defining f^{t=0}_\theta(u_h) = u_{initial condition}.
- How does the commonly used pushforward trick fit into equation 5 (https://openreview.
net/pdf?id=vSix3HPYKSU)?
- The authors point out that existing benchmarks are limited by "temporal generalization" (l.46) and that they "introduce APEBench to address [this] limitation". The authors define temporal generalization as "[running] stably and accurately for more time steps". If I understand this definition correctly, the authors would imply that existing benchmark datasets do not evaluate methods on autoregressive rollouts which seems to be wrong. I would recommend the authors use a more precise definition of the research gap that APEBench is addressing.

**Documentation:**

Yes. The appendix introduces the ETDRK solver very well. I did not get a chance to review the code base.

**Ethics:**

No ethical concerns.

**Limitations:**

- Overall, this paper reads more as an introduction to a new data generator rather than a benchmark. Most importantly, the authors evaluate all experiments using an nRMSE which is known to have issues, such as encouraging blurry predictions and overpenalizing areas with extreme values. From a benchmark paper in PDEs I would expect a broader set of evaluation metrics and discussion thereof (see e.g., WeatherBench 2). Would it make sense to adapt the paper title to reflect that the proposed paper is "a data generator for benchmarking autoregressive ..." rather than "a benchmark for autoregressive..."? Publishing a new dataset is perfectly admissible in the NeurIPS D&B track and is seen as an equally important contribution in my point of view.

**Opportunities For Improvement:**

- This paper has done a great amount of valid work. The paper only seems limited to me in the way of presentation. In case this paper does not get accepted, I would highly recommend to point out the research gap more precisely and engage in multiple rounds of internal reviews to smoothen out hard-to-understand sections and then resubmit the paper.
- There are multiple statements that are neither referenced nor backed up by the results section. Thus, it is unclear to me if they are true. I would recommend to modify those sentences to make sure they are backed up by scientific evidence or presented as suggestions or hypotheses. For example:
-- "The FNO’s global nature and low-frequency bias makes it less suitable for this class of problems." (better: The FNO’s global nature and low-frequency bias may make it less suitable for this class of problems [reference on low-frequeny bias].)
-- "Learned emulators can use this to their advantage, and outperform their numerical counterparts for a specific operating regime."
-- "Their inductive biases for long-range dependencies reduce their ability to produce the best results in this relatively simple scenario." (this is a hypothesis; couldn't there also be other factors that explain this result?)
-- "highlighting the increasing importance of skip connections for effectively capturing information across larger spatial scales"
- The result in Fig. 5 that "FNO's performance remains almost unaffected by unrolling" was really interesting. It would be interesting to another 1-2 sentences to discuss this result.

**Relation To Prior Work:**

- Line 139-145 lists related works, but it was not clear to me which research gap exists within existing work. Could the authors please provide a clear statement on what the existing research gap within benchmark datasets is that this work is filling? It seems that before your study, there did not exist a benchmark dataset within semi-linear PDEs that allows for autodifferentible solvers; is that correct?
- Line 151-152 lists existing autodifferentiable data generators, but agian it is not clear to me what the existing research gap is. I am assuming that your work would be the first to use an autodifferentiable data generators and use it to compare multiple model architectures that were trained with multi-step training?

**Summary And Contributions:**

The authors propose a novel code base in JAX that can generate data for 46 semi-linear PDEs in 1,2, and 3D using an ETDRK solver. In comparison to other benchmark datasets, this code allows backpropagation through the solver. The authors demonstrate the dataset by evaluating the effect of multi-step finetuning and various deep learning architectures.

---

> ### Author Rebuttal · Authors · 2024-08-15
>
> Dear reviewer rrDt,
>
> Thank you for your valuable and detailed feedback. We appreciate that you enjoyed the detailed explanation of the ETDRK solver suite. Regarding the research gap (APEBench targets unrolled strategies, temporal generalization, and autodifferentiable solvers) and the focus on the nRMSE metric, please refer to the general answer above. We comment on your other points below and are happy to provide further clarification during the discussion phase:
>
> 1. **Equal Parameter Count mention:** We will mention the equal number of parameters for architectures in the main text.
> 2. **Unreferenced Interpretations:** We appreciate your comments regarding these statements. Many of them are based on observations from APEBench experiments, which we will reference and discuss in our paper correspondingly. Specifically:
>    1. “The FNO’s global nature and low-frequency bias makes it less suitable for this class of problems.”
>       1. FNOs can learn to predict solutions with more active modes than they have by learning to control the energy transfer into the higher modes (which is done by the nonlinear activation function; see also our reply to reviewer psus). Since there are explicit weights for the lower modes (lower frequencies) and only an indirect influence on the higher modes, we wrote “low-frequency bias”. Hence, we think that an FNO can only perform well on spectrally rich problems if there is strong decay over the wavenumbers, explaining the suboptimal performance in Figure 6b. Since reaction-diffusion equations develop sophisticated patterns (considerable energy in the higher modes), we conclude that the FNO is not an optimal architecture in this case.
>    2. “Learned emulators can use this to their advantage and outperform their numerical counterparts for a specific operating regime.”
>       1. We argue this is evident from Figure 2\. Since the emulators have stencil entries different from the FOU scheme, they are inconsistent (for a kernel size of two, the FOU scheme is the only numerical method that arises from the advection equation via Taylor expansion). Consistency is a strong numerical constraint, which it hence violates. Still, it achieves lower error than the (consistent) FOU scheme (for some initial steps), which we subsequently interpreted as “outperforming \[the\] numerical counterpart.”
>    3. “Their inductive biases for long-range dependencies reduce their ability to produce the best results in this relatively simple scenario.”
>       1. The UNet and Dilated ResNet performed best in complex nonlinear scenarios requiring larger receptive fields (see Figure 6b). However, for problems with smaller receptive fields, some of their parameters related to long-range interactions may be underutilized. This is what we meant by "inductive bias for long-range dependencies," as seen in Figure 3 at lower difficulties.
>    4. “Highlighting the increasing importance of skip connections for effectively capturing information across larger spatial scales”
>       1. We will revise this sentence to simply state, "highlighting the increasing importance of skip connections," removing the latter part.
> 3. **FNO’s invariance to unrolled training for 2D advection emulation:** Our interpretation of these results is that unrolling boosts performance for architectures with a limited receptive field, especially for challenging dynamics (=high $\\gamma$ values). Since the FNO can see virtually all of the domain and its performance for linear (and bandlimited) problems is unchanged once it has enough active modes (see additional results in our comments to reviewer “psus”, and the results of Figure 3 indicating performance invariance w.r.t. $\\gamma\_1$), this effect is reduced. FNOs are excellent emulators for these linear PDEs on periodic BCs because they essentially behave like the data-generating solver, a pseudo-spectral method.
> 4. **Focus on data generator:** We agree that the ETDRK-based solver is a central contribution of APEBench (which we will also release as an individual Python package). At the same time, we think it is likewise important to address the research gap of benchmarks including a numerical simulator, as outlined in the general answer above.

---

> > ### Author Rebuttal · Authors · 2024-08-15
> >
> > 5. **On Eq. (5) and the taxonomy of unrolled strategies:** Indeed, there was a problem with Eq. (5). Thank you for pointing this out to us. The limits of the first sum should be $\\sum\_{t=0}^{T-B}$ instead of $\\sum\_{t=1}^{T-B+1}$. Hence, the correct formula becomes $$L(\\theta) = \\mathbb{E}\_{u\_h \\propto \\mathcal{D}\_h}\\left[ \\sum\_{t=0}^{T-B} \sum_{b=1}^B l \\left ( f\_{\\theta}^{t+b}(u_h), \mathcal{P}\_h^b(f\_\theta^t(u_h)) \\right ) \\right].$$ We used the notation that a function raised to an exponent denotes an autoregressive/recursive application. If it is raised to zero, this should be interpreted as the function not being applied (i.e., resorting to the identity). In this case, supervised unrolling ($T=B$) would only leave the application of the fine solver $\mathcal{P}\_h$ and the second sum over the branch chain (together with $t=0$), which then reads $$L\_{\\text{supervised unrolled}}(\\theta) = \\mathbb{E}\_{u\_h \\propto \\mathcal{D}\_h}\\left[ \sum\_{b=1}^{B=T}l(f\_\theta^b(u\_h), \mathcal{P}\_h^b(u\_h)) \\right].$$ Hence, there are as many loss contributions as time steps in the main chain in this case. In the case of simple diverted chain learning with T freely chosen and B=1, only the first sum would remain, and we get (with $b=1$) $$ L\_{\\text{diverted chain (branch length 1)}}(\\theta) = \\mathbb{E}\_{u\_h \\propto \\mathcal{D}\_h}\\left[ \sum\_{t=0}^{T-1} l(f\_\theta^{t+1}(u\_h), \mathcal{P}\_h(f\_\theta^t(u\_h))) \right].$$ Again, there are as many loss contributions as time steps in the main chain. APEBench also supports the most general case with T freely chosen and $B\ge2$. In this case, one would get cross terms, and both sums remain. For brevity, we did not present any results under such a configuration. We acknowledge that this section in the text can benefit from an enhanced explanation. We will also add a schematic of a general diverted chain case ($B\ge2$) to the appendix. Note that the typo in Eq. 5 does not affect the experimental results since the implementations in APEBench do not use this formula directly.
> > 6. **Pushforward Trick in Taxonomy:** The pushforward trick of Brandstetter et al. (2022) requires the more general form of Eq. (79), which includes weights for each time level. We chose not to display this most general form in the main text due to brevity, but this is the interface used by APEBench to set up unrolled training strategies. Hence, pushforward would be $T=B=2$ with the weights $w\_1 \= 0$ and $w\_2=1$ (weighing only the second step) and the backpropagation-through-time being disabled (see also our remarks in section D.2).
> > 7. **Temporal Generalization**: We acknowledge that the wording in the introduction could be improved. Our intention was not to imply that previous benchmarks did not explore temporal generalization but rather that they focused on aggregated metrics rather than error rollouts. For instance, PDEBench uses rollout metrics primarily to ablate training choices or to show behavior beyond the temporal horizon of training data. We elaborate further on this in our general author rebuttal. We will revise the introduction accordingly.

---

> > ### Author Rebuttal · Authors · 2024-08-15
> >
> > 8. **Clarity and Specific Recommendations**: We appreciate your suggestions for improving clarity and will consider them. Below are some specific responses:
> >    1. **Differentiable Physics in Training:** Below, we provide a decision guide as to which setup requires the autodifferentiable solver. For each version, a ✅ indicates whether we need an autodifferentiable solver. A setup with ☑️ does require a solver to be called on-the-fly, but does not require differentiability. To the best of our knowledge, configurations with either of the two ticks are unique to APEBench and have not yet been part of other benchmark publications.
> >       1. One-Step supervised training ($T=B=1$):
> >          1. Pure prediction:
> >             1. No autodifferentiable solver required.
> >          2. Correction:
> >             1. Dynamically called coarse solver required, but not autodifferentiable ☑️.
> >       2. Supervised Unrolling ($T=B$):
> >          1. Pure prediction:
> >             1. No autodifferentiable solver required.
> >          2. Correction:
> >             1. If using a coarse solver without modifications: Autodifferentiable coarse solver required ✅.
> >             2. If backpropagation-through-time (BPTT) is disabled: Dynamically called coarse solver required, does not need to be differentiable ☑️.
> >       3. Diverted Chain Unrolling ($1 \\le B \< T$):
> >          1. Pure prediction:
> >             1. If using the fine solver without modifications: Autodifferentiable fine solver required ✅.
> >             2. If the diverted chain is cut: Dynamically called fine solver required, does not need to be differentiable ☑️.
> >          2. Correction:
> >             1. If using both fine and coarse solvers without modifications: Both autodifferentiable fine and coarse solvers required ✅.
> >             2. If the gradients through the diverted chain are cut: Dynamically called but not autodifferentiable fine solver required, along with an autodifferentiable coarse solver ✅.
> >             3. If BPTT is cut: Dynamically called but not autodifferentiable coarse solver required, along with an autodifferentiable fine solver ✅.
> >    2. **Intuition of Diverted Chain:**  The diverted chain training method aims to combine the benefits of one-step supervised training, which provides direct feedback on the emulator's predictions, with autoregressive unrolling. The latter accounts for long-term feedback and distribution shifts over multiple steps. (Please cf. reviewer psus)
> >    3. **Parameter Fit in Section 2:** For efficiency, the parameters in the motivational example are fitted using a Newton optimizer since the problem is approximately convex (see section G.2).
> >    4. **On the purpose of section 2:** Without rollout metrics, it would be challenging to identify regimes where emulators are superior. We agree that the relevance of PDE identifiers was not elaborated sufficiently and will address this. (For an intuition, please also refer to the answers for reviewer psus)
> >    5. **Enhancing the clarity of section 5.1** with a schematic of receptive fields is a great suggestion we will incorporate in a future revision.
> >    6. We will add **references to the appendix** where applicable.
> >    7. Yes, conceptually, the **difference between the used ResNet and ConvNet** is that the former has skip connections.
> >    8. **Model Comparability**: The five models were selected as representatives of three architecture classes (local convolutional, long-range convolutional, and pseudo-spectral) to explore fundamental differences in behavior. We tuned the hyperparameters to ensure similar parameter sizes, and the parameter ablation is discussed in section H.4.

---

> > > ### Comment · Reviewer_rrDt · 2024-08-15
> > > **good clarification + score increase**
> > >
> > > Overall the authors rebuttal has resolved many of my confusions and the authors seem committed to changing the main text to resolve the mentioned issues. Thus, I have increased my score from 5 to 6.
> > >
> > > - Your comments on "unreferenced interpretations" in 2.1 and 2.3 have helped my understanding. Thank you for formulating the answers and incorporating them in the main text.
> > > - My confusion regarding the unrolled training had been addressed by your answers in 5 and 6. Your answer was very clear and I'm glad to hear that results are not impacted by it.
> > > - The research gap has been somewhat clarified. The list of 1-8 other packages had greatly helped me understand how the ETDRK solver package is contributing to existing packages; although I would need to see experimental evidence to agree to it being the faster solver package. I'm still a bit confused on the difference between APEBench, PDEBench, and PDEArena. It seems that the main benefit of APEBench is that it uses the ETDRK solver, uses Jax instead of pytorch. The difference in evaluation of the unrolled training in plotting errors  over time instead of only showing time-aggregated metrics and integrating over 100 instead of 5 steps seem minor changes, but allow for a bit more comprehensive evaluation, I would agree.
> > > - It is good to hear that you're adding more metrics beyond nRMSE for the camera ready NeurIPS version.
> > >
> > > Thank you!

---

> > > > ### Author Rebuttal · Authors · 2024-08-20
> > > >
> > > > Dear Reviewer,
> > > >
> > > > Thank you for your prompt response and for increasing the score of our paper. We appreciate your thoughtful feedback and the opportunity to address your concerns.

---

### Official Review · Reviewer_HkPL · 2024-07-24
**Very good and comprehensive paper**

**Rating:** 9
**Confidence:** 3
**Clarity:** The paper is well-written.

**Review:**

This work is well-constructed with a robust methodological framework. The paper is clearly written, with well-structured sections that guide the reader through the functionality and advantages of APEBench.  Introducing a differentiable simulation suite and a unique dynamics identifier in a benchmark are novel features that set APEBench apart from existing benchmarks.

As acknowledged in the Section 5.5 Limitations, this work is limited to periodic boundary conditions and uniform Cartesian grids. It lacks consideration of non-Cartesian and irregular geometries which are common in real-world applications.
I only have some minor concerns:
- In equation 2, what is the meaning of the star symbol?
- The paper does not describe the meaning of $\theta_{center}$ and $\theta_{right}$ in Section 2.
- The range of the y-axis of Figure 16 needs to be revised. Some lines are partly invisible.

**Strengths:**

The benchmark’s extensive support for various dynamics and neural architectures makes it a versatile tool for the community. The integration of a JAX-based framework enhances practical utility.

**Additional Feedback:**

N/A

**Correctness:**

The claims and methodologies presented in the paper are technically sound, and the experimental setup is robust.

**Documentation:**

This paper gives a brief overview of two usages of APEBench. The authors will public the codes and datasets once the paper is accepted.

**Ethics:**

No ethical concerns.

**Limitations:**

The authors have acknowledged the limitations related to the scope of boundary conditions and the types of geometries supported. Future work could expand these aspects to increase the benchmark’s utility in practical scenarios.

**Opportunities For Improvement:**

As acknowledged in the Section 5.5 Limitations, extending the benchmark to include non-periodic boundary conditions and non-uniform geometries could broaden this work’s applicability to real-world scenarios.

**Relation To Prior Work:**

The paper adequately discusses how APEBench differs from and improves upon existing benchmarks like PDEBench and PDEArena.

**Summary And Contributions:**

This paper comprehensively investigates how dynamics, neural architecture, and simulator-emulator interaction affect performance. It also explores neural-hybrid emulators and differentiable physics training. APEBench offers four key contributions:
- APEBench provides 46 distinct PDE dynamics across 1D, 2D, and 3D configurations, enhancing the benchmark’s utility across different research domains.
- For each distinct type of dynamics, APEBench provides a unique identifier that encodes its difficulty of emulation.
- A JAX-based framework that supports differentiable physics training and the evaluation of neural-hybrid emulators.
- Introduces a framework for examining the impact of various training paradigms on emulator performance.

---

> ### Author Rebuttal · Authors · 2024-08-14
>
> Dear Reviewer,
>
> Thank you for the evaluation of our paper. We are glad that you are convinced that the methodological basis is solid. We will address your questions below:
>
> 1. **The meaning of the star symbol in Eq. 2:** This refers to “cross-correlation,” which is internally done by ConvNets in the major machine learning frameworks in Python ([https://en.wikipedia.org/wiki/Cross-correlation\#Properties](https://en.wikipedia.org/wiki/Cross-correlation\#Properties)), i.e., PyTorch, TensorFlow and JAX. If we did true convolution, it required a flip/reverse on the filter. See also here: [https://pytorch.org/docs/stable/generated/torch.nn.Conv2d.html](https://pytorch.org/docs/stable/generated/torch.nn.Conv2d.html) .
> 2. **The meaning of $\\theta\_{\text{center}}$ and $\\theta\_{\text{right}}$”:** For a “convolution” (or better cross-correlation) of filter/kernel size 2, there are two learnable parameters. If the filter size is even, “convolutions” in the Python deep learning frameworks are biased to the right. Hence, the first entry in the filter is the central weight ($\theta_{\text{center}}$), and the second entry is the weight associated with the degree of freedom to the right ($\theta_{\text{right}}$). There is no filter weight for the dof to the left. We will enhance the clarity on this point in future revisions.
> 3. **Figure 16 Adjustment**: We will revise the limits of Figure 16 to ensure that the curve representing the FNO no longer clips at the bottom.

---

> > ### Comment · Reviewer_HkPL · 2024-08-28
> >
> > Thank you for the response. The authors have addressed my concerns and I do not have other questions. I will keep my rating at 9.

---

### Official Review · Reviewer_oS3v · 2024-07-24
**Nice benchmark for autoregressive models**

**Rating:** 7
**Confidence:** 4
**Correctness:** The methodology is very sound. Reprod…
**Clarity:** The paper is written in a clear and u…

**Review:**

The authors provide a complete framework to easily evaluate different
types of architectures on different time-dependent PDE problems. Their
focus within their framework lies on autoregressive unrolled neural
operators.

Their implementation is inspired by PDEArena. Compared to
e.g. PDEArena or PDEBench, they extend the amount of problems
provided. On the other hand, the presented problems are restricted to
periodic boundary conditions; though the authors claim that they could
do Neumann or Dirichlet boundary conditions as well. Furthermore,
pre-computed datasets are not provided, but have to be computed.

The authors provide a holistic framework for easy evaluation of
different neural architectures on their problems, this is however
limited within their framework, as soon as you want to train your own
model within a different framework like TensorFlow or pyTorch. You
would just call 'get_train_data()' on one of their scenarios and then
loose all of the other functionality.

The paper presents vast benchmark results that are reproducible. It
seems less to be the aim of the paper to provide a tool that can be
used/extended by users.

Another stated focus of the paper lies on the taxonomy of unrolled
neural operators. This is essentially only grouping them with respect
to T, the number of unrolled steps, and B, the length of the branch
chain. Thus nothing unexpected.

**Strengths:**

A large set of PDEs and NNs.

The benchmark suite is very easy to use. A minimal config to start
experiments is very short and easily understandable. Their web-based
volume rendering is a nice addon to have.

A lot of supplementary material is provided.

**Additional Feedback:**

What does "this relatively simple scenario" refer to in line 259? If
the performance of UNets and ResNets deteriorates with increasing
difficulty, then why is the ability to produce best results in the
simpler scenario reduced?

Please correct the spelling in the references for names and acronyms
such as pde -> PDE.

**Documentation:**

The provided  documentation is excellent, covers code snippets for a
quick start, but also examples with more depth to it.

Reproducibility is given.

**Ethics:**

Unproblematic

**Limitations:**

Limitations of the proposed method are addressed.

**Opportunities For Improvement:**

An example on how own architectures can be evaluated is provided,
however it only covers architectures within their own framework,
called pdequinox. It seems that own models from different frameworks
cannot be evaluated within APEBench. Datasets can be exported
however.

The compulsory Croissant metadata format is missing.

**Relation To Prior Work:**

The authors mention several important prior papers in
section 3. However, they lack to state in which points their work
differs.  One part they mention is their focus on autoregressive
unrolled models.  This is however more part of the models handling of
data and not the benchmark.

**Summary And Contributions:**

The authors provide a complete framework for benchmarking so-called
neural emulators (or neural networks as surrogates) for solving
PDEs. Their novel contribution is their focus on unrolled
autoregressive models for a PDE-based benchmark.

---

> ### Author Rebuttal · Authors · 2024-08-15
>
> Dear Reviewer,
>
> Thank you for your valuable feedback. We are pleased that you found the paper clear to understand and valued the available documentation. We will revise the abbreviations in the bibliography as suggested. For a detailed discussion of our work's relation to prior research, please refer to the general author's rebuttal. Below, we address the specific points raised in your review:
>
> 1. **Croissant Format, and Precomputed Datasets**: Upon acceptance, we will provide a subset of the benchmark (for the 46 dynamics) as precomputed data trajectories, including separate sets for training and testing. These datasets will be independent of the APEBench suite and can be used for purely supervised approaches with various frameworks, such as PyTorch, TensorFlow, Julia, or other JAX-based deep learning libraries like Flax. The datasets will be accessible through HuggingFace with a Croissant identifier. Given that APEBench generates all data procedurally, we found that the Croissant format was not the best fit for our current needs. In section I of the appendix, we elaborate on this.
> 2. **Expandability:** We acknowledge that APEBench is a tightly integrated suite that requires architectures to be available in an Equinox format. However, we think this is the only way to enable such a deep integration of the differentiable solver to allow correction learning (=neural-hybrid emulators) or the range of unrolled training strategies. We found the JAX/Equinox ecosystem the most suitable for this.
> 3. **Taxonomy of Unrolled Strategies**: Beyond the grouping with T (number of autoregressively unrolled steps at training time) & B (length of the branch chain, i.e., the number of autoregressive steps of the reference solver $\\mathcal{P}\_h$) as described in Eq. 5, APEBench allows for further customization, such as setting time-level weights and adjusting the gradient flow, as detailed in section D of the appendix. This flexibility enables the implementation of advanced techniques like the pushforward trick (Brandstetter et al., 2022\) or gradient chain truncations (List et al., 2022). For more information, please refer to our response to reviewer rrDt.
> 4. **Clarification on “Relatively Simple Scenario”**: The term "relatively simple" refers to the well-behaved nature of the advection equation. As a hyperbolic PDE, it provides a priori knowledge of the receptive field, and being a linear PDE with periodic boundary conditions, the state remains bandlimited.
> 5. **The performance of UNets and \[Dilated\] ResNets in Section 5.1:** We apologize if the wording was unclear. The two findings — (I) that performance “deteriorates with increasing difficulty” and (II) that their “ability to produce the best results in the simpler scenario is reduced” — are separate observations. The latter (II) refers to the UNet and Dilated ResNet performing worse than the two local convolutional architectures (ResNet and ConvNet with depth 10\) for $\\gamma\_1=0.5$ and $\\gamma\_1=2.5$, despite having sufficient receptive field. We attribute this to their inductive bias toward long-range dependencies, dedicating many parameters to interactions not needed for this scenario. The former (I) is observed in the steeper red and orange curves in Figure 3a as $\\gamma\_1$ increases.

---

> > ### Comment · Reviewer_oS3v · 2024-08-31
> >
> > I thank the authors for the explanations. I am still convinced that it is a good paper.

---

### Official Review · Reviewer_psus · 2024-07-25
**Dataset for spatiotemporal PDEs to evaluate autoregressive neural emulators and connection with numerical methods**

**Rating:** 7
**Confidence:** 3
**Correctness:** Seems correct
**Clarity:** Yes, although the language could be s…

**Review:**

Well-written paper focusing on an extremely important problem in emulating long-term dynamics and connections to numerical methods. The paper is mostly clear and significant

**Strengths:**

* Well-written
* Comprehensive dataset of many PDEs and benchmarked with several NN models
* Strong motivations (simple examples to motivate the issue are very nice)
* long-term stability of emulators are very important at scale.

**Additional Feedback:**

-

**Documentation:**

Yes

**Ethics:**

No concerns

**Limitations:**

Discussed

**Opportunities For Improvement:**

* I think overall some of the language used is a little dense and could be made simpler for the reader to understand.
* I don't think I followed what the diverted chain is. I think the authors could explain a bit more here. Is a coarse simulation produced using the neural network prediction? Similarly, I roughly followed the pde identifiers section and connections to CFL but maybe the authors could add additional explanations here with the different gamma values.
* There is an overall sense the unrolling during training helps in overall accuracy. It would be nice to see if it degrades other metrics like spectral metrics. The blurring problem in weather forecasting is well-known where autoregressive training leads to better accuracies but smoother solutions. It would be interesting to see if this connects to the \gamma values.
* I didn't follow some of the architecture explanations. The authors say FNO is good for underresolved physics. Does that mean its good for smooth problems? They also say limited active modes limit the FNO. What happens if you don't limit the active modes? Would the FNO performance increase? It would be good to validate that explanation.
* It might be good to add another global model to validate some of the above explanations like a ViT

**Relation To Prior Work:**

Yes

**Summary And Contributions:**

Authors introduce a PDE dataset with several classes of problems in 1d, 2d, 3d keeping in mind long-term spatiotemporal dynamics and connect the performance of several neural network emulators to concepts from numerical methods. This helps in improving our understanding of these emulators and the connections between architecture, PDE dynamics type and training methods. Several NN models are benchmarked in autoregressive performance to demonstrate the above.

---

> ### Author Rebuttal · Authors · 2024-08-15
>
> Dear Reviewer,
>
> Thank you for your constructive feedback on our manuscript. It is encouraging to see the positive impression of the motivational example. Based on your input and feedback from other reviewers, we plan to soften the language in certain sections, particularly when explaining the taxonomy of unrolled training strategies. Below, we reply to your specific comments (we will add the additional ablations also to future versions of our manuscript):
>
> 1. **Intuition behind Diverted Chain Training:** The diverted chain training method aims to combine the benefits of one-step supervised training, which provides direct feedback on the emulator's predictions, with autoregressive unrolling. The latter accounts for long-term feedback and distribution shifts over multiple steps. In this approach, as depicted in Figure 8, the reference solver diverges (=branches off) after each network prediction (hence the name "diverted chain"). This setup in Figure 8 uses T=3 and B=1, while the supervised unrolled training with T=B=3 is illustrated in Figure 7\. Unlike the fully pre-computable reference data in supervised unrolled settings, the solver in the diverted chain setup is dynamically called on the network state and differentiated during the reverse pass, which is efficiently managed within APEBench. Diverted-chain training is independent of whether the learning goal is prediction or correction, hence there is not necessarily a coarse solver involved. In our reply to reviewer rrDt, we added a decision guide which setup/learning goal requires differentiable solvers and are hence unique to APEBench.
> 2. **Explanations of PDE Identifiers:** The motivation behind the PDE identifiers is to provide clarity on the difficulty of the emulation task. Simply naming a PDE (e.g., Burgers) does not convey how challenging it is for the emulator, as factors like constitutive parameters, domain size, resolution, and time step size significantly impact this. We aimed to introduce a concise set of information that accurately describes what is being emulated, serving as an exchange protocol or an identifier for an experiment. It is already possible to absorb domain size ($L$) and time step size ($\\Delta t$) into, what we call, normalized dynamics (outlined in section B.7). These normalized dynamics are independent of the spatial resolution ($N$). Hence, we introduced the difficulties ($\\gamma$ and $\\delta$) that scale the dynamics based on the resolution. The intuition behind this is that for convolutional networks (or more generally speaking, any explicit finite difference method) it is harder to emulate a problem such as advection (with the same normalized dynamics) on a higher resolution than on a lower resolution. For advection, this is exactly described by the CFL number which is also the interpretation of $\\gamma\_1$. The CFL number poses a stability criterion for the first-order upwind method to the advection equation, i.e., the most compact finite difference scheme. For the second order problem of the diffusion equation, the most compact finite difference scheme is the FTCS method ([https://en.wikipedia.org/wiki/FTCS\_scheme](https://en.wikipedia.org/wiki/FTCS\_scheme) ), that also has a stability constraint which we use as the $\\gamma\_2 \= 2 \\nu \\Delta t (L/N)^2$ (FTCS stability requires $\\gamma\_2 \\le 1$). This can be generalized for all linear derivative operators in Eq 77\. We also define a similarly motivated identifier for the nonlinear components in Eq 78\. Ultimately, in order to identify a dynamic (and its difficulty of emulation), it is sufficient to know the respective $\\gamma$ and $\\delta$ values (listed in Table 2\) and the resolution $N$ (and the dimension $D$).
> 3. **Does unrolled training, while improving the nRMSE metric, degrade other metrics?:** This is a great question that we will elaborate on in future versions as outlined below. In the general author rebuttal we discussed our intention of adding derivative based metrics (=Sobolev norms). In the meantime, we repeated the experiment of section 5.2 and present below the geometric mean over the error rollout for 100 test time steps with two different metrics: the classical nRMSE metric and an nRMSE metric for a specific frequency range (akin to PDEBench), here looking at modes 6 and upwards (i.e., the nRMSE after high-pass filtering). The results suggest that unrolled training positively impacts both the overall metric and higher modes, indicating that it does not lead to blurry predictions for our test cases.
> | Metric         |       | Agg. nRMSE |       | high-pass Agg. nRMSE |
> |----------------|------:|-----------:|------:|---------------------:|
> | Unrolled Steps |     1 |          5 |     1 |                    5 |
> | Burgers        | 0.012 |      0.010 | 0.032 |                0.027 |
> | KdV            | 0.803 |      0.105 | 1.196 |                0.190 |
> | KS             | 0.274 |      0.242 | 0.223 |                0.180 |

---

> > ### Author Rebuttal · Authors · 2024-08-15
> >
> > 5. **FNO and underresolved physics:** We call problems “underresolved” if the spectrum produced by an equation is too rich for the chosen resolution $N$. In other words, the highest resolved mode carries non-zero energy and it is clear the hypothetical higher modes would too. This is typical for Navier-Stokes at high Reynolds number (turbulent cascades) but we also have it prototypically with the KS equation. Since the FNO learns to predict the energy in modes beyond its active modes via the nonlinearity it is prone to aliasing (for a discussion on this, see McCabe et al. (2024)). We hypothesize that this aliasing might help the FNO infer behavior beyond the resolvable spectrum, and we will revise our manuscript to make clear that it is a hypothesis.
> > 6. **On FNO and its active modes:** If a problem has more active modes than the FNO has (which all nonlinear problems in APEBench do), the FNO has to learn the prediction in those higher modes indirectly via its nonlinearity (which moves energy between the modes) and cannot learn weights for these modes directly. Below, we present the results of the FNO after adjusting the active modes (all the way to 80 modes; there are 81 total modes on the default 1D resolution of $N=160$) for the geometrically aggregated rollout error over 100 time steps (similar to the results of Figure 6). `num_modes=12` was the default of the paper. Since the advection example was bandlimited to have a cutoff at K=5, the FNO needs at least six active modes (because this also includes the mean/zero mode). Beyond that, there is no further improvement in its performance. The Burgers problem sees a similar jump in performance because it also requires the six active modes to understand the smooth initial condition. It then further benefits from additional modes because the spectrum is fully populated. However, interestingly, it does not need all modes (=80) to achieve the lowest performance, confirming our hypothesis that a significant part of the FNO’s prediction comes from the movement of energy into higher modes due to its pseudo-spectral nature. The KS example directly starts within the spectrally rich chaotic attractor. Yet it also achieves an almost similar performance with less than a third of the highest possible number of modes. Overall since the performance of 20 modes is better than the performance at 12 modes for the two nonlinear problems, this should confirm our statement that “\[...\] while limited active modes hinder the FNO’s performance in some cases” (caption of Figure 6), especially in the KS example.
> > | Num Modes |    1 |    4 |    5 |    6 |   10 |   12 |   20 |   25 |   40 |   80 |
> > |-----------|-----:|-----:|-----:|-----:|-----:|-----:|-----:|-----:|-----:|-----:|
> > | Advection | 1.00 | 1.08 | 1.06 | 0.03 | 0.03 | 0.03 | 0.03 | 0.03 | 0.03 | 0.03 |
> > | Burgers   | 0.88 | 0.75 | 0.32 | 0.18 | 0.08 | 0.07 | 0.03 | 0.03 | 0.03 | 0.03 |
> > | KS        | 0.78 | 0.78 | 0.76 | 0.79 | 0.70 | 0.55 | 0.18 | 0.11 | 0.10 | 0.10 |
> > 8. **Adding the Vision Transformer.** Thank you for this suggestion. We agree that transformers are increasingly relevant in neural PDE emulation, and we plan to incorporate them in a future version of the benchmark.

---

### Author Rebuttal · Authors · 2024-08-15

Dear Reviewers,

Thank you for your valuable input. It is encouraging to see your positive remarks that this benchmark suite could be a valuable contribution to the community. Below we will address two generic points: the clarity of the research gap and the exclusive use of the nRMSE metric in our evaluation. We will include these discussions in future versions of our submission to enhance readability:

1. **Clarity of research gap:** The primary contribution of APEBench lies in its tightly integrated pseudo-spectral ETDRK solver, which, to our knowledge, is the fastest available method for generating reference data for emulator training. This integration enables, e.g., correction learning (i.e., neural-hybrid emulators) that is not addressed by existing benchmarks. This is facilitated by a systematic taxonomy of learning methodologies. Below, we provide a more detailed discussion of the research gap and how APEBench addresses it:
   1. **Neural-Hybrid Emulators:** Although seminal works such as Kochkov et al. (2022), Um et al. (2020), and most recently NeuralGCM (Kochkov et al. (2024)) have utilized hybrid solvers with coarse solver components, no other benchmark, to our knowledge, includes setups for these hybrid approaches. Existing benchmarks like PDEArena and PDEBench primarily offer fixed datasets. They release solvers or their configuration files, but they do not directly support the flexible use of these solvers during training. Additionally, some solvers are implemented in NumPy or other programming languages, making them essentially non-differentiable. Therefore, there is a clear need for a benchmark that incorporates hybrid approaches.
   2. **The ETDRK solver:** To the best of our knowledge, no other benchmark comes with a solver that is as fast or efficient as APEBench’s solver. The solver uses state-of-the-art mathematical methods for semi-linear problems (ref. Montanelli & Bootland (2020)) combined with JAX’ rich feature set. Below, we give details on how this differs from other popular PDE solvers with high-level interfaces:
      1. JAX-CFD: While JAX-CFD has a spectral submodule, it focuses on fluid-like problems in 1D and 2D. It supports many integration schemes, but no ETDRK methods, and not many types of equations are directly available.
      2. JAX-MD: is primarily built for molecular dynamics which is different from the PDEs addressed in APEBench.
      3. JAX-Fluids: focuses on finite volume methods for strongly hyperbolic problems which is a different class of problems than what APEBench can solve, as outlined in section B.3.
      4. Warp: contains many low-level routines and tools for finite element solutions which are also differentiable. It does not provide solvers for pre-built dynamics like the one in APEBench.
      5. PhiFlow: focuses primarily on fluid simulation in primary variables and multiple boundary conditions. This requires custom grid data structures, and the interface with neural networks is non-trivial. It does not support ETDRK methods.
      6. Dedalus: is a spectral solver with support for ETDRK methods, but it uses NumPy and does not come with pre-built dynamics. It is not differentiable.
      7. Chebfun (in MATLAB): provides pseudo-spectral ETDRK methods, but does not integrate with deep learning in Python. It also lacks features like GPU execution or automatic vectorization.
      8. FourierFlows.jl (in Julia): Is a set of tools to build pseudo-spectral ETDRK solvers. It does not have pre-built dynamics, but is differentiable and should work on GPUs, however only within the Julia ecosystem.
   3. **Broad selection of dynamics and unique identifiers**: While some PDEs in our work have appeared in other benchmarks, we enhance clarity by assigning unique identifiers to each emulation target. Additionally, APEBench supports PDEs such as the 3D KS equation, 3D reaction-diffusion equations, and dispersion/hyper-diffusion equations, which, to our knowledge, have not yet been explored for emulator learning. This is possible in APEBench because the ETDRK solver suite can handle these (linearly) stiff equations easily. Deterministic and procedural data generation also avoids downloading large datasets in the order of hundreds of GBs or even TBs. Instead, installing a single pip package of \~10 MB is sufficient. (See also our reply to reviewer oS3v)
   4. **Taxonomy of unrolled training methodologies:** While previous benchmarks also consider unrolled training (e.g., in PDEBench, it is called “autoregressive” training), to the best of our knowledge, no benchmark has yet taken a similarly systematic and broad approach. There is still little work on understanding the impact of unrolled training, which we believe APEBench can foster due to its flexible setup of how simulator and emulator interact during training.
   5. **Focus on temporal generalization via rollout metrics:** The two seminal benchmark papers, PDEBench and PDEArena, do focus on time-dependent problems but primarily report aggregated metrics over the rollout trajectory. E.g., PDEArena reports the mean over a rollout of only 5-time steps. Our results show that this is too short to reliably evaluate whether an autoregressive emulator is stable, which in turn indicates that having longer-term error rollouts as first-class citizens is important. For example, without them, it would not be directly evident to show that the emulator beats the simulator in the motivational example (figure 2\) or that there is a regime in which the ResNet performs almost identical to the FNO (table 8).
   6. **Emulator Architectures in JAX:** Whereas PDEBench and PDEArena include an array of representative neural architectures in PyTorch, APEBench re-implements the majority of them agnostic to the spatial dimension and boundary condition in JAX. To the best of our knowledge, there is currently no equally comprehensive repository of architectures.

---

> ### Author Rebuttal · Authors · 2024-08-15
>
> 2. **Exclusive usage of nRMSE:** APEBench is designed to be used with more than just MSE-based metrics. It already provides built-in support for different metrics, e.g., RMSE metrics that look at a specific range of frequencies (similar to PDEBench). This was also demonstrated in the supplemental material, e.g., in ([https://anonymous.4open.science/r/apebench-EC1E/apebench/\_base\_scenario.py](https://anonymous.4open.science/r/apebench-EC1E/apebench/\_base\_scenario.py) line 793). (Also see our reply to reviewer psus.) We intend to add more typical metrics like Sobolev losses (i.e., based on derivatives of the fields) in future versions. We have experimented with many different metrics in the past but found that nRMSE reliably summarizes an emulator's behavior for the investigated PDEs. Hence, for brevity, we have only provided this metric in our submission. Nonetheless, based on your feedback, we will compute and add additional metrics to motivate this choice.

---

### Decision · Program_Chairs · 2024-09-26

**Decision:**

Accept (Poster)

**Comment:**

This paper presents APEBench, a comprehensive JAX-based benchmark suite designed to evaluate autoregressive neural emulators for partial differential equations (PDEs). The benchmark includes 46 distinct PDEs across 1D, 2D, and 3D configurations, focusing on long-term temporal accuracy and differentiable physics training, distinguishing it from existing benchmarks like PDEBench and PDEArena.

Strengths:
- The submission makes a valuable contribution by offering a flexible, differentiable simulation framework integrated with advanced pseudo-spectral methods.
- APEBench provides extensive support for diverse PDE dynamics, facilitating a broad range of experiments across neural architectures.
Reviewers commended the clear documentation, the comprehensive dataset, and the emphasis on rollout metrics for temporal generalization, which enables deeper insights into emulator stability over time.
- The benchmark’s integration of neural-hybrid emulators, which leverages both learned and classical methods, is novel and offers potential for significant advances in scientific machine learning.

Weaknesses:
- Reviewers highlighted the need for clearer articulation of the research gap APEBench addresses, particularly in contrast to existing benchmarks. While the authors attempted to clarify this in their rebuttal, the distinction could be further emphasized in the final version.
- Some reviewers noted that the paper's presentation is dense, particularly in technical sections like the discussion of unrolled training strategies and PDE identifiers. Simplifying the language in key sections could enhance readability.
- The exclusive use of nRMSE as an evaluation metric was seen as a limitation. Reviewers recommended incorporating additional metrics, such as Sobolev losses, to provide a more nuanced assessment of emulator performance.

Overall Assessment: The paper offers a robust benchmark suite that fills a critical gap in the evaluation of autoregressive neural PDE emulators, particularly with its support for differentiable physics training. The inclusion of JAX-based architectures and the novel taxonomy for unrolled training strategies further distinguish it from existing benchmarks. However, the clarity of the manuscript, particularly regarding the research gap and certain technical explanations, could be improved to make it more accessible.